# CitySeeker: How Do VLMs Explore Embodied Urban Navigation with Implicit Human Needs?

**Siqi Wang**[1,2,3]**, Chao Liang**[4]**, Yunfan Gao**[3]**, Erxin Yu**[2]**, Sen Li**[2]**, Jing Li**[1,2,*]**Haofen Wang**[3*]

[1]PolyU Shenzhen Research Institute, Shenzhen, China
[2]Department of Computing, The Hong Kong Polytechnic University, Hong Kong SAR, China
[3]Tongji University, Shanghai, China
[4]Nanjing Institute of Geography and Limnology, Chinese Academy of Sciences, Nanjing, China

```
{siqi23.wang, erxin.yu}@connect.polyu.hk
liangchao@niglas.ac.cn, 2311821@tongji.edu.cn, slien@connect.ust.hk
jing-amelia.li@polyu.edu.hk, carter.whfcarter@gmail.com
```

## Abstract

Vision-Language Models (VLMs) have made significant progress in explicit instruction-based navigation; however, their ability to interpret *implicit human needs* (e.g., "I am thirsty") in dynamic urban environments remains underexplored. This paper introduces **CitySeeker**, a novel benchmark designed to assess VLMs' spatial reasoning and decision-making capabilities for exploring embodied urban navigation to address implicit needs. CitySeeker includes 6,440 trajectories across 8 cities, capturing diverse visual characteristics and implicit needs in 7 goal-driven scenarios. Extensive experiments reveal that even top-performing models (e.g., Qwen2.5-VL-32B-Instruct) achieve only 21.1% task completion. We find key bottlenecks in error accumulation in long-horizon reasoning, inadequate spatial cognition, and deficient experiential recall. To further analyze them, we investigate a series of exploratory strategies—Backtracking Mechanisms, Enriching Spatial Cognition, and Memory-Based Retrieval (BCR), inspired by human cognitive mapping's emphasis on iterative observation-reasoning cycles and adaptive path optimization. Our analysis provides actionable insights for developing VLMs with robust spatial intelligence required for tackling "last-mile" navigation challenges.

## 1 Introduction

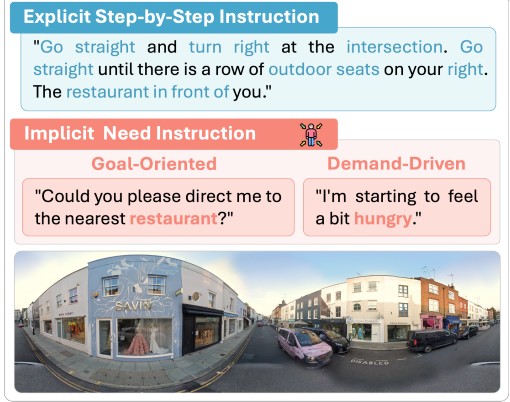

Figure 1: Navigation Instructions indicating **explicit needs** (top) and **implicit needs** (bottom).

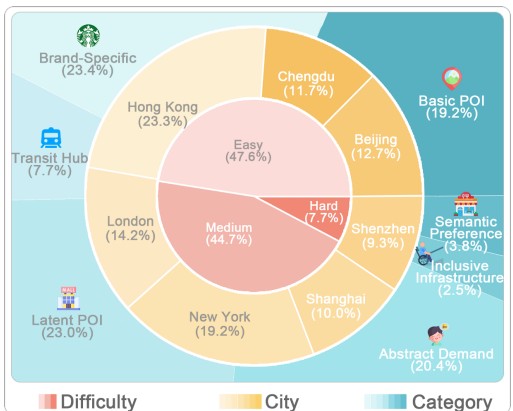

Figure 2: The statistics of **CitySeeker** with 6,440 trajectories in diverse scenarios.

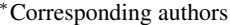

*Corresponding authors

Vision-Language Models (VLMs), with their advanced vision-grounded reasoning and language generation capabilities, are increasingly being applied to complex tasks like **Embodied Urban Navigation** (Zhang et al., 2024a). Autonomous embodied navigation in open urban environments is a cornerstone for realizing a new generation of intelligent services, leading to a rising demand for VLMs that can guide service robots, drones, or an AR assistant through urban settings. However, the capabilities of VLMs in this domain remain underexplored, as recent advances have focused on agents that follow explicit *step-by-step* instructions (e.g., "Proceed straight until the sculpture fountain, turn right, then continue until reaching McDonald's"). Such approaches, which we describe as **explicit needs**, rely on pre-constructed navigation directives rather than natural human commands, and face critical limitations in dynamic or novel urban scenarios (Wang et al., 2021; 2024b).

Table 1: Overview of Vision-Language Navigation datasets.

| Dataset | Instruction Type | Instruction | Environment | Source | City | Nodes | Avg.Length | Avg.Token |
|---|---|---|---|---|---|---|---|---|
| Talk the Walk (De Vries et al., 2018) | Explicit | 786 | GridWorld | 3D Rendering | 1 | 100 | 6.8 | 34.5 |
| Room-to-Room (Anderson et al., 2018) | Explicit | 21,567 | Indoor | Panoramas | 1 | 10,800 | 6.0 | 29.0 |
| Touchdown (Chen et al., 2019) | Explicit | 9,326 | Outdoor | Street View | 1 | 29,641 | 35.2 | 89.6 |
| Talk2Nav (Vasudevan et al., 2021) | Explicit | 10,714 | Outdoor | Panoramas and Map | 1 | 21,233 | 40.0 | 68.8 |
| StreetNav (Jain et al., 2024) | Explicit | 644,415 | Outdoor | Street View | 2 | - | 1,194m | 7.13 |
| map2seq (Schumann & Riezler, 2020) | Explicit | 7,672 | Outdoor | OpenStreetMap | 1 | 29,641 | 40.0 | 55.1 |
| **CitySeeker** | **Implicit** | **6,440** | **Outdoor+Dynamic** | **Street View and Map** | **8** | **41,128** | **18.3** | **11.11** |

In contrast, real-world human instructions often involve *abstract goals* that pertain to **implicit needs** (Zhou et al., 2024b). These needs are often unannotated on traditional maps or lack the granularity needed for last-mile navigation, and are implicit on multiple aspects: **functional** (e.g., when finding a restroom, recognizing the affordance that one is also available in a McDonald's, or inferring that "I'm thirsty" can be resolved by a convenience store or even a public water fountain), **spatial** (e.g., upon seeing a more visually prominent landmark like a shopping complex, inferring that a Starbucks is likely nearby, or realizing that a cinema is often hidden inside a mall), and **semantic** (e.g., subjective qualities like a "romantic" or "upscale" restaurant). Addressing these common needs is a fundamental challenge for goal-oriented navigation, requiring an agent to ground environmental intent through active exploration and spatial reasoning to solve the crucial "last-mile" problem, especially in contexts like city walks (Gao et al., 2024). We illustrate this concept in Figure 1.

The ability to understand human needs in urban space is fundamental to Embodied Urban Navigation (Liu et al., 2024; Hirose et al., 2023). While prior work has explored VLM's ability to interpret human needs, these efforts have been largely confined to constrained settings like indoor environments and 3D games (Wu et al., 2024a; Hu et al., 2025; Zhang et al., 2024b). This raises a critical scientific question: *Can VLMs develop intrinsic spatial awareness (Yang et al., 2024) for Embodied Navigation in open-world urban settings to address implicit human needs*? This task introduces unique challenges: (1) **Dynamic Visual Complexity**, with diverse and changing road networks and storefronts; (2) **Free-Form Instruction Parsing** for goals in flexible language; and (3) **Long-Horizon Reasoning** across extensive distances. The latter not merely about path length but requires robust, multi-hop reasoning that couples semantic inference with visual grounding. For instance, to address "I need a temporary place with Wi-Fi to work," an agent must ground the abstract function of "working." While it may infer POI categories like Cafe or Library, the final decision depends on visually grounding a location's *suitability* by dynamically searching for real-world cues—such as a storefront's ambiance, patrons with laptops, or even a "Free Wi-Fi" sign—that confirm it meets the user's need in real time.

Humans address these challenges using *cognitive maps*—mental representations of spatial relationships and environmental attributes (Epstein et al., 2017; Tolman, 1948). By combining observation with prior knowledge, humans can dynamically update their spatial understanding, infer latent properties, and formulate grounded plans from abstract goals—capabilities yet to be replicated in VLMs for outdoor navigation (Clemenson et al., 2021; Farzanfar et al., 2023; Momennejad et al., 2024; Wu et al., 2024b). Inspired by this, we propose **CitySeeker**, a novel benchmark for autonomous embodied urban navigation. It assesses **Implicit-Need-Driven Visual Grounding**: the process of translating an implicit need into a concrete visual search by using semantic inference to infer possible targets and grounding this understanding in a continuous stream of observations. To systematically probe this ability, its 7 task categories represent a spectrum of varying cognitive difficulty, from direct recognition ("Basic POI") to highly abstract reasoning ("Abstract Demand," "Semantic Preference"). The benchmark is implemented through 6,440 trajectories across 8 globally distributed urban regions with diverse layouts and visual characteristics (Table 1).

Our extensive experiments reveal that existing VLMs generally underperform, exhibiting significant trajectory deviation and deficient spatial cognition. Building on these benchmark findings, we

further investigate advanced strategies that endow the agent with human-like cognitive capabilities. We propose and analyze a triad of exploratory approaches: **B**acktracking mechanisms mimic self-correction, Spatial **C**ognition Enrichment mimics mental map building, and Memory-Based **R**etrieval mirrors recalling past experiences. These **BCR** strategies offer a concrete roadmap to elevate VLM spatial intelligence. From a cognitive science perspective, our research is aligned with exciting developments in AI concerning **Spatial Mental Models** within LLMs (Momennejad et al., 2024; Yang et al., 2024; Wu et al., 2024b), probing the intrinsic spatial intelligence of these models. In summary, our contributions are threefold:

• CitySeeker is the first large-scale benchmark for embodied urban navigation that addresses implicit needs across diverse multi-city settings, incorporating real-world visual diversity, long-horizon planning, and unstructured instructions.

• We design a VLM-based framework and a suite of human-inspired cognitive strategies (BCR) that translate implicit needs into multi-step plans through iterative observation-reasoning cycles.

• Through extensive experiments, we surface key bottlenecks in VLM spatial reasoning and crystallize them—and their fixes—into the BCR triad of Backtracking, Cognitive-map enrichment, and Retrieval-augmented memory, offering insightful guidance for advancing spatial intelligence.

## 2 RELATED WORK

**Spatial Cognition and Mental Models.** The concept of spatial cognition—internal mental representations of space—originates from cognitive science and has long been considered fundamental to how humans navigate (Tolman, 1948; Epstein et al., 2017). A recent paradigm shift has been spurred by findings that large models can develop emergent "world models" and representations of space and time without supervised training (Gurnee & Tegmark, 2023). This has inspired a new line of inquiry dedicated to probing the intrinsic spatial intelligence of these models (Yang et al., 2025; Yin et al., 2025; Feng et al., 2025; Manh et al., 2025). Our work provides the first attempt to systematically evaluate the emergent spatial cognition of VLMs through complex urban navigation tasks.

**Vision-Language Navigation (VLN).** Research in VLN has explored diverse instruction formats, including step-by-step, dialog-based (Goetting et al., 2024; Zhou et al., 2024a), and **goal- or intention-oriented instructions** (Wang et al., 2023; Qi et al., 2020a; Chiang et al., 2024; Liu et al., 2025). However, the majority of outdoor VLN research, such as Touchdown (Chen et al., 2019), have focused on agents following explicit step-by-step directives. This has led to the development of various methods (Xu et al., 2024; Schumann et al., 2024; Dorbala et al., 2022; Pan et al., 2024; Zhao et al., 2025) —excel at direct text-visual matching but are less equipped for the abstract reasoning required by implicit goals. Moreover, many solutions are limited by a reliance on static representations of predefined environment and poor generalizability (Cao et al., 2024; Elnoor et al., 2024; Zhan et al., 2024), revealing a gap in handling dynamic, real-world urban navigation. Goal-oriented navigation requires agents to use perception and commonsense reasoning to ground abstract concepts. Its form focuses on zero-shot open-world navigation (Mirowski et al., 2018; Majumdar et al., 2022; Zhou et al., 2024b), which has largely been limited to indoor environments. **CitySeeker** is the first benchmark to evaluate this goal-oriented reasoning for implicit needs in open-world cities.

## 3 CITYSEEKER BENCHMARK

The CitySeeker dataset comprises 6,440 route instances and corresponding natural language instructions covering 7 carefully curated categories of everyday human needs. These categories span typical requests from target recognition (e.g., finding nearby facilities or brands), contextual inference (e.g., inferring that a restroom is likely available in McDonald's) and attribute analysis (e.g., guiding to a bank with accessible facilities), to abstract and subjective reasoning (e.g., finding a restaurant suitable for a team gathering). The design of these categories is guided by three core criteria: (1) semantic complexity of goals, (2) spatial reasoning requirements, and (3) real-world applicability.

CitySeeker is collected from eight distinct metropolitan areas—Beijing, Shanghai, Shenzhen, Chengdu, Hong Kong, London, and New York—capturing diverse architectural layouts and dynamic street-level visuals. Overall, the benchmark includes over 41,128 street-view panoramas (from Google or Baidu Maps since 2024) to ensure realistic appearance variations. As shown in Figure 2,

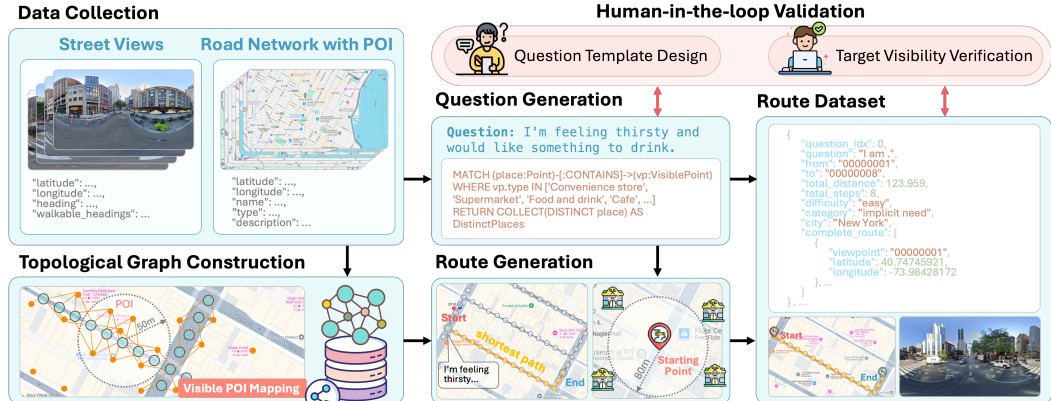

Figure 3: Construction Process of the CitySeeker Dataset.

our dataset balances instruction types, urban regions, and navigation difficulty (based on trajectory length). These categories collectively capture a broad range and diversity of real-world navigation challenges, with 19.2% involving POI navigation, 20.4% requiring the interpretation of abstract demands, and other major portions dedicated to brand-specific searches (23.4%) and latent POI discovery (23.0%). To facilitate quantitative evaluation, we further sampled 1,257 route instances as the final benchmark test set, balancing coverage across all metropolitan areas and categories.

## 3.1 BENCHMARK CONSTRUCTION

We employ a demand-driven pipeline to generate high-quality navigation routes at scale, as summarized in Figure 3. By integrating panoramic data, POI filtering, and graph-based distance queries, it yields high-fidelity routes spanning a broad spectrum of real-life urban navigation needs (see Appendix B for details).

**Data Collection and Topology Construction.** We gather street-level panoramas from both Google and Baidu Street View, covering diverse urban areas (e.g., Western cities, Hong Kong, Mainland China). To discretize each city's road network, we place a node every 20 meters and capture its panoramic imagery with metadata (e.g., latitude/longitude, heading, navigable headings). These nodes and edges represent navigable connections annotated with azimuths and distances, forming a graph-based structure stored in Neo4j. This layout facilitates spatial queries and ensures consistent global connectivity. Each collected Points of Interest (POI) is then linked to any vantage node within a 50-meter radius, reflecting the idea of a visible or discoverable place from that viewpoint. This step updates the knowledge graph with triplets of the form (Node) –has–> (VisiblePOI).

**Question Design and Generation**. To cover common daily needs, we first identify a set of high-frequency POI categories. These are sourced primarily from map providers (e.g., "restaurant," "coffee shop") and are supplemented with other visually rich POIs identified from street-view images and manually verified, capturing targets often unannotated on maps. This enables us to design specific templates for different query types. Each question type is manually associated with one or more POI categories; for instance, "I'm feeling thirsty" may map to beverage shops, water fountains, or cafés.

**Route Generation and Validation.** We generate navigation paths by selecting start and end nodes based on the intended POI category. We determine the starting nodes based on constraints to ensure that the shortest path falls within a controllable minimum radius (e.g., ensuring no other similar POIs lie within an overly small radius), thereby guaranteeing a meaningful navigation distance. The shortest paths between start and target nodes (containing the target POI) are then calculated using a shortest-path algorithm (e.g., A*). This process yields effective routes ranging from 5 to 25 steps. Finally, we manually verify each route by confirming the target POI is indeed visible or indirectly visible at the terminal node. To further empirically validate the rationality of our Need-to-POI mappings and mitigate potential designer bias, we conducted a cross-cultural human consensus survey ($N = 120$), which demonstrated a high degree of agreement (83.39%) with our ground truth categories (see Appendix B for details). Any route failing this check is refined or discarded.

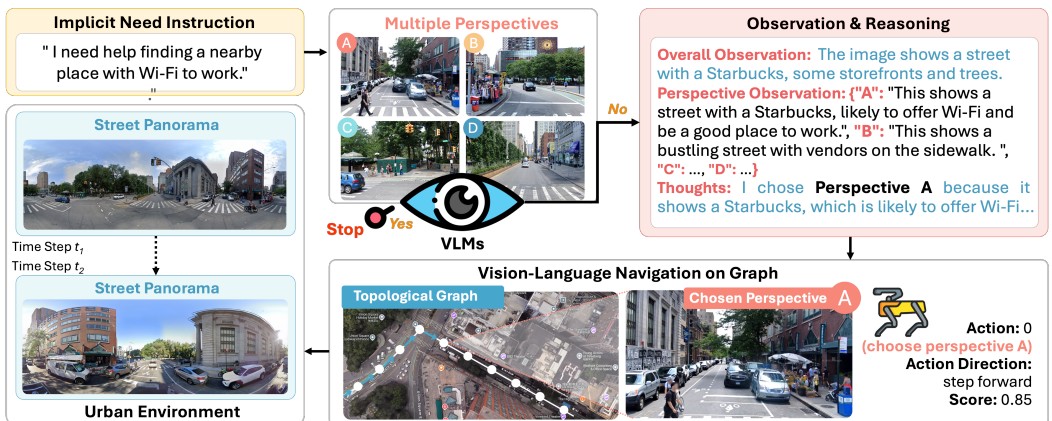

Figure 4: The CitySeeker Implicit-Need-Driven Embodied Urban Navigation Framework.

# 4 EVALUATION ON CITYSEEKER

## 4.1 OVERVIEW

**Task Formulation.** We formulate the VLN task within a navigation graph $\mathcal{G} = (\mathcal{V}, \mathcal{E})$, where $\mathcal{V}$ represents the set of georeferenced nodes and $\mathcal{E}$ denotes edges. At time step $t$, the agent occupies a node $v_t \in \mathcal{V}$. The agent receives a natural language instruction $\mathcal{W}$ and an observation set $\mathcal{O}_t = \{o_{t,1}, o_{t,2}, \ldots, o_{t,n}\}$, where $n$ is the number of perspective views at $v_t$. The agent maintains a state $s_t = (v_t, \mathcal{O}_t)$ representing current environmental context. The agent operates via a policy $\pi_\Theta$ that sequentially generates a reasoning rationale followed by an action: $(\Phi_t, a_t, c_t) = \pi_\Theta(\mathcal{W}, s_t)$. Here, $\Phi_t$ is the reasoning rationale, and $a_t \in \mathcal{A}_t = \{1, \ldots, n\}$ is the index of the selected perspective view with confidence $c_t \in [0, 1]$. The environment transitions to the next node $v_{t+1}$ based on the dynamics $T(v_t, a_t)$. A reasoning process is recorded as: $\tau = \{\mathcal{W}, (s_1, \Phi_1, a_1, c_1), \ldots, (s_T, \Phi_T, a_T, c_T)\}$, where the process terminates once the stop condition is met.

**Navigation Framework.** Figure 4 illustrates the navigation pipeline. At each time step, the panoramic image at the current viewpoint is subdivided into multiple perspective views, each corresponding to a feasible heading. Guided by a ReAct-style reasoning procedure (Yao et al., 2023), the VLM processes the current observation (Observe), infers a navigation intention (Think), selects one perspective view to move toward (Act), and finally outputs a confidence score as reflection (Reflect). This process iterates until the agent deems it has reached the goal or surpasses a maximum step limit (35 steps). To isolate the model's core spatial reasoning ability, we intentionally keep each step independent: the agent does not maintain persistent memory or feed previous internal states into subsequent decisions.

## 4.2 EVALUATION SETUP

**Benchmark Models.** We comprehensively evaluate 27 multi-image capable VLMs across diverse model families, encompassing various parameter scales and architecture. For proprietary models, we consider GPT-4o, GPT-4o-Mini, o4-mini, Gemini-1.5-Pro and Gemini-2.5-Pro (Team, 2024). For open-source models, we evaluate models from Qwen2-VL series (7/72B Instruct) (Wang et al., 2024a), Qwen2.5-VL series (7/32/72B Instruct) (Bai et al., 2025), InternVL2.5 (8/26/38B) (Chen et al., 2024), InternVL3 (8/14/38B), Llama-3.2 (11/90B Vision variants) (Dubey et al., 2024), Llama-4 (Scout-17B-16E-Instruct/Maverick-17B-128E-Instruct), LLaVA derivatives (Llama3-llava-next-8b, LLaVA-OneVision-Qwen2-7B) (Liu et al., 2023), MiniCPM derivatives (V-2.6/o-2.6) (Yao et al., 2024; Yu et al., 2024), Phi series (3.5-Vision-Instruct/4-Multimodal-Instruct) (Abdin et al., 2024; Microsoft, 2025) and MiniMax-01 (Li et al., 2025). All evaluations are conducted under zero-shot settings using unified prompts (see Appendix C.1 for full details).

**Metric Design.** We adopt a comprehensive evaluation protocol that extends standard VLN metrics (Qi et al., 2020b) to assess task success and navigation efficiency. Our evaluation includes: Task Completion (TC), measured with metrics of varying granularity: (1) TC-Exact (TCE) requires a strict, single-node endpoint match. To account for the fact that a target is often visible from multiple nearby viewpoints, (2) TC-Proximity (TCP) grants success within a geodesic threshold of $\leq 50m$ to address

Table 2: The performance of CitySeeker Framework. For Subcategory evaluations, the TCP score is reported. Top performers are highlighted in bold, while secondary leaders are underlined. For details on the AS metric and a more comprehensive evaluation of the models, please refer to Appendix C.3.

| Models | Overall | | | | | | Subcategories | | | | | | |
|---|---|---|---|---|---|---|---|---|---|---|---|---|---|
| | TCE | TCP | TCC | SPL | SPD | nDTW | Basic POI | Brand-Specific | Transit Hub | Latent POI | Abstract Demand | Inclusive Infra. | Semantic Pref. |
| GPT-4o | 2.4 | 18.3 | 6.8 | **13.3** | 125.4 | 136.9 | 18.9 | 21.1 | **19.6** | 9.7 | 18.9 | 14.9 | **26.0** |
| GPT-4o-mini | 1.1 | 12.3 | **7.6** | 7.5 | 202.0 | 325.2 | 12.3 | 21.1 | 14.3 | 7.4 | 11.8 | 7.5 | 10.6 |
| o4-mini | **2.6** | 17.9 | 6.8 | 11.6 | 130.1 | 156.3 | 20.2 | 19.9 | 12.5 | 13.1 | 16.7 | 9.0 | 24.0 |
| Gemini-2.5-pro | 1.8 | 17.3 | 5.0 | 12.1 | 121.8 | 121.2 | 20.4 | 18.0 | 16.1 | 10.2 | 20.6 | 9.0 | 13.5 |
| InternVL2.5-8B | 1.0 | 14.6 | 4.4 | 5.6 | 118.5 | 140.4 | 15.7 | 14.3 | 1.8 | **14.8** | 18.4 | 10.4 | 10.6 |
| InternVL2.5-26B | 1.6 | 15.3 | 3.7 | 7.3 | 109.5 | 106.0 | 20.4 | 12.4 | 8.9 | 10.8 | 17.1 | 7.5 | 8.7 |
| InternVL2.5-38B | 2.2 | 18.1 | 7.2 | 11.2 | 136.6 | 169.2 | 17.6 | 24.2 | 14.3 | 13.1 | 22.8 | 10.4 | 16.4 |
| InternVL3-8B | 1.3 | 15.8 | 4.9 | 6.4 | 118.3 | 144.5 | 17.2 | 15.5 | 7.1 | 14.2 | 16.7 | 16.4 | 15.4 |
| InternVL3-14B | 1.7 | 15.1 | 6.4 | 8.5 | 136.2 | 170.8 | 13.8 | 20.5 | 17.9 | 8.0 | 19.3 | 10.5 | 17.3 |
| InternVL3-38B | 2.5 | 19.3 | 6.7 | 10.6 | 115.8 | 128.3 | 18.9 | 25.5 | 12.5 | 13.1 | 23.3 | 11.9 | 21.2 |
| Qwen2-VL-7B | 0.7 | 11.1 | 1.7 | 5.2 | 111.4 | 114.4 | 11.4 | 9.9 | 14.3 | 4.6 | 13.2 | 13.4 | 15.4 |
| Qwen2-VL-72B | 1.0 | 11.9 | 2.3 | 9.0 | 113.0 | 89.5 | 14.8 | 8.7 | 7.1 | 7.4 | 14.5 | 1.5 | 15.4 |
| Qwen2.5-VL-7B | 0.5 | 15.8 | 4.3 | 4.6 | 119.0 | 151.8 | 20.2 | 17.4 | 12.5 | 10.2 | 13.6 | 7.5 | 15.4 |
| Qwen2.5-VL-32B | **2.6** | **21.1** | 6.2 | 12.7 | 122.6 | 147.0 | **22.2** | **30.4** | 10.7 | 14.2 | **24.1** | 9.0 | 20.2 |
| Qwen2.5-VL-72B | 2.0 | 14.6 | 7.2 | 9.1 | 174.9 | 250.2 | 14.8 | 21.7 | 14.3 | 6.8 | 15.8 | 11.9 | 14.4 |
| Llama3.2-90B | 0.9 | 12.5 | 3.7 | 7.3 | 124.5 | 123.1 | 13.6 | 12.4 | 1.8 | 9.7 | 14.0 | 10.4 | 16.4 |
| Llama-4-Scout-17B | 1.8 | 14.1 | 7.0 | 6.7 | 145.0 | 211.4 | 12.3 | 18.6 | 14.3 | 8.5 | 17.5 | **17.9** | 14.4 |
| Llama-4-Maverick-17B | 0.9 | 10.8 | 1.6 | 4.0 | 107.1 | 110.5 | 12.7 | 10.6 | 8.9 | 10.2 | 12.7 | 4.5 | 4.8 |
| MiniMax-01 | 1.5 | 13.6 | 6.8 | 8.8 | 172.2 | 236.6 | 15.5 | 16.2 | 8.9 | 6.8 | 14.9 | 11.9 | 13.5 |
| MiniCPM-V-2.6 | 0.9 | 11.7 | 3.5 | 4.0 | 122.2 | 152.2 | 12.5 | 10.6 | **19.6** | 9.7 | 13.2 | 7.5 | 8.7 |
| MiniCPM-o-2.6 | 1.4 | 15.5 | 6.4 | 5.6 | 130.1 | 176.0 | 14.4 | 22.4 | 10.7 | 12.5 | 18.9 | 11.9 | 12.5 |
| Phi-4-Multimodal | 0.6 | 9.2 | 1.1 | 7.1 | 101.1 | 58.1 | 14.0 | 6.2 | 5.4 | 2.8 | 10.5 | 7.5 | 2.9 |
| Llava-Llama3-8B | 0.3 | 10.4 | 0.8 | 5.1 | 104.8 | 86.9 | 14.8 | 8.7 | 5.4 | 8.0 | 8.8 | 4.5 | 7.7 |
| Llava-Qwen2-7B | 0.3 | 6.9 | 0.4 | 6.2 | **98.1** | **49.8** | 12.5 | 2.5 | 3.6 | 2.8 | 6.6 | 3.0 | 1.0 |
| Human | 5.7 | 30.1 | 13.5 | 21.2 | 143.5 | 178.6 | 31.8 | 36.5 | 34.9 | 19.7 | 31.5 | 16.7 | 29.8 |
| Random Choice | 0.7 | 13.9 | 3.2 | 3.8 | 112.4 | 128.3 | 16.6 | 10.6 | 5.4 | 10.8 | 14.9 | 16.4 | 13.5 |
| Forward Direction | 0.2 | 7.2 | 0.4 | 1.8 | 100.8 | 99.3 | 13.3 | 3.1 | 3.6 | 2.3 | 6.6 | 1.5 | 1.9 |

this spatial ambiguity; (3) TC-Category (TCC) evaluates whether the final destination belongs to the same category as the intended target, acknowledging practical flexibility in location-based tasks (e.g., reaching any restaurant rather than the closest one); Path Quality, assessed via (4) Normalized Dynamic Time Warping (nDTW) (Xu et al., 2024) to quantify trajectory alignment with ground truth, (5) Success weighted by Path Length (SPL) to rigorously measure navigation efficiency by balancing success rates against trajectory length, and (6) Average Steps (AS) for decision efficiency; Distance-Based Metrics, (7) Shortest-Path Distance (SPD) for straight-line distance to the goal.

**Reference Baselines.** To comprehensively assess model capabilities, we introduce three evaluation baselines. We establish a **(1) Human Baseline** using an interactive platform where 10 participants of diverse backgrounds performed the navigation tasks. We also include a **(2) Random Choice Baseline**, which selects a random direction at each step, and a **(3) Forward Direction Baseline**, which always chooses the forward direction as a simple heuristic.

## 5 MAIN RESULTS AND ANALYSIS

### 5.1 OVERALL PERFORMANCE

Table 2 presents the overall results. In general, the models exhibit relatively low success rates, particularly under the stricter criterion TCE. Larger models (e.g., Qwen2.5-VL-32B, GPT-4o, Gemini-2.5-Pro) perform slightly better—likely due to stronger internal representations—but the gains over smaller models remain modest. Notably, open-source models like the Qwen2.5-VL and InternVL3 series demonstrate competitive performance. This superiority likely stems from specific architectural and training alignments with CitySeeker's demands: Qwen2.5-VL benefits from spatial-aware SFT and efficient high-resolution processing crucial for identifying street-level details, while InternVL3 leverages native multimodal pre-training and Mixed Preference Optimization (MPO) to enhance the complex cross-modal reasoning required for implicit needs. However, no model consistently excels, with some even underperforming random baselines. Ablation study further revealed that providing agents with a global 2D map surprisingly degraded task completion, underscoring the challenge of

fusing map-based information with visual grounding (see Appendix D). Human participants achieved the best overall performance, outperforming all models on both TCP and TCC. They exhibit a clear advantage across diverse task dimensions, especially in tasks requiring urban commonsense like "Transit Hub" navigation (a TCP of 34.9% for humans versus 10.7% for the best VLM Qwen2.5-VL-32B) and "Basic POI" finding (31.8% vs. 22.2%).

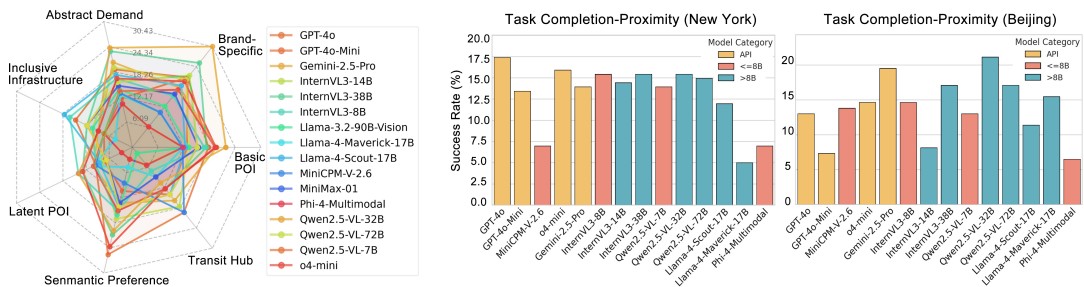

Figure 5: Radar Chart: TCP performance of different models across various subcategories. Bar Chart: TCP performance of models in different cities (left: New York, right: Beijing).

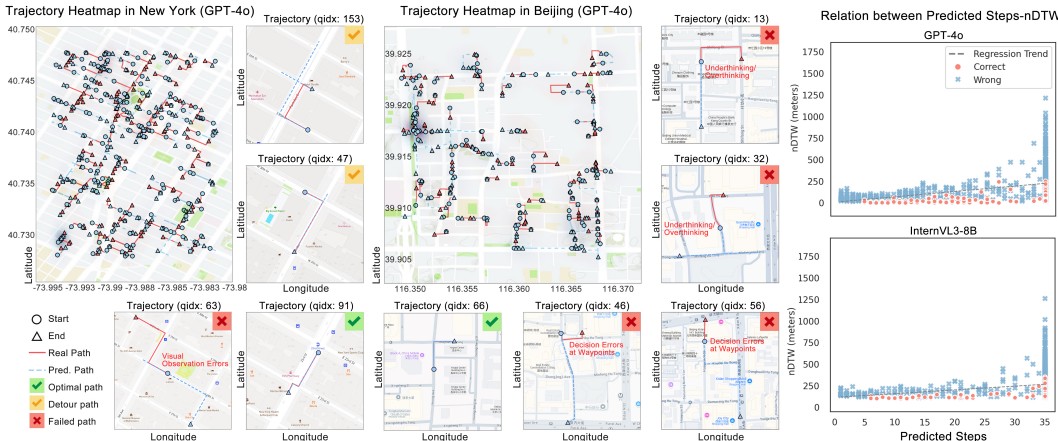

Figure 6: Heat Map: GPT-4o's navigation trajectory distribution in New York (left) and Beijing (right). Scatter Plot: Relationship between model-generated trajectory steps and nDTW.

**Task Category Analysis.** Breaking down performance by query category reveals a clear divide between tasks requiring direct recognition versus those demanding deeper inference, as shown in Table 2 and radar chart of Figure 5. Models performed best on **Brand-Specific** navigation, where recognizable brand names (e.g., Starbucks) serve as strong lexical and visual anchors. In sharp contrast, one of the most challenging categories was **Latent POI**, which requires agents to reason about indirectly observable targets that are not explicitly signed (e.g., inferring a restroom is inside a McDonald's). This finding highlights a key VLM limitation: while adept at direct recognition, they falter when tasks require nuanced, commonsense inference about the environment.

**Variation Across Cities.** Performance also varies across different metropolitan areas. As shown on bar chart of Figure 5, interestingly, GPT-4o variants perform poorly in Beijing but achieve highest scores in New York—this may reflect biases in the training data or the more grid-like street layouts in the United States. To test for linguistic bias, we conducted a cross-lingual experiment, which revealed that this performance gap is not primarily driven by language factors (see Appendix C.4 for details).

## 5.2 IN-DEPTH ANALYSIS

As shown in Figure 6, model performance tends to degrade with increased route length. When the number of steps is fewer than 20, nDTW values remain relatively small and correct trajectories are more common. However, at around 35 steps, nDTW metrics become highly scattered, indicating that longer horizons require robust sequential reasoning—an ability current VLMs often lack, as their errors accumulate without being integrated into a coherent spatial memory.

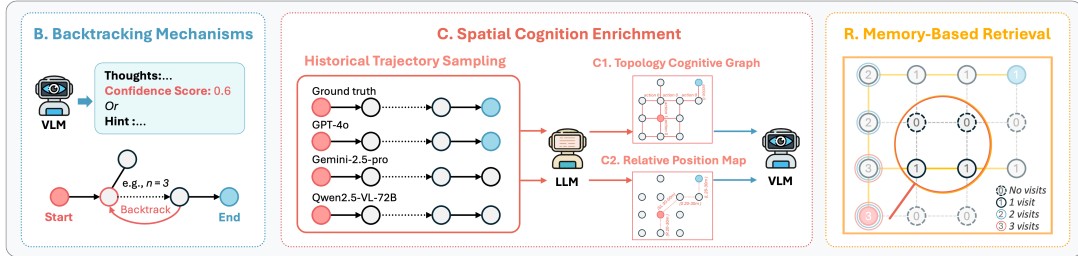

Figure 7: Overview of the **BCR** Approach -**B**acktracking, Spatial **C**ognition, and Memory-Based **R**etrieval to enhance VLM performance in urban VLN tasks.

Table 3: Performance of different models on TCP and nDTW under three strategies. The best results are highlighted in bold.

| Model | Baseline | | B1 | | B2 | | B3 | | C1 | | C2 | | R1 | | R2 | | R3 | |
|---|---|---|---|---|---|---|---|---|---|---|---|---|---|---|---|---|---|---|
| | TCP | nDTW | TCP | nDTW | TCP | nDTW | TCP | nDTW | TCP | nDTW | TCP | nDTW | TCP | nDTW | TCP | nDTW | TCP | nDTW |
| GPT-4o-Mini | 12.5 | 337.1 | 15.7 | 289.5 | 17.4 | 277.9 | 18.2 | 258.3 | 17.2 | 242.3 | 15.1 | 239.7 | 18.3 | **136.6** | 17.9 | 312.2 | 19.4 | 176.9 |
| InternVL3-8B | 15.0 | 142.9 | 16.7 | 139.9 | 18.5 | **137.5** | 17.2 | 138.6 | 17.9 | 145.2 | 16.4 | 197.6 | 18.6 | 148.9 | 16.9 | **128.4** | 17.9 | 134.8 |
| MiniCPM-V-2.6 | 11.3 | 159.0 | 9.1 | **120.5** | 13.1 | 153.1 | 13.0 | **137.2** | 11.6 | **123.0** | 8.5 | 144.4 | 14.6 | 143.3 | 14.8 | 146.7 | 15.7 | **111.5** |
| Qwen2.5-VL-7B-Instruct | 12.5 | **122.2** | 14.0 | 140.5 | 14.4 | 156.7 | 15.3 | 141.8 | 15.3 | 142.5 | 12.3 | **135.8** | 16.1 | 156.4 | 15.9 | 441.4 | 16.2 | 141.8 |
| Qwen2.5-VL-32B-Instruct | **19.9** | 167.7 | **21.9** | 164.5 | **22.2** | 154.6 | **23.1** | 159.4 | **21.2** | 153.5 | **19.6** | 162.1 | **26.9** | 173.2 | **25.9** | 172.8 | **25.4** | 152.8 |

**Trajectory Patterns.** In Figure 6, two prominent error modes emerge: (1) Trajectory Deviation. This arises from compounded errors at each sequential decision point, a problem exacerbated by sparse or ambiguous instructions. (2) Oscillatory Detours. Some open-source models exceed the optimal path length by 40–60% (e.g., trajectories #47 and #153 in New York), likely attributable to fragmented context handling and incomplete global awareness. We also observe that path efficiency correlates weakly with TC: when uncertain, some models veer off course or backtrack repeatedly, leading to wasted steps. Notably, most models demonstrate looping behavior—visiting the same node multiple times (e.g., trajectory #63 in New York and #32 in Beijing)—suggesting difficulties with local decision-making. Interestingly, others (e.g., Llava-Qwen2-7B) take relatively concise routes yet still show low accuracy, implying a lack of deeper spatial understanding despite fewer detours.

## 6 EXPLORATORY APPROACHES TO EMBODIED URBAN NAVIGATION

To explore strategies for enhancing VLM performance, we propose a three-pronged approach, BCR, involving **Backtracking Mechanisms**, **Spatial Cognition Enrichment**, and **Memory-Based Retrieval** (Figure 7). These methods aim to mitigate cumulative navigation errors, boost global spatial awareness, and enable memory-informed decision-making. We denote these three method series as **B**, **C**, and **R** respectively in the following content. We conducted initial experiments on a mini-size subset of 650 samples; for full technical details, including an analysis of combined BCR strategies, please refer to Appendix E.

### 6.1 BACKTRACKING MECHANISMS

We introduce backtracking to correct navigational errors. **(B1) Basic Backtracking** is triggered when the agent's internal confidence, averaged over a sliding window of $k$ steps, falls below a predefined threshold $\theta$. The agent then reverts to the last trusted node. This mechanism is self-supervised and does not require external feedback. **(B2) Step-Reward Backtracking** replaces subjective confidence with objective progress metric: the topological distance to the goal, $d_t$. Backtracking is initiated if this distance increases monotonically for $k$ consecutive steps, i.e., when $\bigwedge_{i=0}^{k-1}(d_{t-i} > d_{t-(i+1)})$ holds. This corrects navigational drift away from the target. **(B3) Human-Guided Backtracking** enhances B1 by providing a directional hint after reverting. This hint guides the agent toward the optimal action $a^* = \arg\min_{a \in \mathcal{A}_t} \mathbb{E}[d_{t+1}|a_t = a]$, which minimizes the expected future distance to the target, thereby realigning the agent's trajectory with the shortest path to the goal.

**Empirical Performance.** As shown in Table 3, **B1** generally improves TCP across models, indicating that confidence-based backtracking stabilizes navigation for capable VLMs. However, it underperforms on smaller models like MiniCPM-V-2.6 (TCP drops from 11.3% to 9.1%), suggesting insufficient spatial reasoning for effective self-assessment. **B2** and **B3** are more universally effective, consistently improving TCP across all models. For instance, **B3** boosts GPT-4o-Mini's TCP to 18.2% while significantly improving its path efficiency (nDTW drops from 337.1 to 258.3). These results highlight that while simple self-correction benefits capable models, external signals (like progress rewards or timely hints, in B2 and B3) offer more robust improvements.

## 6.2 SPATIAL COGNITION ENRICHMENT

To improve the agent's environmental awareness, we generate enriched spatial cues by using GPT-4.1 to synthesize successful and erroneous trajectories from various VLMs (e.g., GPT-4o, Gemini-2.5-pro). This synthesized knowledge is then injected into the agent's prompt in one of two formats. **(C1) Topology Cognitive Graph** provides a structured graph where nodes represent locations and edges represent actionable transitions. This format forces the agent to ground its decisions in explicit environmental connectivity, discouraging exploration of invalid paths. **(C2) Relative Position Map** offers a more intuitive spatial context, describing relationships between locations with directional cues (e.g., "left", "slightly right") and estimated distances. While lacking explicit connectivity, this method allows the agent to construct a flexible mental map based on relative positional relationships.

**Empirical Performance.** Table 3 illustrates that **C1** improves TCP across models by grounding decisions in topological structure. For instance, GPT-4o-Mini achieves an increase from 12.5% to 17.2%. The impact of **C2** is more mixed; while it can improve path efficiency (e.g., for GPT-4o-Mini, nDTW drops to 239.7), it sometimes slightly reduces task completion (e.g., for MiniCPM-V-2.6, TCP drops from 11.3% to 8.5%). Overall, C1 appears more reliable for improving task success, whereas C2 supports a more flexible and exploratory approach but sacrifices some success rate.

## 6.3 MEMORY-BASED RETRIEVAL

To overcome fragmented decision-making, we implement a graph-based memory module, enabling agents to retrieve or append past rationales and actions. **(R1) Topology-based Retrieval** aggregates multi-round node and edge metadata based on graph connectivity. At each step, it retrieves a local subgraph of $h$-hop neighbors, accessing metadata like node visitation counts, previous decisions, edge transition success rates and confidence scores. This helps avoid repeating past mistakes and promotes the reuse of successful paths. **(R2) Spatial-based Retrieval** complements this by retrieving a subgraph of nodes and relationships within a fixed Euclidean radius, which emphasizes geographic proximity. **(R3) Historical Trajectory Lookup** introduces short-term memory by appending recent intra-round navigation history into VLM's context. This includes spatial data, action choices, and prior decision rationale over a sliding window. Unlike R1 and R2, R3 does not rely on database traversal, making it a lightweight method to stabilize reasoning within a single episode.

**Empirical Performance.** Table 3 reveals that the R-series strategies are the most powerful overall. For GPT-4o-Mini, **R3** yields the highest TCP of all strategies (19.4%), while **R1** achieves the best path efficiency with a dramatic nDTW reduction to 136.6. This demonstrates that different memory-based approaches can simultaneously enhance both task completion and navigation efficiency. Across all models, R-series strategies consistently produce some of the highest TCP scores, with **R1** pushing the top-performing Qwen2.5-VL-32B-Instruct to an impressive **26.9%** TCP. This confirms that memory-aware navigation is crucial for improving both the reliability and effectiveness of the agent.

**Trade-off Analysis and Application Scenarios.** The **B-Series** strategies (especially B2 and B3) reliably improve task completion, making them suitable when success is prioritized. The **C-Series** is applicable when external spatial information can be provided; C1 generally favors accuracy, while C2 may trade success for better path efficiency. Overall, the **R-Series** is the most robust approach for long-horizon navigation, consistently yielding the best task completion rates. Lightweight models like InternVL3-8B are suitable for time-sensitive tasks, while heavier models like Qwen2.5-VL-32B-Instruct, augmented with R-series strategies, are ideal for precision-critical applications.

## 7    CONCLUSION

CitySeeker establishes a new standard for embodied urban navigation by systematically evaluating how well VLMs tackle implicit human needs in diverse, real-world cityscapes. Our extensive experiments uncover that today's models frequently struggle with the complexities of this task, often deviating from optimal routes due to limited spatial reasoning. To address this, we introduced a human-inspired cognitive framework and outlined three insightful strategies—backtracking, spatial cognition enrichment, and memory-based retrieval—which demonstrably improve performance. These findings offer a clear pathway toward developing more robust, human-like agents capable of solving the crucial "last-mile" problem in real-world navigation.

## ACKNOWLEDGMENTS

This work is supported by the National Natural Science Foundation of China (Grant No. U23B2057), the STIC Shenzhen Natural Science Foundation (No. JCYJ202506041913329039), a grant from the Research Grants Council of the Hong Kong Special Administrative Region, China (Project No. PolyU/25200821), the Shanghai Pilot Program for Basic Research (Grant No. 22TQ1400300), and the Innovation and Technology Fund (Project No. PRP/047/22FX). We also sincerely thank Dr. Yushi Li for insightful comments and revisions, which significantly improved this paper.

## ETHICS STATEMENT

The study involved human participants for the establishment of a human baseline. All participants were fairly compensated for their time in line with our institution's standard rates for research assistance, and the experimental procedure was clearly explained to them. The street-view imagery used in our experiments was accessed via publicly available APIs (Google Maps and Baidu Maps). To strictly adhere to the API Terms of Service, we do not distribute any raw street-view images or offline copies. Instead, researchers can access the visual data using our provided scripts, which query the official APIs using the released trajectory metadata (e.g., Panorama IDs and coordinates). This approach ensures full compliance while maintaining the scientific reproducibility of the benchmark.

## REPRODUCIBILITY STATEMENT

To ensure the reproducibility of our work while complying with data usage policies, we have made our metadata, code, and supplementary materials publicly available. The **CitySeeker** benchmark release includes: (1) the complete trajectory graph data (nodes, edges, and Panorama IDs), and (2) the full implementation code for our evaluation framework and BCR exploratory strategies. The scripts enable re-fetching the necessary street-view imagery directly from the public APIs. This methodology aligns with established practices in the field (e.g., Touchdown (Chen et al., 2019)) to ensure reproducibility without distributing copyrighted images. Furthermore, we have verified that reproducing test set via these scripts incurs no financial cost under standard API usage tiers. All resources can be found at anonymous repository linked in Appendix A. The main paper and appendix provide comprehensive details on our experimental setup (Appendix C), the exact prompts used for all VLM evaluations (Appendix C.2), and the specifics of our evaluation metrics (Section 4.2).

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

APPENDIX

## A  DATA AND CODE AVAILABILITY

To promote community engagement and ensure reproducibility, the **CitySeeker** dataset along with the full implementation code are publicly available at:
`https://github.com/anonymous-cityseeker/CitySeeker.`

## B  BENCHMARK DETAILS

The CitySeeker dataset comprises 6,440 trajectories across 8 diverse urban regions, paired with navigation instructions addressing implicit human needs. To ensure strict adherence to image licensing and usage rights (e.g., Google Maps Platform Terms of Service), our public release strategy dissociates raw visual data from trajectory metadata. Specifically, we release the complete trajectory graphs (including node coordinates and Panorama IDs) and navigation instructions. Rather than distributing offline copies of copyrighted street-view imagery, we provide automated scripts that allow researchers to re-fetch the necessary visual data directly from the official APIs. This methodology ensures full compliance with non-commercial, research-only usage policies while maintaining the benchmark's reproducibility. Route-map visualizations are generated from OpenStreetMap data (ODbL) and shared under compatible terms.

### B.1  IMPLICIT NEEDS INSTRUCTION GENERATION PROCESS

Each instruction type is manually associated with one or more Point of Interest (POI) categories. For **Basic POI Navigation**, **Brand-Specific Navigation**, and **Transportation Hub Navigation** categories, the corresponding POI types are directly indicated in the questions; For **Inclusive Infrastructure Navigation** and **Semantic Preference Navigation**, we leverage POI metadata descriptions provided by map vendors; For **Latent POI Navigation** and **Abstract Demand Navigation** questions, we manually define the mapping between the questions and POI categories based on everyday life experience. Table 4 and 5 present some mapping examples.

Table 4: Latent POI Navigation Mapping Examples

| User Query | Mapped POI Categories |
|---|---|
| *"Please find the nearest cinema."* | `Movie theater`, `Shopping mall` |
| *"Please find the nearest ATM."* | `ATM`, `Bank` |
| *"Please find the nearest coffee shop."* | `Cafe`, `Coffee shop`, `Shopping mall` |
| *"Please find the nearest gym."* | `Physical fitness program`, `Gym`, `Fitness center`, `Gym and fitness centre` |
| *"Please find the nearest parking lot."* | `Parking lot`, `Free parking lot`, `Parking garage`, `Public parking space` |
| *"Please find the nearest restroom."* | `Public bathroom`, `Public wheelchair-accessible bathroom`, `Subway station`, `Shopping mall`, `McDonald's`, `KFC` |

Table 5: Abstract Demand Navigation Mapping Examples

| User Query | Mapped POI Categories |
|---|---|
| *"I need to go to the airport and would like assistance in finding the best way there. Could you help me?"* | `Subway station`, `Bus station`, `Bus stop`, `Taxi service`, `Car leasing service`, `Car rental agency` |
| *"I want to bring my child to play and need assistance finding a suitable place nearby. Could you help me?"* | `Shopping mall`, `Park`, `City park`, `Museum`, `Art museum`, `Children's amusement center`, `Playground` |
| *"I want to buy fruits and vegetables and need assistance finding a suitable place nearby. Could you help me?"* | `Supermarket`, `Fruit and vegetable store`, `Greengrocer`, `Farmers' market` |
| *"I want to exercise and need assistance finding a suitable place nearby."* Could you help me? | `Park`, `City park`, `Physical fitness program`, `Gym`, `Yoga studio`, `Swimming pool` |
| *"I want to rest and read and need help finding a suitable place nearby. Could you assist me?"* | `Cafe`, `Coffee shop`, `Public library`, `Library`, `Book store`, `Park` |
| *"I want to work with Wi-Fi and need assistance finding a suitable place nearby. Could you help me?"* | `Cafe`, `Coffee shop`, `Public library`, `Library`, `Book store`, `McDonald's`, `KFC`, `Five Guys` |
| *"I have run out of phone credit and need to recharge it. Could you assist me in finding a nearby place where I can do so?"* | `Cafe`, `Coffee shop`, `Public library`, `Library`, `Supermarket`, `Telecommunications service provider`, `Cell phone store`, `Target`, `McDonald's`, `KFC`, `7-ELEVEn`, `Best Buy` |
| *"I'm feeling hungry and would like something to eat. Could you help me find a nearby place?"* | `Convenience store`, `Supermarket`, `Market`, `Dessert shop`, `Food truck`, `Food stall`, `Shopping mall`, `Restaurant`, `Diner` |
| *"I'm feeling thirsty and would like something to drink. Could you help me find a nearby place?"* | `Convenience store`, `Supermarket`, `Shopping mall`, `Cafe`, `Bubble tea store`, `Water fountain`, `Vending machine`, `Dessert shop`, `Juice shop` |
| *"I'm not feeling well and need assistance finding a suitable place nearby. Could you help me?"* | `Clinics`, `Pharmacies`, `Hospitals` |

**Enrichment with Visually-Grounded POIs.**  To enrich our POI beyond standard map data, we address the challenge of temporary or purely visual POIs not available from map providers. We employ a powerful VLM to identify such facilities (e.g., temporary food stalls or vendors) within street-views. To ensure data quality, all identified POIs then undergo a manual verification process. Once confirmed, these visual POIs are added to our graph database with their corresponding locations, making them available for the query-to-POI mapping process. This step is crucial for ensuring our benchmark reflects the dynamic and visually rich nature of urban environments.

**Query POI Cypher Statement Example.**  To retrieve specific POI instances, we construct Cypher queries that consider category attributes, name-based keyword patterns, or descriptive information within the graph database. The queries are adjusted based on the different POI categories provided by various map vendors, ensuring flexibility in retrieving relevant POIs efficiently. For Inclusive Infrastructure and Semantic Preference, we query the corresponding description information for each POI, which is stored in the retrieved POI metadata. This includes descriptive terms that specify human-centric attributes of the POI, such as romantic, upscale, family-friendly, cozy, outdoor seating, wheelchair accessible entrance, etc. The Cypher query combines these descriptive words with category-specific terms, as shown below.

```
Cypher Query Example - Find the nearest group or family-friendly restaurant.

MATCH (place:Point)-[:CONTAINS]->(vp:VisiblePoint)
WHERE (ANY(key IN keys(vp)
              WHERE (key CONTAINS 'introduction' OR key CONTAINS '
                  result') AND
                  (vp[key] CONTAINS 'Groups'
                   OR vp[key] CONTAINS 'Family-friendly')))
                  AND (vp.category CONTAINS 'restaurant'
                      OR vp.category CONTAINS 'diner')
RETURN COLLECT(DISTINCT place) AS validPOIs
```

For Abstract Demand, a query such as "I want to rest and read" triggers a Cypher query that retrieves POIs from categories like `Cafe`, `Coffee Shop`, `Public Library`, `Library`, `Book Store`, and `Park`, while simultaneously considering relevant keywords like "park" and "book store" in the POI names. This dual consideration of category and keywords ensures that the query matches a comprehensive set of potential POIs. The Cypher query is as follows:

```
Cypher Query Example - I want to rest and read and need help finding a suitable place nearby.

MATCH (place:Point)-[:CONTAINS]->(vp:VisiblePoint)
WHERE vp.category IN ['Cafe', 'Coffee shop', 'Public library',
                      'Library', 'Book store', 'Park']
   OR any(keyword IN ['park', 'book store']
       WHERE toLower(vp.name) CONTAINS toLower(keyword))
RETURN COLLECT(DISTINCT place) AS validPOIs
```

**Manual Trajectory Validation.**  To mitigate potential mapping incompleteness, we implement manual verification to re-inspect generated trajectories, ensuring no alternative POIs that could fulfill the user's request exist along the path except at the designated endpoint. By doing so, we ensure that the generated trajectories are comprehensive and accurate, reducing the likelihood of missing target POIs or introducing redundant POIs that could lead to incorrect ground truth.

To further prove that our common sense mappings are not designer bias but reflect robust human common sense with cross-cultural generalization, we conducted a new supplementary survey. We surveyed 120 participants across four regions (30 North America, 40 Asia, 30 Europe, 20 Other). We presented our 10 abstract queries (e.g., "I'm feeling thirsty..."). For each question, participants selected from 50 POI categories. These options included our pre-defined mapping categories mixed into a large, systematic pool of ∼40 other POI categories drawn from the entire Google/Baidu POI classification system.

The results show a massive consensus cliff. Our pre-defined mapping options received a Global Average Consensus of **83.39%**, while the unselected other POI categories received a Global Average Consensus of only **1.90%**. This 81% gap proves that our ground truth mappings are not designer

Table 6: Full Cross-Cultural Consistency Statistics (Part 1). Survey results ($N = 120$) validating Need-to-POI mappings. SD indicates Cross-Cultural Standard Deviation.

| User Query | POI Category | Statistics (Mean / SD) |
|---|---|---|
| *"I'm feeling hungry and would like something to eat. Could you help me find a nearby place?"* | Supermarket | Mean: 88.89% (SD: 7.70%) |
| | Restaurant | Mean: 88.33% (SD: 7.26%) |
| | Diner | Mean: 85.28% (SD: 8.01%) |
| | Convenience store | Mean: 80.56% (SD: 4.19%) |
| | Shopping mall | Mean: 73.89% (SD: 7.88%) |
| | Market | Mean: 64.44% (SD: 21.43%) |
| | Food stall | Mean: 60.56% (SD: 4.19%) |
| | Dessert shop | Mean: 50.83% (SD: 11.27%) |
| | Food truck | Mean: 46.67% (SD: 25.17%) |
| *"I'm feeling thirsty and would like something to drink. Could you help me find a nearby place?"* | Convenience store | Mean: 91.94% (SD: 5.02%) |
| | Cafe | Mean: 87.50% (SD: 4.64%) |
| | Vending machine | Mean: 80.28% (SD: 2.93%) |
| | Supermarket | Mean: 79.17% (SD: 3.63%) |
| | Water fountain | Mean: 66.11% (SD: 18.58%) |
| | Bubble tea store | Mean: 63.06% (SD: 29.58%) |
| | Dessert shop | Mean: 61.94% (SD: 8.83%) |
| | Juice shop | Mean: 55.83% (SD: 8.46%) |
| | Shopping mall | Mean: 45.00% (SD: 5.00%) |
| *"I'm not feeling well and need assistance finding a suitable place nearby. Could you help me?"* | Pharmacies | Mean: 96.11% (SD: 3.47%) |
| | Hospitals | Mean: 87.50% (SD: 4.33%) |
| | Clinics | Mean: 82.50% (SD: 7.95%) |
| *"I want to rest and read and need help finding a suitable place nearby. Could you assist me?"* | Public library | Mean: 96.11% (SD: 3.47%) |
| | Park | Mean: 94.44% (SD: 3.85%) |
| | Library | Mean: 89.17% (SD: 1.44%) |
| | Coffee shop | Mean: 83.33% (SD: 8.82%) |
| | Cafe | Mean: 82.22% (SD: 3.85%) |
| | Book store | Mean: 75.56% (SD: 7.70%) |
| *"I want to work with Wi-Fi and need assistance finding a suitable place nearby. Could you help me?"* | Cafe | Mean: 94.17% (SD: 2.20%) |
| | Coffee shop | Mean: 93.89% (SD: 3.47%) |
| | Library | Mean: 91.94% (SD: 1.73%) |
| | Public library | Mean: 91.39% (SD: 5.55%) |
| | Book store | Mean: 66.67% (SD: 3.33%) |
| | McDonald's | Mean: 66.11% (SD: 9.77%) |
| | KFC | Mean: 63.06% (SD: 8.35%) |
| | Five Guys | Mean: 50.56% (SD: 17.35%) |
| *"I have run out of phone credit and need to recharge it. Could you assist me in finding a nearby place where I can do so?"* | Cell phone store | Mean: 93.61% (SD: 3.76%) |
| | Telecom.  service provider | Mean: 90.28% (SD: 2.93%) |
| | 7-ELEVEn | Mean: 82.78% (SD: 17.02%) |
| | Supermarket | Mean: 77.22% (SD: 2.55%) |
| | Convenience store | Mean: 74.72% (SD: 6.47%) |
| | Best Buy | Mean: 62.50% (SD: 34.59%) |
| | Cafe | Mean: 61.94% (SD: 5.02%) |
| | Library | Mean: 55.83% (SD: 15.07%) |
| | Coffee shop | Mean: 54.44% (SD: 5.09%) |
| | Target | Mean: 51.39% (SD: 42.95%) |
| | Public library | Mean: 48.61% (SD: 10.55%) |
| | McDonald's | Mean: 46.67% (SD: 11.55%) |
| | KFC | Mean: 45.00% (SD: 13.23%) |

Table 7: Full Cross-Cultural Consistency Statistics (Part 2).

| User Query | POI Category | Statistics (Mean / SD) |
|---|---|---|
| *"I want to exercise and need assistance finding a suitable place nearby. Could you help me?"* | Park | Mean: 95.00% (SD: 1.67%) |
| | Gym | Mean: 91.67% (SD: 6.01%) |
| | City park | Mean: 87.50% (SD: 6.61%) |
| | Physical fitness program | Mean: 76.11% (SD: 6.74%) |
| | Swimming pool | Mean: 73.33% (SD: 11.55%) |
| | Yoga studio | Mean: 64.44% (SD: 7.70%) |
| *"I want to bring my child to play and need assistance finding a suitable place nearby. Could you help me?"* | City park | Mean: 94.17% (SD: 2.20%) |
| | Playground | Mean: 91.94% (SD: 8.83%) |
| | Park | Mean: 88.89% (SD: 1.92%) |
| | Museum | Mean: 85.83% (SD: 2.20%) |
| | Children's amusement center | Mean: 81.67% (SD: 2.89%) |
| | Shopping mall | Mean: 65.83% (SD: 10.64%) |
| | Art museum | Mean: 65.00% (SD: 1.67%) |
| *"I need to go to the airport and would like assistance in finding the best way there. Could you help me?"* | Subway station | Mean: 92.22% (SD: 6.94%) |
| | Bus stop | Mean: 80.83% (SD: 8.78%) |
| | Taxi service | Mean: 78.89% (SD: 1.92%) |
| | Bus station | Mean: 78.61% (SD: 10.55%) |
| | Car leasing service | Mean: 67.50% (SD: 4.64%) |
| | Car rental agency | Mean: 65.28% (SD: 6.47%) |
| *"I want to buy fruits and vegetables and need assistance finding a suitable place nearby."* | Supermarket | Mean: 97.22% (SD: 2.55%) |
| | Fruit and vegetable store | Mean: 90.28% (SD: 2.93%) |
| | Farmers' market | Mean: 79.72% (SD: 11.56%) |
| | Greengrocer | Mean: 75.00% (SD: 18.33%) |

bias but are the clear, emergent human common sense that surfaces from the noise. Furthermore, the consensus was highly consistent on average. The Average Cross-Cultural Standard Deviation (SD) for our ground truth options was only **8.40%** (NA, EA, EU). This low SD value quantitatively proves high overall consistency across cultural regions. Tables 6 and 7 present the detailed statistics for questions.

## B.2 INSTRUCTION CATEGORIES AND EXAMPLES

To provide a comprehensive overview of our benchmark, we present additional explanations and question examples from the seven categories in Table 8. These categories were intentionally designed as a hierarchical metric to probe different levels of a VLM's reasoning capabilities. They represent a deliberate scaffolding of different cognitive and semantic difficulty, spanning from **Direct Recognition** (e.g., "Basic POI", "Brand-Specific") and **Contextual Inference** (e.g., locating an un-signed "restroom"), to **Fine-Grained Attribute Reasoning** (e.g., an "accessible entrance") and culminating in highly **Subjective and Abstract Reasoning** (e.g., interpreting "I'm feeling hungry").

## C EXPERIMENTAL DETAILS

### C.1 IMPLEMENTATION DETAILS

We evaluate 27 commercial and open-source VLMs with multi-image input capabilities, selected based on three criteria: (1) recency and popularity, (2) coverage of the full parameter spectrum, from lightweight ($\leq$ 8B) to very large ($\geq$ 70B), (3) architectural diversity covering both Transformer-based models and mixture-of-experts (MoE) designs (e.g., LLaMA-4 series), and (4) varied training methodologies including reinforcement learning and reasoning-enhanced approaches (e.g., CoT-based o4-mini and Gemini-2.5-pro). All commercial models are accessed through their official APIs, while open-source implementations leverage the Hugging Face ecosystem. Inference uses vendors' default

Table 8: Navigation Instruction Categories and Examples.

| Category | Description | Example |
|---|---|---|
| 1. Basic POI Navigation | Request common urban facilities. | *"Please find the nearest restaurant."* 
 *"Please find the nearest convenience store."* 
 *"Please find the nearest shopping mall."* 
 *"Please find the nearest bank."* 
 *"Please find the nearest healthcare facility."* |
| 2. Brand-Specific Navigation | Seek specific commercial brand locations. | *"Please find the nearest Starbucks."* 
 *"Please find the nearest KFC or McDonald's."* 
 *"Please find the nearest 7-Eleven."* 
 *"Please find the nearest Chase Bank."* 
 *"Please find the nearest Apple."* |
| 3. Transportation Hub Navigation | Ask for public transit locations. | *"Please find the nearest subway station."* 
 *"Please find the nearest bus station."* |
| 4. Latent POI Navigation | Indirectly observable targets requiring contextual reasoning. | *"Please find the nearest restroom."* 
 *"Please find the nearest gym."* 
 *"Please find the nearest ATM."* 
 *"Please find the nearest parking lot."* 
 *"Please find the nearest cinema."* |
| 5. Abstract Demand Navigation | Express abstract human needs through contextual clues. | *"I'm feeling hungry and would like something to eat. Could you help me find a nearby place?"* 
 *"I'm feeling thirsty and would like something to drink. Could you help me find a nearby place?"* 
 *"I have run out of phone credit and need to recharge it. Could you assist me in finding a nearby place where I can do so?"* 
 *"I'm not feeling well and need assistance finding a suitable place nearby. Could you help me?"* 
 *"I want to bring my child to play and need assistance finding a suitable place nearby. Could you help me?"* 
 *"I need to go to the airport and would like assistance in finding the best way there. Could you help me?"* |
| 6. Inclusive Infrastructure Navigation | Prioritize inclusive infrastructure. | *"Please find the nearest restaurant with an accessible entrance."* 
 *"Please find the nearest clothing store with an accessible entrance."* 
 *"Please find the nearest apartment building with an accessible entrance."* 
 *"Please find the nearest office building with an accessible entrance."* 
 *"Please find the nearest bank with an accessible entrance."* |
| 7. Semantic Preference Navigation | Use descriptive language for subjective criteria. | *"Please find the nearest upscale restaurant."* 
 *"Please find the nearest restaurant with outdoor seating."* 
 *"Please find the nearest restaurant with roadside parking."* 
 *"Please find the nearest romantic restaurant."* 
 *"Please find the nearest group or family-friendly restaurant."* |

settings or greedy decoding, except for o4-mini, where we set temperature = 1.0. Experiments are executed on machines equipped with NVIDIA RTX 4090 and A100 GPUs.

Human evaluation involved 10 participants—5 undergraduates and 5 graduates—recruited to reflect varied cultural and gender backgrounds. They were compensated at $15/hour , in line with standard part-time research assistant rates at our institution. Participants were selected based on both availability and familiarity with navigation tasks; notably, three of them had previously lived in three or more of the cities featured in our benchmark. Our annotation platform is shown in Figure 8, and it was configured exactly like the model evaluation environment, with a maximum step limit of 35.

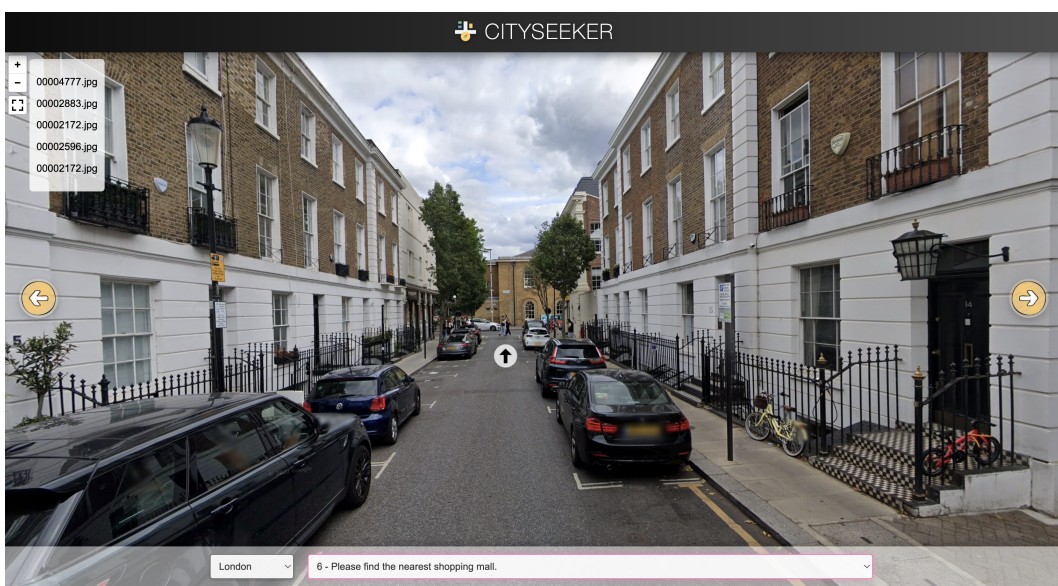

Figure 8: Developed Human Annotation Platform.

## C.2 EXPERIMENT PROMPTS

CitySeeker mainly employs two structured prompt types. Both prompts require a JSON output. Deterministic parsing is applied, and unparseable response automatically falls back to `action 0`. For extended experiments **BCR**, additional contextual signals `{backtrack_prompt}`, `{surrounding_prompt}`, and `{history_prompt}` are provided.

**Stop prompt**: The model inspects a panoramic view at the current node and outputs a single JSON block that fuses the *overall observation* with its *thoughts / rationale*. It then decides whether to continue (`action 0`) or terminate (`action -1`), and returns an accompanying confidence score.

**Choice prompt**: The model compares the perspective images generated for each navigable heading and selects the most promising direction. Two constraints apply: (1) `{perspective_prompt}` enumerates the available perspectives; (2) `{direction_prompt}` reveals the agent's current facing orientation, helping it avoid reflexively choosing a backward path.

---

**Stop Prompt**

#Instruction: You are a helpful robot that analyses images according to question and helps them find the way to reach their destination. Given the question, state whether the scene content satisfies the user's requirement.
#Output Format: {format_instructions}
#Example:
##Input:

```
[
    {"type": "text", "text": "I am hungry"},
    {"type": "image_url", "image_url": "..."}
]
```

```
##Output:

    {
        "overall observation": "There are residential buildings, a bookstore, and a bus station.",
        "thoughts": "I am hungry, so I should find a restaurant. No restaurant here, keep going.",
        "action": 0
    }

If you think you have already arrived at the destination, output action -1 for stop; otherwise 0.
#Now it's your turn.
##Input:
{query}
You are now at viewpoint {viewpoint_filename}.
{backtrack_prompt}
{image_content}
```

**Choice Prompt**

```
#Instruction: You are a helpful robot that analyses multiple candidate images and selects the one that best
answers the user's question. Return its index (A, B, C . . . ) together with a confidence score in [0, 1].
#Output Format: {format_instructions}
#Example:
##Input:

    [
        {"type": "text", "text": "I am hungry"},
        {"type": "image_url", "image_url": "..."},
        {"type": "image_url", "image_url": "..."}
    ]

##Output:

    {
        "perspective observation": {
            "A": "A narrow side street with no shops or amenities nearby.",
            "B": "A broad avenue lined with numerous office buildings, suggesting a higher chance of
                    restaurants and other services."
        },
        "thoughts": "I'm hungry and need the route most likely to lead to food. The broad avenue (B) lined
                with office buildings is much more likely to have restaurants, while the narrow side street (A)
                lacks any amenities. Therefore, I should head toward B."
        "action": "B",
        "score": 0.78
    }

{perspective_prompt}. Make sure the number of observations equals the number of perspectives provided.
{direction_prompt}. Prioritise the FORWARD, LEFT, and RIGHT directions; move BACK only if no better
option exists.
You have just backtracked, please choose image index {index}.
Historical context from previous rounds: {surrounding_prompt}.
Visit trajectory so far: {history_nodes_prompt}.
{query}
You are now at viewpoint {viewpoint_filename}.
{image_content}
```

## C.3 BENCHMARK EVALUATION RESULTS DETAILS

For completeness, we report the outcomes of each individual run along with all performance metrics
in Tables 9, 10, and 11, providing a more granular view of the models' behavior.

## C.4 CROSS-LINGUAL EXPERIMENT ON LINGUISTIC BIAS

To investigate whether the performance disparity between cities like New York and Beijing was due
to linguistic bias in our English-only prompts, we conducted a cross-lingual experiment. We selected

Table 9: The overall performance of CitySeeker Framework.

| Model | TCE | TCP-40m | TCP-50m | TCP-60m | TCC | SPD | nDTW | AS |
|---|---|---|---|---|---|---|---|---|
| GPT-4o | 2.39% | 14.96% | 18.30% | 23.07% | 6.84% | 125.4 | 136.97 | 21.17 |
| GPT-4o-mini | 1.11% | 9.94% | 12.33% | 15.43% | 7.56% | 201.97 | 325.17 | 31.62 |
| o4-mini | 2.63% | 14.80% | 17.90% | 21.80% | 6.76% | 130.09 | 156.33 | 22.96 |
| Gemini-1.5-pro | 1.91% | 13.13% | 15.43% | 17.82% | 7.48% | 157.14 | 241.86 | 31.73 |
| Gemini-2.5-pro | 1.83% | 13.84% | 17.34% | 20.92% | 5.01% | 121.75 | 121.15 | 18.43 |
| InternVL2.5-8B | 0.95% | 10.66% | 14.56% | 18.85% | 4.38% | 118.47 | 140.43 | 28.88 |
| InternVL2.5-26B | 1.59% | 11.06% | 15.27% | 19.09% | 3.66% | 109.53 | 106 | 23.4 |
| InternVL2.5-38B | 2.23% | 14.32% | 18.14% | 22.28% | 7.16% | 136.55 | 169.18 | 25.03 |
| InternVL3-8B | 1.27% | 11.85% | 15.83% | 20.29% | 4.85% | 118.27 | 144.51 | 30.13 |
| InternVL3-14B | 1.67% | 12.41% | 15.12% | 19.17% | 6.44% | 136.17 | 170.81 | 25.66 |
| InternVL3-38B | 2.47% | 15.04% | 19.25% | 23.95% | 6.68% | 115.82 | 128.34 | 25.62 |
| Qwen2-VL-7B | 0.72% | 7.96% | 11.14% | 15.27% | 1.67% | 111.4 | 114.36 | 24.79 |
| Qwen2-VL-72B | 0.95% | 9.07% | 11.93% | 16.79% | 2.31% | 113.01 | 89.52 | 13.13 |
| Qwen2.5-VL-7B | 0.48% | 11.30% | 15.83% | 21.16% | 4.30% | 119.02 | 151.84 | 32 |
| Qwen2.5-VL-32B | 2.55% | 17.66% | 21.08% | 24.90% | 6.21% | 122.6 | 146.95 | 24.55 |
| Qwen2.5-VL-72B | 1.99% | 11.14% | 14.56% | 17.58% | 7.24% | 174.92 | 250.22 | 28.63 |
| Llama3.2-90B | 0.88% | 9.15% | 12.49% | 15.19% | 3.74% | 124.51 | 123.05 | 18.99 |
| Llama-4-Scout-17B | 1.75% | 12.01% | 14.08% | 18.22% | 7.00% | 145 | 211.35 | 31.74 |
| Llama-4-Maverick-17B | 0.88% | 7.64% | 10.82% | 15.99% | 1.59% | 107.14 | 110.51 | 30.97 |
| Llama-3.2-11B-Vision | 0.48% | 9.15% | 12.49% | 16.79% | 2.39% | 112.93 | 112.44 | 26.07 |
| Llava-Llama3-8B | 0.32% | 6.92% | 10.42% | 14.32% | 0.80% | 104.79 | 86.85 | 20.78 |
| Llava-Qwen2-7B | 0.32% | 4.38% | 6.92% | 11.46% | 0.40% | 98.07 | 49.82 | 4.95 |
| MiniMax01 | 1.51% | 11.61% | 13.60% | 16.47% | 6.84% | 172.05 | 236.55 | 25.7 |
| MiniCPM-V-2.6 | 0.88% | 8.91% | 11.69% | 15.12% | 3.50% | 122.23 | 152.17 | 32.33 |
| MiniCPM-o-2.6 | 1.35% | 11.38% | 15.51% | 20.76% | 6.44% | 130.05 | 175.96 | 32.04 |
| Phi-3.5-Vision | 0.32% | 7.40% | 10.90% | 14.72% | 0.95% | 107.11 | 81.95 | 15.88 |
| Phi-4-Multimodal | 0.56% | 6.36% | 9.15% | 13.44% | 1.11% | 101.06 | 58.05 | 9.05 |
| Random Choice | 0.72% | 8.83% | 13.92% | 19.17% | 3.18% | 112.41 | 128.28 | 35 |
| Forward Direction | 0.24% | 5.01% | 7.24% | 13.21% | 0.40% | 100.83 | 99.29 | 35 |

Table 10: The subcategory performance of CitySeeker Framework-Part1.

| Model | Basic POI | | | | | Brand-Specific | | | | | Transit Hub | | | | |
|---|---|---|---|---|---|---|---|---|---|---|---|---|---|---|---|
| | TCE (%) | TCP-50m(%) | TCC (%) | SPD | nDTW | TCE (%) | TCP-50m(%) | TCC (%) | SPD | nDTW | TCE (%) | TCP-50m(%) | TCC (%) | SPD | nDTW |
| GPT-4o | 1.1 | 18.9 | 3.0 | 118.3 | 99.6 | 3.7 | 21.1 | 8.7 | 125.7 | 163.4 | 5.4 | 19.6 | 7.1 | 114.3 | 98.2 |
| GPT-4o-mini | 0.9 | 12.3 | 4.3 | 203.2 | 323.8 | 3.7 | 21.1 | 8.7 | 176.3 | 286.8 | 1.8 | 14.3 | 7.1 | 225.1 | 331.9 |
| o4-mini | 1.1 | 20.2 | 2.2 | 129.1 | 141.6 | 6.2 | 19.9 | 11.8 | 134.3 | 177.6 | 1.8 | 12.5 | 3.6 | 146.3 | 187.9 |
| Gemini-1.5-pro | 1.9 | 17.6 | 3.0 | 157.4 | 230.8 | 4.4 | 20.5 | 8.7 | 160.6 | 255.2 | 0.0 | 12.5 | 5.4 | 131.5 | 138.8 |
| Gemini-2.5-pro | 1.5 | 20.4 | 2.8 | 121.3 | 110.9 | 3.7 | 18.0 | 8.1 | 122.5 | 150.4 | 0.0 | 16.1 | 0.0 | 109.2 | 106.8 |
| InternVL2.5-8B | 0.9 | 15.7 | 3.4 | 121.7 | 133.9 | 1.2 | 14.3 | 3.1 | 105.5 | 103.7 | 1.8 | 1.8 | 1.8 | 119.3 | 111.4 |
| InternVL2.5-26B | 1.1 | 20.4 | 1.9 | 110.2 | 105.9 | 5.0 | 12.4 | 6.8 | 108.7 | 104.0 | 0.0 | 8.9 | 0.0 | 120.5 | 119.9 |
| InternVL2.5-38B | 1.5 | 17.6 | 2.8 | 145.3 | 170.8 | 7.5 | 24.2 | 10.6 | 115.7 | 136.8 | 0.0 | 14.3 | 1.8 | 126.3 | 145.2 |
| InternVL3-8B | 0.2 | 17.2 | 2.8 | 125.7 | 153.2 | 1.9 | 15.5 | 5.6 | 107.3 | 113.3 | 0.0 | 7.1 | 1.8 | 118.5 | 98.0 |
| InternVL3-14B | 0.9 | 13.8 | 2.8 | 147.1 | 178.5 | 4.4 | 20.5 | 8.1 | 112.1 | 128.4 | 1.8 | 17.9 | 3.6 | 136.4 | 151.6 |
| InternVL3-38B | 1.7 | 18.9 | 2.6 | 122.2 | 128.6 | 7.5 | 25.5 | 11.8 | 98.0 | 103.8 | 0.0 | 12.5 | 5.4 | 128.5 | 114.0 |
| Qwen2-VL-7B | 0.9 | 11.4 | 0.9 | 116.4 | 111.6 | 0.6 | 9.9 | 1.2 | 104.8 | 99.7 | 0.0 | 14.3 | 1.8 | 102.5 | 85.0 |
| Qwen2-VL-72B | 0.9 | 14.8 | 0.4 | 111.7 | 72.2 | 1.9 | 8.7 | 2.5 | 114.6 | 92.1 | 0.0 | 7.1 | 1.8 | 110.6 | 68.6 |
| Qwen2.5-VL-7B | 0.4 | 20.2 | 1.1 | 121.0 | 151.4 | 0.6 | 17.4 | 5.0 | 112.5 | 142.3 | 1.8 | 12.5 | 10.7 | 127.3 | 153.6 |
| Qwen2.5-VL-32B | 1.7 | 22.2 | 2.6 | 122.7 | 132.0 | 3.7 | 30.4 | 8.7 | 115.4 | 165.0 | 0.0 | 10.7 | 5.4 | 132.9 | 127.3 |
| Qwen2.5-VL-72B | 1.5 | 14.8 | 3.9 | 182.9 | 251.0 | 2.5 | 21.7 | 5.0 | 158.1 | 237.4 | 1.8 | 14.3 | 7.1 | 159.8 | 206.0 |
| Llama3.2-90B | 0.9 | 13.6 | 2.2 | 135.2 | 140.5 | 0.6 | 12.4 | 3.1 | 115.7 | 114.6 | 1.8 | 1.8 | 1.8 | 124.3 | 106.3 |
| Llama-4-Scout-17B | 1.7 | 12.3 | 4.5 | 153.8 | 221.2 | 3.1 | 18.6 | 8.7 | 136.8 | 196.1 | 3.6 | 14.3 | 5.4 | 158.1 | 213.3 |
| Llama-4-Maverick-17B | 1.1 | 12.7 | 0.9 | 113.3 | 116.5 | 1.2 | 10.6 | 2.5 | 103.0 | 111.1 | 1.8 | 8.9 | 1.8 | 99.8 | 84.0 |
| Llama-3.2-11B-Vision | 0.9 | 15.1 | 1.1 | 117.2 | 114.3 | 0.0 | 14.3 | 1.2 | 107.0 | 111.4 | 0.0 | 7.1 | 0.0 | 113.2 | 94.2 |
| Llava-Llama3-8B | 0.4 | 14.8 | 0.7 | 108.4 | 90.1 | 0.0 | 8.7 | 0.6 | 103.4 | 87.2 | 1.8 | 5.4 | 1.8 | 113.0 | 101.6 |
| Llava-Qwen2-7B | 0.4 | 12.5 | 0.2 | 102.2 | 51.7 | 0.0 | 2.5 | 0.0 | 95.5 | 49.3 | 0.0 | 3.6 | 0.0 | 92.9 | 46.9 |
| MiniMax-01 | 2.2 | 15.5 | 5.6 | 171.0 | 226.3 | 1.2 | 16.2 | 2.5 | 166.2 | 218.3 | 1.8 | 8.9 | 1.8 | 158.9 | 148.5 |
| MiniCPM-V-2.6 | 0.7 | 12.5 | 2.6 | 128.2 | 153.2 | 1.2 | 10.6 | 3.1 | 120.6 | 154.7 | 3.6 | 19.6 | 8.9 | 111.5 | 129.7 |
| MiniCPM-o-2.6 | 1.3 | 14.4 | 2.6 | 137.4 | 185.7 | 3.7 | 22.4 | 10.6 | 112.6 | 144.7 | 0.0 | 10.7 | 1.8 | 132.7 | 160.1 |
| Phi-3_5-Vision | 0.7 | 14.4 | 0.7 | 112.0 | 87.7 | 0.0 | 6.8 | 0.6 | 100.9 | 72.9 | 0.0 | 3.6 | 1.8 | 108.3 | 67.5 |
| Phi-4-Multimodal | 0.9 | 14.0 | 0.7 | 102.5 | 59.4 | 0.0 | 6.2 | 0.0 | 96.7 | 54.0 | 0.0 | 5.4 | 1.8 | 100.8 | 49.5 |
| Forward Direction | 0.7 | 13.3 | 0.2 | 105.0 | 102.3 | 0.0 | 3.1 | 0.0 | 97.8 | 103.0 | 0.0 | 3.6 | 0.0 | 99.2 | 93.8 |
| Random Choice | 0.7 | 16.6 | 1.7 | 116.9 | 129.4 | 0.6 | 10.6 | 4.4 | 110.6 | 130.1 | 1.8 | 5.4 | 3.6 | 126.0 | 138.0 |

Table 11: The subcategory performance of CitySeeker Framework-Part2.

| Model | Latent POI | | | | | Abstract Demand | | | | | Inclusive Infrastructure | | | | | Semantic Preference | | | | |
|---|---|---|---|---|---|---|---|---|---|---|---|---|---|---|---|---|---|---|---|---|
| | TCE (%) | TCP-50m(%) | TCC (%) | SPD | nDTW | TCE (%) | TCP-50m(%) | TCC (%) | SPD | nDTW | TCE (%) | TCP-50m(%) | TCC (%) | SPD | nDTW | TCE (%) | TCP-50m(%) | TCC (%) | SPD | nDTW |
| GPT-4o | 1.7 | 9.7 | 10.2 | 158.1 | 208.0 | 4.4 | 18.9 | 11.4 | 129.1 | 155.6 | 1.5 | 14.9 | 1.5 | 111.1 | 111.2 | 1.9 | 26.0 | 8.7 | 108.4 | 139.9 |
| GPT-4o-mini | 0.0 | 7.4 | 8.0 | 233.5 | 379.8 | 0.9 | 11.8 | 13.6 | 215.5 | 374.4 | 0.0 | 7.5 | 4.5 | 175.1 | 282.7 | 1.0 | 10.6 | 8.7 | 158.3 | 214.4 |
| o4-mini | 1.1 | 13.1 | 9.7 | 144.5 | 201.4 | 4.8 | 16.7 | 12.3 | 133.3 | 160.1 | 0.0 | 9.0 | 0.0 | 107.5 | 104.6 | 3.9 | 24.0 | 8.7 | 102.8 | 121.3 |
| Gemini-1.5-pro | 0.0 | 11.4 | 11.4 | 183.6 | 291.7 | 3.1 | 14.5 | 13.6 | 155.4 | 248.7 | 0.0 | 6.0 | 6.0 | 154.5 | 259.7 | 1.0 | 14.4 | 7.7 | 125.3 | 215.8 |
| Gemini-2.5-pro | 0.6 | 10.2 | 3.4 | 139.2 | 140.7 | 3.5 | 20.6 | 10.5 | 118.7 | 129.6 | 0.0 | 9.0 | 0.0 | 111.6 | 81.8 | 1.0 | 13.5 | 6.7 | 113.1 | 103.2 |
| InternVL2.5-8B | 1.7 | 14.8 | 4.6 | 125.8 | 167.8 | 0.9 | 18.4 | 8.8 | 113.9 | 146.6 | 0.0 | 10.5 | 1.5 | 115.8 | 156.7 | 0.0 | 10.6 | 3.9 | 122.8 | 171.9 |
| InternVL2.5-26B | 0.6 | 10.8 | 3.4 | 115.5 | 117.6 | 2.6 | 17.1 | 7.9 | 107.7 | 102.4 | 0.0 | 7.5 | 1.5 | 91.7 | 81.6 | 0.0 | 8.7 | 1.0 | 107.2 | 105.9 |
| InternVL2.5-38B | 0.6 | 13.1 | 6.8 | 157.7 | 218.4 | 3.5 | 22.8 | 14.0 | 132.1 | 169.6 | 0.0 | 10.5 | 6.0 | 115.3 | 127.6 | 0.0 | 16.4 | 10.6 | 122.8 | 167.6 |
| InternVL3-8B | 2.8 | 14.2 | 7.4 | 125.8 | 167.7 | 1.8 | 16.7 | 7.5 | 110.9 | 145.8 | 1.5 | 16.4 | 3.0 | 115.4 | 138.1 | 1.9 | 15.4 | 5.8 | 107.6 | 141.4 |
| InternVL3-14B | 1.7 | 8.0 | 6.3 | 149.3 | 195.1 | 1.8 | 19.3 | 10.5 | 134.3 | 184.0 | 1.5 | 10.5 | 6.0 | 110.0 | 99.4 | 1.0 | 17.3 | 13.5 | 122.9 | 188.4 |
| InternVL3-38B | 0.6 | 13.1 | 5.1 | 129.6 | 163.7 | 3.1 | 23.3 | 14.0 | 117.1 | 134.4 | 1.5 | 11.9 | 4.5 | 100.6 | 99.8 | 1.9 | 21.2 | 5.8 | 91.9 | 118.3 |
| Qwen2-VL-7B | 0.0 | 4.6 | 1.1 | 116.4 | 113.0 | 1.8 | 13.2 | 4.4 | 104.4 | 108.7 | 0.0 | 13.4 | 0.0 | 111.0 | 150.1 | 0.0 | 15.4 | 1.9 | 111.3 | 157.0 |
| Qwen2-VL-72B | 0.6 | 7.4 | 2.8 | 111.8 | 81.1 | 1.8 | 14.5 | 5.3 | 114.4 | 120.5 | 0.0 | 1.5 | 1.5 | 112.0 | 78.9 | 0.0 | 15.4 | 3.9 | 117.6 | 127.4 |
| Qwen2.5-VL-7B | 0.6 | 10.2 | 5.7 | 129.4 | 163.7 | 0.4 | 13.6 | 5.2 | 117.2 | 153.9 | 0.0 | 7.5 | 4.5 | 117.4 | 156.6 | 0.0 | 15.4 | 4.8 | 103.1 | 139.9 |
| Qwen2.5-VL-32B | 1.7 | 14.2 | 6.3 | 146.3 | 197.9 | 5.3 | 24.1 | 10.5 | 116.3 | 155.9 | 1.5 | 9.0 | 7.5 | 115.6 | 125.1 | 1.9 | 20.2 | 8.7 | 106.0 | 104.7 |
| Qwen2.5-VL-72B | 0.0 | 6.8 | 6.8 | 217.0 | 319.7 | 2.6 | 15.8 | 11.0 | 160.7 | 231.2 | 9.0 | 11.9 | 16.4 | 139.3 | 185.8 | 1.0 | 14.4 | 12.5 | 156.7 | 256.3 |
| Llama3.2-90B | 0.6 | 9.7 | 3.4 | 125.4 | 117.4 | 0.9 | 14.0 | 6.1 | 116.4 | 109.5 | 0.0 | 10.5 | 3.0 | 104.9 | 83.7 | 1.9 | 16.4 | 8.7 | 119.2 | 131.4 |
| Llama-4-Scout-17B | 0.6 | 8.5 | 6.8 | 160.8 | 239.0 | 2.2 | 17.5 | 11.8 | 132.8 | 201.8 | 0.0 | 17.9 | 6.0 | 119.9 | 173.0 | 1.0 | 14.4 | 6.7 | 127.5 | 188.7 |
| Llama-4-Maverick-17B | 1.1 | 10.2 | 2.8 | 105.5 | 104.7 | 0.4 | 12.7 | 2.2 | 107.0 | 116.3 | 0.0 | 4.5 | 0.0 | 93.3 | 84.2 | 0.0 | 4.8 | 1.0 | 101.7 | 111.2 |
| Llama-3.2-11B-Vision | 0.0 | 9.7 | 2.8 | 107.9 | 92.1 | 0.4 | 12.3 | 5.3 | 114.7 | 125.3 | 1.5 | 9.0 | 4.5 | 108.1 | 126.6 | 0.0 | 8.7 | 2.9 | 110.5 | 112.8 |
| Llava-Llama3-8B | 0.0 | 8.0 | 1.1 | 102.1 | 80.4 | 0.4 | 8.8 | 1.3 | 103.5 | 83.7 | 0.0 | 4.5 | 0.0 | 100.6 | 88.5 | 0.0 | 7.7 | 0.0 | 96.8 | 80.7 |
| Llava-Qwen2-7B | 0.0 | 2.8 | 0.6 | 99.5 | 51.4 | 0.9 | 6.6 | 1.3 | 94.6 | 47.4 | 0.0 | 3.0 | 0.0 | 90.1 | 45.5 | 0.0 | 1.0 | 0.0 | 96.7 | 49.1 |
| MiniMax-01 | 0.0 | 6.8 | 7.4 | 213.6 | 320.6 | 1.8 | 14.9 | 12.7 | 172.2 | 273.8 | 1.5 | 11.9 | 4.5 | 119.0 | 123.7 | 1.0 | 13.5 | 9.6 | 156.8 | 206.9 |
| MiniCPM-V-2.6 | 0.6 | 9.7 | 2.8 | 132.5 | 176.2 | 1.3 | 13.2 | 4.4 | 112.6 | 141.2 | 0.0 | 7.5 | 3.0 | 114.8 | 145.6 | 0.0 | 8.7 | 4.8 | 112.4 | 143.6 |
| MiniCPM-o-2.6 | 1.1 | 12.5 | 8.5 | 143.7 | 190.0 | 1.3 | 18.9 | 10.5 | 123.5 | 175.6 | 0.0 | 11.9 | 4.5 | 118.3 | 169.3 | 0.0 | 12.5 | 8.7 | 121.7 | 170.7 |
| Phi-3_5-Vision | 0.6 | 9.1 | 1.1 | 107.8 | 83.0 | 0.0 | 11.8 | 1.3 | 103.4 | 76.8 | 0.0 | 9.0 | 3.0 | 110.2 | 106.6 | 0.0 | 7.7 | 0.0 | 99.4 | 71.7 |
| Phi-4-Multimodal | 0.0 | 2.8 | 0.6 | 104.1 | 55.4 | 1.3 | 10.5 | 4.0 | 101.9 | 64.0 | 0.0 | 7.5 | 0.0 | 92.8 | 49.9 | 0.0 | 2.9 | 0.0 | 99.6 | 59.7 |
| Forward Direction | 0.0 | 2.3 | 0.6 | 99.4 | 93.2 | 0.0 | 6.6 | 1.3 | 101.0 | 104.1 | 0.0 | 1.5 | 0.0 | 93.8 | 95.0 | 0.0 | 1.9 | 0.0 | 94.7 | 85.6 |
| Random Choice | 0.0 | 10.8 | 1.7 | 114.9 | 131.3 | 1.3 | 14.9 | 6.6 | 108.9 | 126.3 | 0.0 | 16.4 | 3.0 | 96.3 | 113.9 | 1.0 | 13.5 | 2.9 | 101.7 | 123.7 |

representative models and ran them on tasks in both Beijing and New York, comparing performance when using our standard English prompts versus fully localized Chinese prompts and outputs.

Table 12: Cross-lingual performance in Beijing and New York. Localizing prompts to Chinese did not yield consistent improvements, suggesting linguistic bias is not the primary performance driver.

| Model | City | Language | TCP (%) | TCE (%) | TCC (%) | nDTW |
|---|---|---|---|---|---|---|
| Qwen2.5-VL-32B | Beijing | English | 25.95 | 4.32 | 10.81 | 122.40 |
| | | Chinese | 23.91 | 3.44 | 8.88 | 120.36 |
| | New York | English | 15.42 | 3.48 | 3.48 | 144.78 |
| | | Chinese | 14.93 | 2.99 | 4.98 | 133.13 |
| InternVL3-8B | Beijing | English | 14.63 | 0.00 | 3.25 | 143.41 |
| | | Chinese | 15.45 | 1.63 | 1.63 | 148.51 |
| | New York | English | 15.42 | 2.49 | 6.97 | 136.13 |
| | | Chinese | 9.45 | 1.49 | 2.99 | 140.07 |
| GPT-4o-mini | Beijing | English | 7.32 | 0.81 | 3.25 | 345.36 |
| | | Chinese | 7.32 | 2.44 | 2.44 | 217.31 |
| | New York | English | 13.43 | 1.00 | 9.45 | 332.95 |
| | | Chinese | 14.93 | 2.49 | 8.96 | 251.84 |

The results, detailed in Table 12, show that localizing the prompts to Chinese does not provide a consistent performance benefit, and the impact varies significantly by model and city. For instance, the top-performing Qwen2.5-VL-32B saw its TCP *decrease* in both cities when using Chinese. Conversely, GPT-4o-mini's performance *increased* in New York with Chinese prompts, while InternVL3-8B's performance dropped sharply. This high degree of variability strongly suggests that the observed performance gaps between cities are not primarily driven by linguistic factors but are likely rooted in deeper visual and geographic biases within the models' training data.

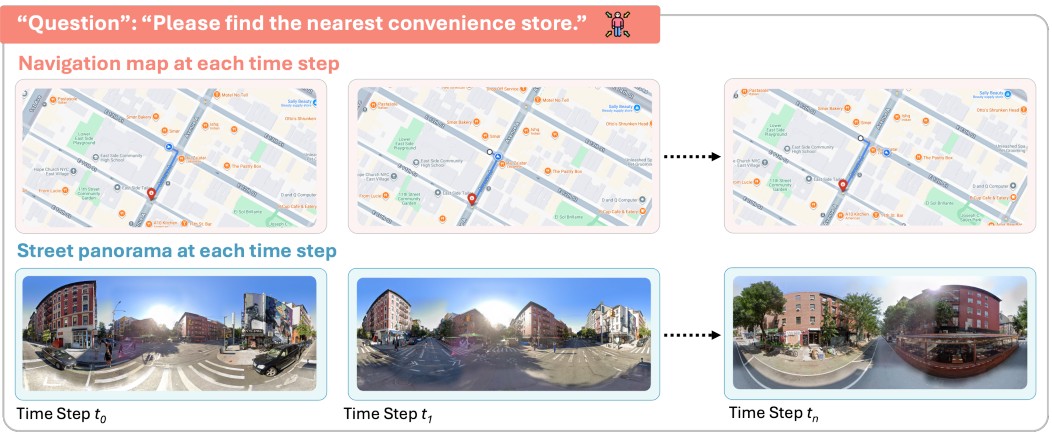

Figure 9: Illustration of the agent's input in the map-augmented setting. At each time step, the VLM receives both a global navigation map, which updates with its current position, and the corresponding first-person street panorama.

## D    ABLATION STUDY: THE IMPACT OF GLOBAL MAP INFORMATION

A primary design choice in **CitySeeker** is the focus on a VLM's ability to navigate based on its first-person visual perception and intrinsic world knowledge. To better isolate and understand these intrinsic reasoning abilities, we conducted an ablation study to quantify the impact of providing explicit global map information. This study explores an alternative, map-augmented setting where, at every step, the VLM was provided with both its first-person street view and an interactive 2D global map. This map dynamically updated with the agent's current position and heading, mimicking the experience of a human using a modern navigation application.

### D.1    EXPERIMENTAL SETUP

In this setting, the agent's prompt was augmented at each step with both a dynamically rendered 2D map and the corresponding first-person street panorama, as illustrated in Figure 9. This interactive map always showed the full planned route while centering on the agent's current position and heading, providing continuous global context just as a user would see in navigation software. The VLM was explicitly instructed to first analyze this map to determine the geometrically optimal next step and then select the corresponding street-view perspective. This setup was designed to test if providing such real-time, human-like map guidance would simplify the task and improve performance.

### D.2    RESULTS AND ANALYSIS

We evaluated two top-performing models, GPT-4o and Qwen2.5-VL-32B-Instruct, on the full benchmark test set under this map-augmented condition. The results were counter-intuitive and revealing, as summarized in Table 13.

Table 13: Performance comparison between the map-free and map-augmented navigation settings on the full test set. The new results reveal a trade-off where map guidance improves path following (nDTW) but degrades task completion (TCP/TCE).

| Model | Map-Free | | | Map-Augmented | | |
|---|---|---|---|---|---|---|
| | TCE (%) | TCP (%) | nDTW | TCE (%) | TCP (%) | nDTW |
| GPT-4o | 2.4 | 18.3 | 136.9 | 0.9 | 11.7 | 75.7 |
| Qwen2.5-VL-32B-Instruct | 2.6 | 21.1 | 147.0 | 0.2 | 7.6 | 54.4 |

Contrary to expectations, providing a global map at each step led to a surprising trade-off: while path-following efficiency improved dramatically (as shown by lower nDTW scores), the models'

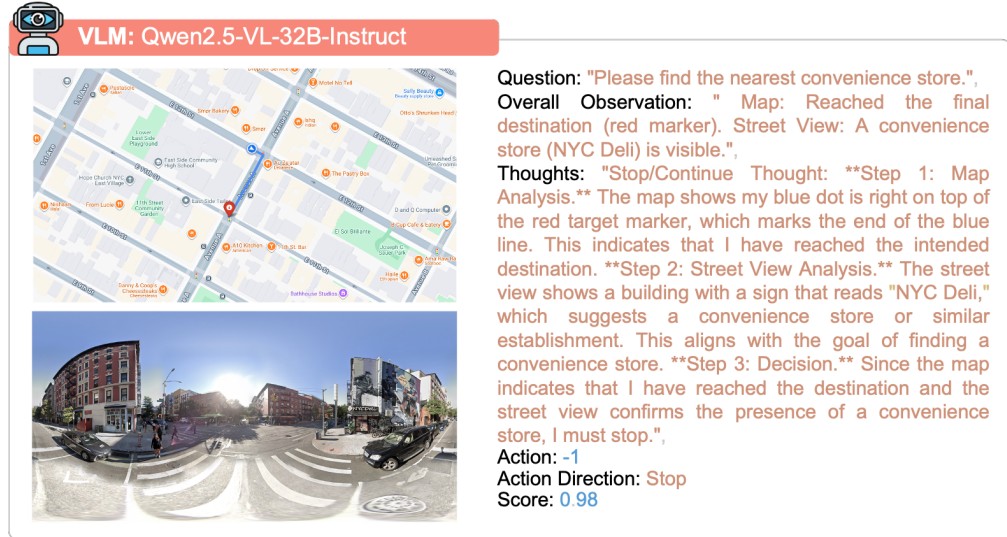

Figure 10: An example of a typical reasoning failure from Qwen2.5-VL-32B-Instruct.

ability to complete the actual task collapsed. For both models, Task Completion (both TCP and TCE) dropped significantly. For instance, the TCP for Qwen2.5-VL-32B-Instruct fell from 21.1% to just 7.6%, even as its path alignment score (nDTW) improved from 147.0 to 54.4. This suggests that while the map helps the agent follow a geometric route, it severely hinders its ability to perform the core task of semantic discovery. Our analysis of the models' reasoning traces, an example of which is shown in Figure 10, revealed primary failure modes:

(1) **Poor Cognition of 2D Map Geometry:** The models demonstrated a weak understanding of direction and distance on the 2D map. They frequently failed to perform correct "mental rotations"; for instance, when their heading was North but the route required moving South, they struggled to select the "back" perspective. Furthermore, they showed little awareness of distance, often hallucinating that they were near the destination when still at the starting point.

(2) **Over-reliance and Failure to Align:** The VLMs tended to fixate on the 2D map's abstract geometric instructions, failing to properly align them with the labeled, first-person street views. For instance, after inferring a "left turn" from the map, a model would still frequently select the "front" perspective, despite being explicitly told which perspective corresponded to the "left" direction. This demonstrates a deep failure in grounding an abstract command, even when provided with all necessary information.

(3) **Trivialization of the Core Challenge:** The map effectively turned the task from one of *discovery and semantic reasoning* into a simpler, but more brittle, geometric path-following exercise. The core challenge of our benchmark—inferring that "I'm thirsty" means looking for a café and then visually identifying one—was often ignored. The models would become "map followers", focusing only on the blue line and failing to perform the crucial visual exploration needed to identify the actual POI that satisfied the user's implicit need.

### D.3 CONCLUSION

This ablation study demonstrates that simply providing a global map does not necessarily solve the core challenges of implicit-need navigation and can even distract the VLM from the essential task of grounding abstract language in the visual world. Furthermore, the map's limited information and potential for positional deviation can introduce additional sources of error. Therefore, to isolate and rigorously evaluate the agent's **intrinsic spatial cognition and commonsense reasoning abilities**, our primary experimental results are reported under the map-free setting. This approach forces the VLM to build and rely on its own internal mental model of the environment, providing a truer measure of its embodied intelligence. While developing agents that can effectively fuse map and

visual information is a valuable direction for future research, our work focuses on first establishing this crucial baseline for the VLM's intrinsic, perception-driven capabilities.

## E BCR STUDIES DETAILS

### E.1 BACKTRACKING MECHANISMS

We propose three distinct backtracking strategies to mitigate error accumulation over long trajectories in large-scale VLN tasks in urban environments: **(B1)** Basic Backtracking, **(B2)** Step-Reward Backtracking, and **(B3)** Human-Guided Backtracking.

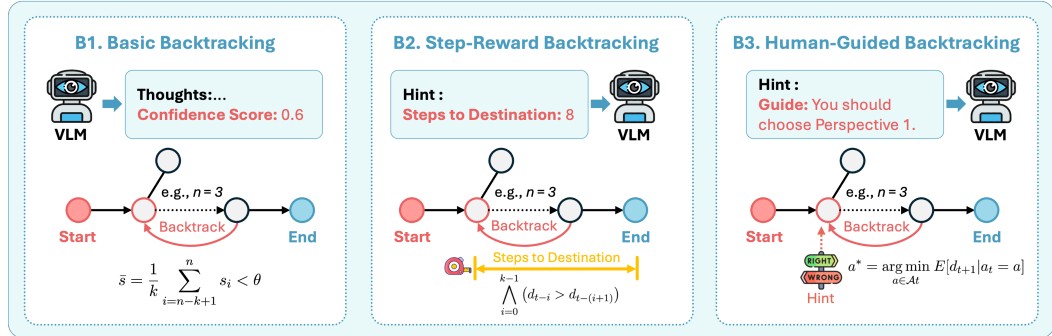

Figure 11: Backtracking Mechanisms: (B1) Basic Backtracking, (B2) Step-Reward Backtracking, and (B3) Human-Guided Backtracking.

**Basic Backtracking (B1).** In this basic backtracking strategy, the agent reverts to the last "trusted" node when its internal confidence falls below a predefined threshold over several consecutive steps. The model tracks its confidence scores during each step, and if the average confidence score over a set number of steps drops below a threshold, the agent will backtrack to the last trusted node. This simple strategy operates independently of any external signals. The confidence metric, stored in a deque, is updated at each step with the current score of the model's prediction.

The implementation process maintains a sliding window of prediction confidence scores $s_t{}_{t=n-k}^{n}$ over $k$ consecutive steps (default $k = 3$). Backtracking triggers when:

$$\bar{s} = \frac{1}{k} \sum_{i=n-k+1}^{n} s_i < \theta \quad (\theta = 0.75) \tag{1}$$

The agent reverts to the last node where the confidence $s_t$ meets or exceeds the threshold $\theta$. The agent reverts $k$ steps when the average confidence $\bar{s}$ falls below $\theta$.

**Step-Reward Backtracking (B2).** This mechanism evaluates progress toward the goal by replacing subjective confidence scores with objective topological distance as the backtracking criterion. Specifically, it calculates the shortest path steps $d_t$ from current viewpoint $v_t$ to target $v_{\text{target}}$ in the graph topology $\mathcal{G}$.

During each step, the agent checks whether the distance to the goal has increased. Backtracking triggers when distances increase monotonically over $k = 3$ consecutive steps:

$$\bigwedge_{i=0}^{k-1} \left( d_{t-i} > d_{t-(i+1)} \right) \tag{2}$$

This condition ensures that if the agent's progress toward the goal stagnates or worsens, it will backtrack.

**Human-Guided Backtracking (B3).**  This strategy extends basic backtracking B1 with corrective guidance, providing minimal external "hint" that suggests the best action to take next after backtracking. The optimal action $a^*$ maximizes path consistency with the shortest path to $v_{target}$ through:

$$a^* = \arg\min_{a \in \mathcal{A}t} \mathbb{E}[d_{t+1}|a_t = a] \tag{3}$$

where $\mathcal{A}t$ denotes available actions at backtracked node $v_t$, and $d_{t+1}$ represents the expected graph-theoretic distance after taking action $a$.

The guidance integrates by dynamically aligning the shortest path directions with navigable headings through topological analysis. The action that maximizes path consistency is selected via:

$$\phi(a) = \mathbb{I}\theta_a \in \Theta_{optimal} \cdot \cos(\theta_a - \theta_{path}) \tag{4}$$

where $\theta_a$ is the heading direction for action $a$, and $\Theta_{optimal}$ is the set of optimal path headings. This external guidance helps ensure that the agent's action maximizes path consistency and progress toward the goal.

### E.2    SPATIAL COGNITION ENRICHMENT

In this section, we explore the potential of providing richer spatial cues to an agent to enhance global awareness and reduce fragmented decision-making during navigation tasks. Specifically, we compare two distinct methods of presenting spatial information: Topology Cognitive Graph (C1) and Relative Position Maps (C2). The goal is to investigate how these different representations influence the agent's ability to understand its environment and effectively plan routes.

We investigate whether supplying richer spatial cues can bolster global awareness and reduce fragmented decision-making by synthesizing multi-model trajectory data through GPT-4.1 as a cognitive summarizer. Our methodology processes ground truth paths and generated trajectories from GPT-4o and Gemini2.5-pro (containing both correct segments and historical errors) for each navigation task. GPT-4.1 analyzes these heterogeneous trajectories using specialized prompts to produce structured spatial representations, which are then integrated into the initial input for downstream VLMs.

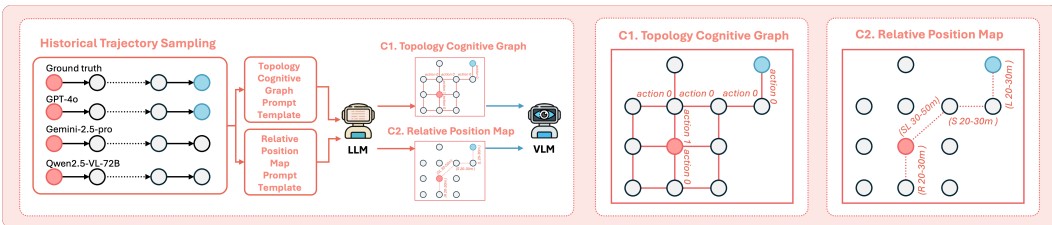

Figure 12: Spatial Cognition Enrichment: (C1) Topology Cognitive Graph, and (C2) Relative Position Maps.

**(C1) Topology Cognitive Graph.**  In this approach, the VLM is provided with a topological graph of recently traversed segments, which explicitly defines the connectivity between various locations. The graph consists of nodes representing locations and directed edges that signify possible actions or transitions between those locations. Each trajectory is annotated with explicit relationships between nodes, forming a clear and structured map of the environment. This graph can be seen as an abstracted representation of the agent's route, where the agent is instructed to focus on the connectivity between locations. This forces VLM to ground decisions in connectivity patterns, reducing exploration of invalid paths.

An example of how this spatial information is represented is shown in the prompt:

---

**Topology Cognitive Graph Generation Prompt**

Given the following trajectories with the same starting point for the same question, connect them and represent the relationships between the nodes in a graph format.
Example:
### Node: 00002929.jpg
**Relationships:**
- '00002929.jpg' → **action 0** → '00002930.jpg'
- '00002929.jpg' → **action 1** → '00002931.jpg'
### Node: 00003210.jpg
**Relationships:**
- '00003210.jpg' → **action 0** → '00003211.jpg'
- '00003210.jpg' → **action 1** → '00003212.jpg'
### Node: 00001111.jpg
**Relationships:**
- '00001111.jpg' → **action 0** → '00001112.jpg'.

---

**(C2) Relative Position Map.** In contrast to the topology-based approach, the Relative Position Map emphasizes the spatial orientation of locations without directly specifying connectivity. Instead of relying on an explicit graph structure, the VLM receives approximate directional cues that describe the relative positions of recently visited nodes. These cues include terms such as "front", "back", "left", "right", and "slightly left/right", along with approximate distances between nodes. This approach provides the VLM with a more intuitive understanding of the space, allowing it to navigate with a flexible mental model of its surroundings, albeit with less precision in terms of connectivity.

An example of the relative directional information provided to the VLM is:

---

**Relative Position Map Generation Prompt**

Given the following trajectories with the same starting point for the same question, describe the spatial relationships between the key nodes. Emphasize the relative directional orientation (such as front, back, left, right, slightly left, slightly right, etc.) and relative distances. Note that, not all nodes are directly connected, so focus on the relative spatial relationships rather than direct connections.
Example:
### 00001190.jpg → 00001192.jpg
- **Direction**: Right
- **Relative Position**: Distance: 20-30 meters
### 00001192.jpg → 00001204.jpg
- **Direction**: Slightly right
- **Relative Position**: Distance: 30-50 meters
### 00001204.jpg → 00001539.jpg
- **Direction**: Slightly left
- **Relative Position**: Distance: 50-70 meters.

---

This method captures the VLM's internal spatial sense, where it perceives locations in terms of relative proximity and directional alignment rather than as distinct, connected points. The benefit of this approach is that it fosters a more adaptable navigation strategy, particularly in dynamic or complex environments where exact node connectivity is not always available or necessary.

### E.3 MEMORY-BASED RETRIEVAL

To address the issue of "fragmented memory", we introduce a memory-based retrieval mechanism based on the Neo4j graph database. This mechanism consists of three core components: (R1) Topology-based Retrieval, (R2) Spatial-based Retrieval, and (R3) Historical Trajectory Lookup. This architecture enables memory formation across multiple reasoning iterations while mitigating error propagation.

**Topology-based Retrieval (R1).** Each navigation query undergoes $n$ sequential rounds of execution. During rounds 1 to $n-1$, the agent explores the environment without memory retrieval, with all trajectories and node metadata (e.g., observations, thoughts, decisions, and confidence scores) stored in Neo4j. In the final round $n$, the agent activates memory retrieval mode, dynamically accessing

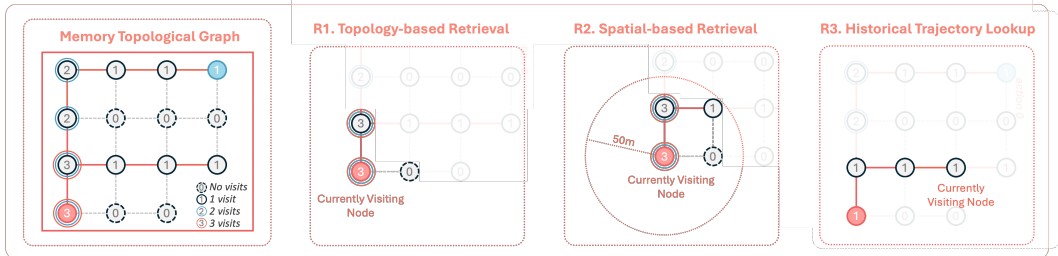

Figure 13: Memory-Based Retrieval: (R1) Topology-based Retrieval, (R2) Spatial-based Retrieval, and (R3) Historical Trajectory Lookup.

historical cross-round information. At each navigation step, the agent queries a local subgraph spanning $h$-hop topological connections (default $h$=1) through Neo4j's graph traversal operations. The returned subgraph contains:

Nodes with cumulative visitation counts and $n-1$ round metadata (thoughts, previous/next actions, directional choices, confidence scores); Edges annotated with historical transition patterns across multiple rounds (successful/failed attempts, directions). For example, a node might reveal that previous rounds chose "action 1" (right turn) when transitioning to a specific neighbor. This topological memory enables the agent to prioritize frequently successful paths while avoiding recurrent error-prone branches.

**Spatial-based Retrieval (R2).** Complementing topological constraints, this mechanism retrieves localized subgraphs based on Euclidean spatial proximity. At each navigation step in the final round $n$, the agent queries all nodes and relationships within a configurable radius (default: 50 meters) from the current position using Neo4j's geospatial index. This dual focus on geographic density and historical usage patterns allows the agent to weigh physically closer options while avoiding previously failed paths. By jointly modeling metric space and graph topology, R2 mitigates the "tunnel vision" of pure topological approaches, particularly in irregular urban layouts where physical proximity often outweighs structural connectivity.

**Historical Trajectory Lookup (R3).** This mechanism introduces short-term working memory by dynamically appending the agent's recent navigation history within the current round. At each time step $t$, the mechanism automatically appends the trajectory segment from the preceding $n$ steps ($n = 3$ by default), including: (1) spatial context - filenames, coordinates, and headings of visited nodes; (2) action traces - movement directions (e.g., "turn left", "proceed straight") and associated confidence scores; (3) decision rationale - preserved observations and reasoning from prior steps.

Unlike cross-round memory in R1/R2, R3 operates through a sliding temporal window that exclusively tracks intra-episode navigation patterns. At step $t$, the $t-1$ to $t-n$ entries get injected into the VLM's input context through template-based natural language formatting, preserving recent decision logic. This short-term memory gets purged upon round completion through automated attribute pruning in Neo4j, ensuring no residual traces affect subsequent trials.

### E.4 PERFORMANCE OF COMBINED BCR STRATEGIES

To explore the synergistic potential of our proposed strategies, we conducted a preliminary experiment where we combined several effective mechanisms (B2, B3, C1, R3). The results, shown in Table 14, indicate a positive but not strictly additive effect. For instance, the combined strategies boosted Qwen2.5-VL-32B's TCP from 19.9% to 27.38%. This suggests complex interactions between the different cognitive tools. A full exploration of optimal strategy combinations is a promising direction for future work.

## F VLM GENERATION EXAMPLES

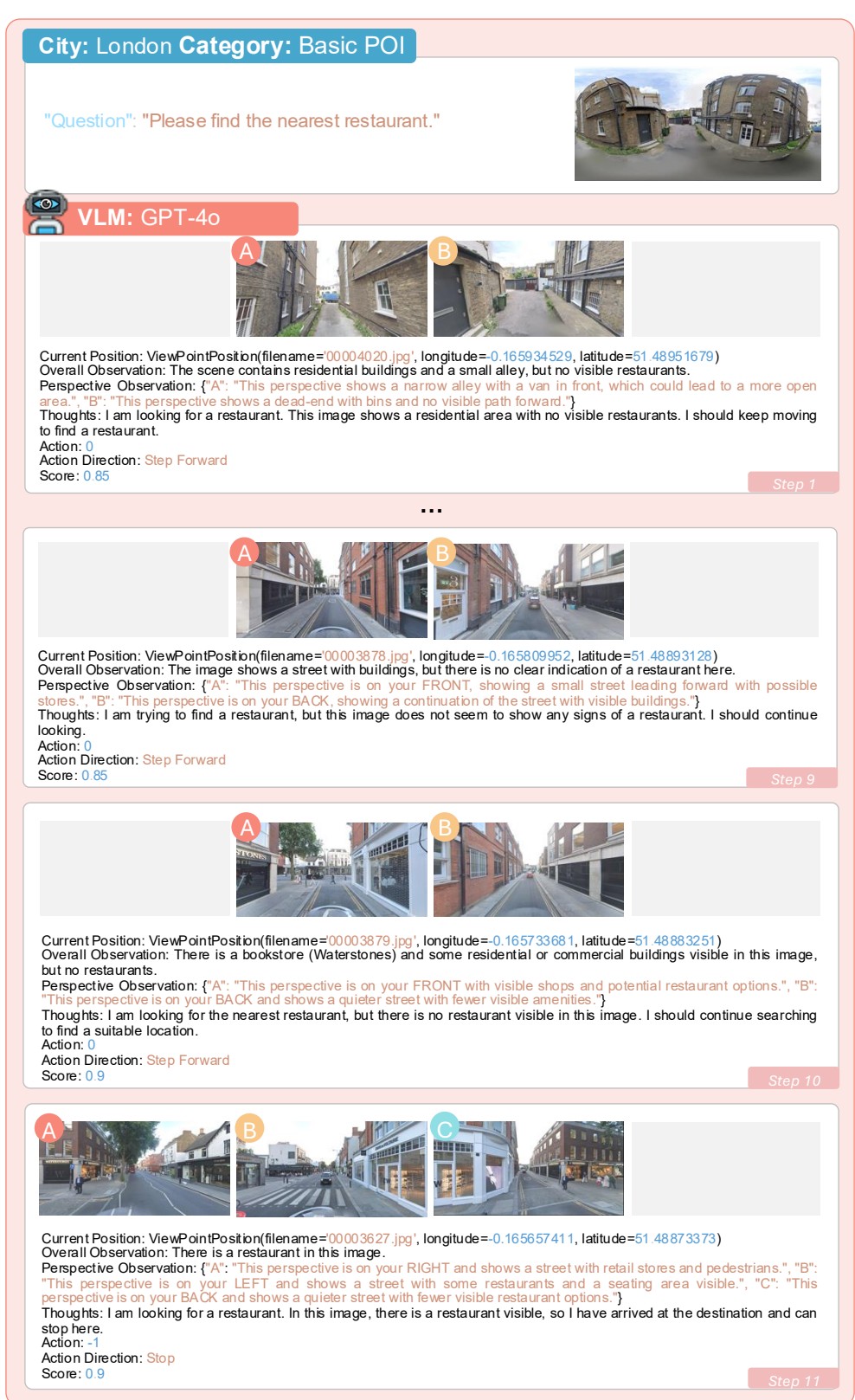

Figure 14: Further illustrative examples drawn from the CitySeeker benchmark.

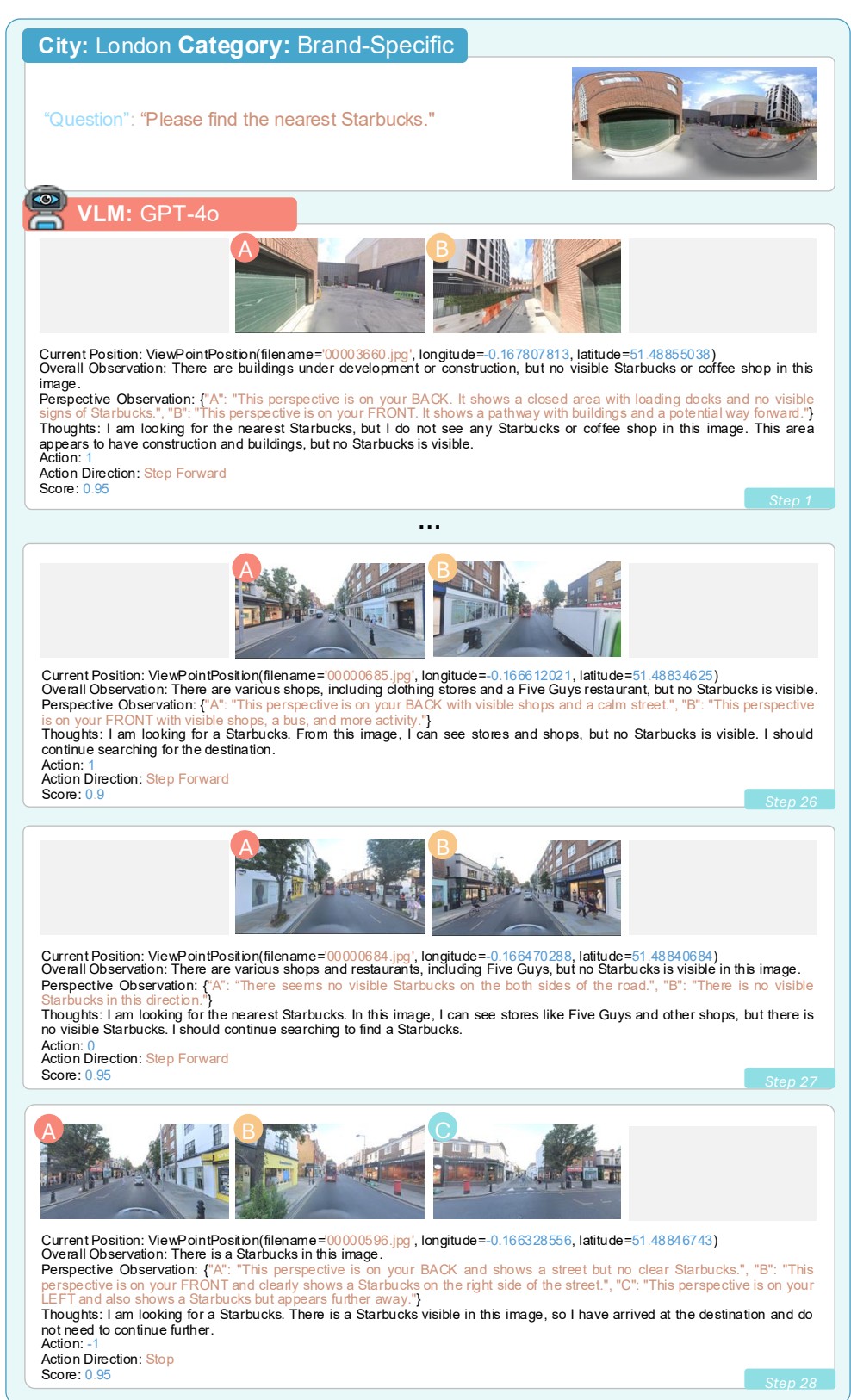

Figure 15: Further illustrative examples drawn from the CitySeeker benchmark.

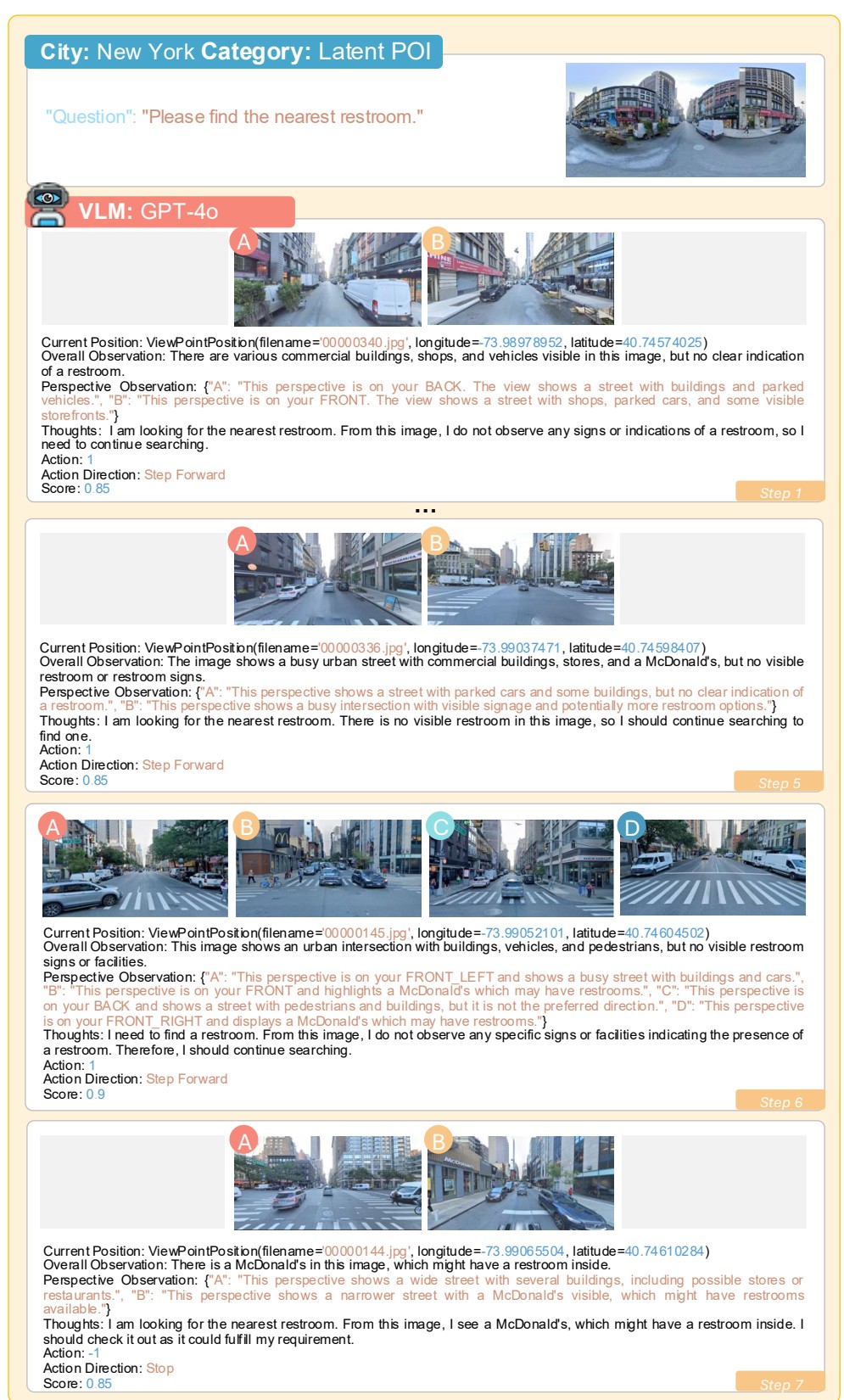

Figure 16: Further illustrative examples drawn from the CitySeeker benchmark.

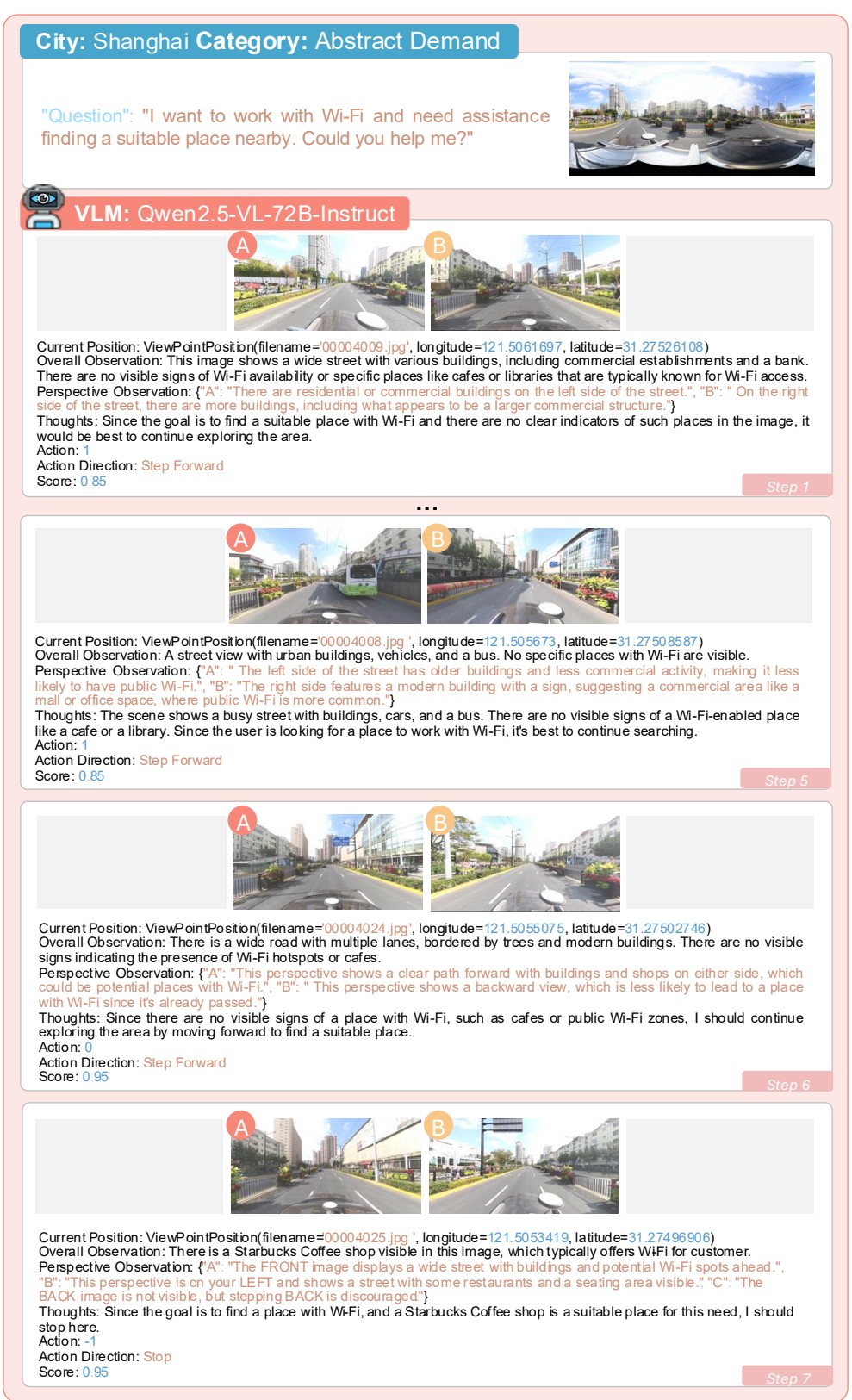

Figure 17: Further illustrative examples drawn from the CitySeeker benchmark.

Table 14: Performance with combined BCR strategies.

| Model | Baseline TCP % | Combined BCR TCP % |
|---|---|---|
| InternVL3-8B | 15.0 | 18.96 |
| Qwen2.5-VL-32B | 19.9 | 27.38 |

# G ERROR ANALYSIS

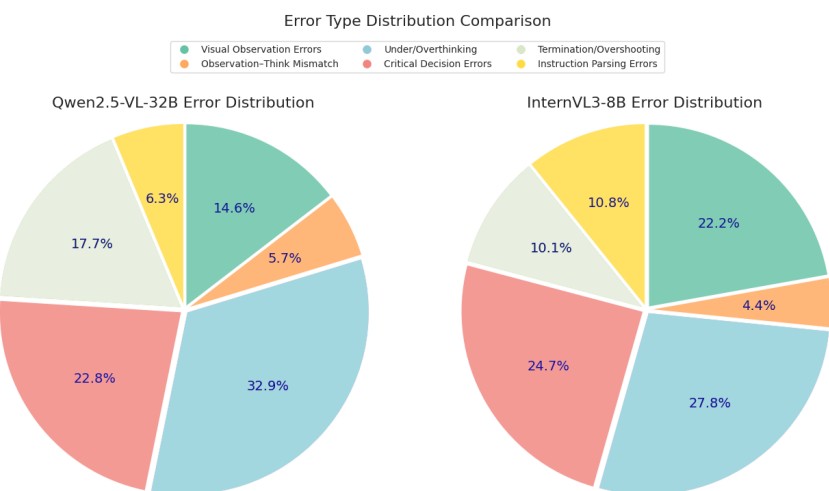

Figure 18: Error Distribution of Primary Error Types in Qwen2.5-VL-32B and InternVL3-8B for Embodied Urban Navigation.

To identify the main bottlenecks in VLMs for Embodied Urban Navigation with implicit human needs, we thoroughly examine and analyze the primary error patterns observed in the 300-sample mini-size subset of the CitySeeker benchmark. By manually inspecting model outputs — including observations, rationales and actions, with corresponding street-view images, we categorize failure modes into six distinct types (visually exemplified in Figure 19):

(1) **Visual Observation Errors:** These arise from failure to recognize target or misidentification of critical visual cues (e.g., overlooking signage while fixating on irrelevant street objects).

(2) **Observation–Think Mismatch:** Even when the VLM correctly observes the target, the rationale diverges from the observation. For example, the model visually identifies a target restaurant but fails to prioritize advancing toward it in its rationale.

(3) **Underthinking or Overthinking:** Underthinking is particularly evident when the model lacks knowledge of non-primary functional affordances of a POI. For example, while recognizing Starbucks, the model ignores its potential to provide Wi-Fi, leading to a missed target. Overthinking might draw unfounded conclusions from limited evidence, leading to incorrect conclusions, such as assuming a store labeled 'CVS Pharmacy' is a convenience store.

(4) **Critical Decision Errors at Waypoints:** Incorrect routing choices at pivotal junctions, leading to irreversible deviations. This occurs when the target is observed within the right field of view, yet the model erroneously selects the leftward path, or when perceptual inaccuracies lead to the choice of a non-viable route that cannot reach the target.

(5) **Premature Termination or Overshooting:** Over-anticipatory stopping before reaching targets or passing by the target leads to not reaching or missing the target.

(6) **Instruction Parsing Errors:** Generation of malformed JSON outputs violating predefined action schemas (e.g., missing "action" fields), rendering trajectories unexecutable.

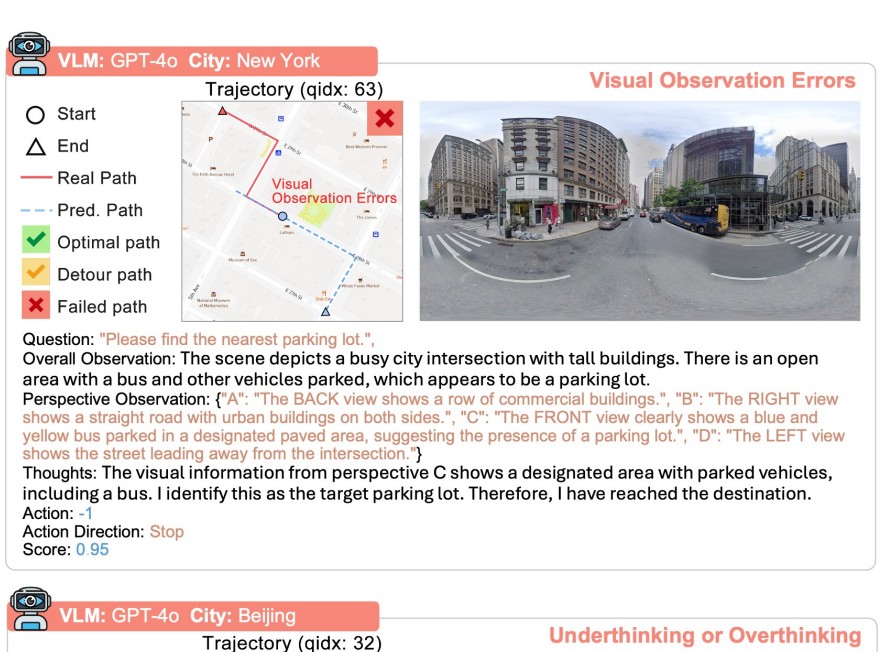

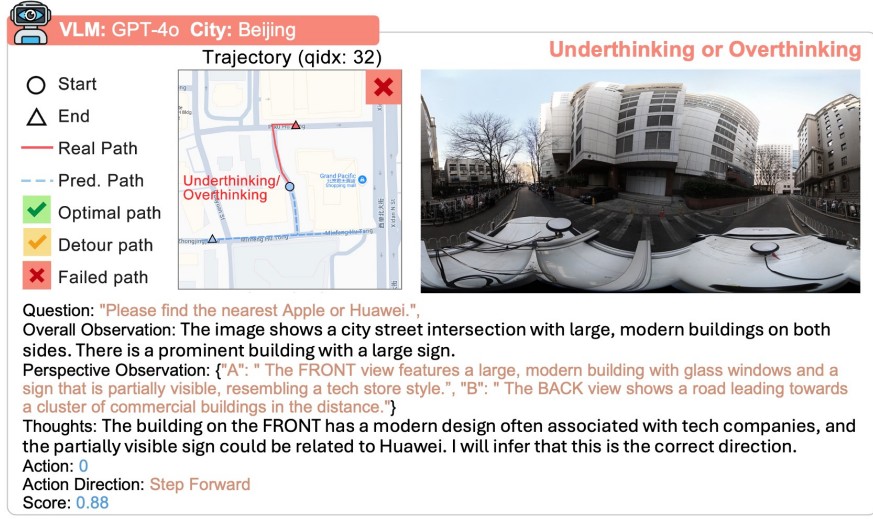

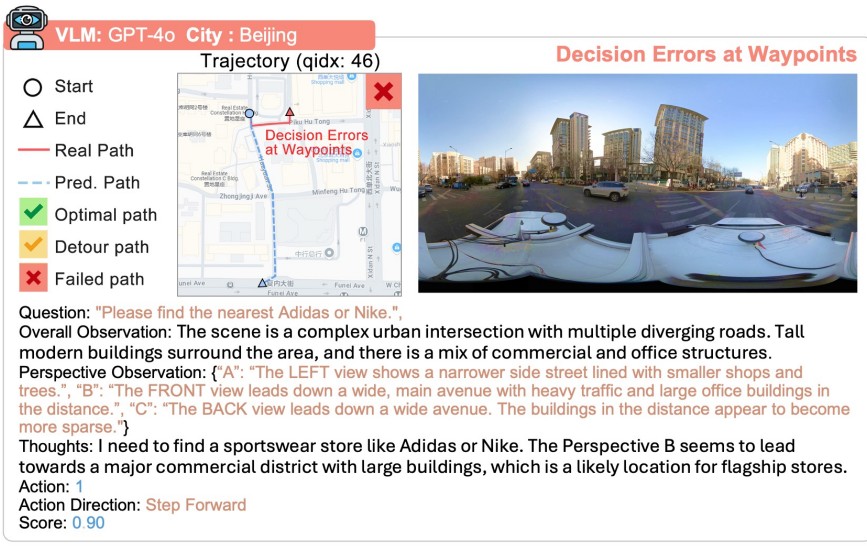

Figure 19: Qualitative examples of primary VLM failure modes in CitySeeker.

Figure 18 compares two representative VLMs - Qwen2.5-VL-32B and InternVL3-8B - revealing distinct error profiles. Both exhibit significant **critical decision errors** (Type 4: 22.8% vs. 24.7%), highlighting persistent challenges in spatial reasoning at navigation waypoints. The larger Qwen model demonstrates stronger visual perception (Type 1: 14.6% vs. 22.2%) but suffers more **under-thinking/overthinking errors** (Type 3: 32.9% vs. 27.8%). This discrepancy likely stems from the model's lack of human-like urban living experience, which limits its ability to discern primary/secondary affordances of POIs. Additionally, the model occasionally draws **unfounded conclusions** from sparse visual evidence - potentially due to overcompensation for missing contextual knowledge. Conversely, InternVL3-8B struggles with **instruction parsing** (Type 6: 10.8% vs. 6.3%) and basic visual recognition, with more misidentifications of street POIs and hallucination. This contrast reveals a scale-driven tradeoff: larger models develop nuanced visual grounding but introduce complex reasoning failures, while smaller models face fundamental perception and instruction-compliance challenges.

**Deeper Dive: Human vs. VLM Failure Modes** Through analysis of error trajectories and participant interviews, a deeper analysis of failure modes reveals a crucial distinction between human and VLM bottlenecks. As summarized in Table 15, VLM failures are predominantly **cognitive**, stemming from a lack of commonsense knowledge. In contrast, the dominant failure mode for humans is **strategic**. Faced with unfamiliar environments and a strict step limit, humans possess the necessary world knowledge to understand the task, but often falter in devising an optimal exploratory plan. This manifests as inefficient, intuition-driven exploration—leading to forgotten paths and repeated loops—or poor management of the step budget, such as overshooting a valid target.

Table 15: Detailed comparison of primary failure modes between humans and the top-performing VLM (Qwen2.5-VL-32B).

| Failure Mode Category | Human Error Profile & Rationale | Model Error Profile & Rationale (Qwen2.5-VL-32B) |
|---|---|---|
| **Strategic & Navigational** (e.g., Termination/Overshooting) | **60.7% (Primary Human Failure)** Humans understand the goal but struggle with devising an optimal exploratory plan. Failures are intuition-driven and stem from: **1. Inefficient Exploration:** Participants noted that poor signage or visually similar streets often led to confusion and forgetting previously explored paths, causing inefficient loops that exhaust the 35-step limit. **2. Overshooting:** A human might find a valid target but continue walking past it "just to see what's around the next corner," a reasonable exploratory behavior that fails under a strict step budget. | **40.5%** The model's errors here are less strategic. It does not suffer from imperfect memory but can still misjudge distance to a visible target or fail to recognize it, leading to premature stops or passing the goal. |
| **Cognitive Failures** (e.g., Under/Over-thinking) | **19.1%** Humans rarely fail basic inferences. Errors occur on nuanced tasks, such as failing to consider a convenience store's secondary function of recharging phone credit, or overthinking whether a restaurant is truly "upscale". | **32.9%** The model's key bottleneck. It struggles to infer non-obvious functions of POIs and lacks the real-world experience to make flexible logical leaps, revealing a critical gap in commonsense reasoning. |
| **Visual & Execution Errors** (e.g., Visual, Parsing, Mismatch) | **20.2%** In an unfamiliar city and under pressure, human perception can falter. This is exacerbated by language barriers (e.g., non-native participants processing English storefront signs) and cultural unfamiliarity with local brands, leading to overlooked or misidentified cues. | **26.6%** The model is prone to a mix of errors: misidentifying visual cues (14.6%), failing to adhere to the output format (6.3%), or having a disconnect between its visual observation and textual rationale (5.7%). |

This contrast reveals a fascinating trade-off. An AI agent has a theoretical advantage with its perfect memory, which should prevent the inefficient looping that humans are prone to. However, this is counteracted by its lack of deep, intuitive commonsense. While humans possess the necessary real-world knowledge to understand the implicit needs, they are hampered by imperfect memory and an intuition-driven approach to planning that can be inefficient in unfamiliar environments. This suggests that future work should focus not just on improving VLM perception, but on endowing them with more robust, human-like intuitive and strategic reasoning capabilities.

## H  LIMITATIONS AND FUTURE WORK

Despite the advancements enabled by CitySeeker, our study highlights several limitations of current VLMs in embodied urban navigation, which in turn point to promising directions for future research.

First, **generalization and adaptability to specific contexts** remain challenges. Physically, while models perform modestly in familiar urban layouts, their effectiveness declines in irregular road networks due to training data biases. Semantically, our framework currently relies on foundational spatial consensus and does not yet account for personalized user preferences or historical behavioral patterns, limiting the model's ability to provide tailored assistance.

Second, **long-context reasoning and memory retention** are key bottlenecks. Long-horizon navigation requires processing extensive sequences of visual and textual information, which consumes a large number of tokens. This poses a challenge for models with context length limitations and can lead to the forgetting of crucial information from earlier steps, resulting in inefficient paths with loops and repeated errors.

Third, our ablation study on map-augmented navigation (Appendix D) revealed a fundamental weakness in **aligning 2D maps with first-person street views**. The models' poor cognition of orientation and distance on the 2D map often led to confusion and hallucinations, degrading performance rather than improving it. This highlights a critical gap in the models' ability to fuse these two distinct modes of spatial information.

Lastly, **real-time adaptability is limited by computational inefficiencies**. The significant latency in VLM decision-making, stemming from redundant visual processing, currently restricts practical deployment in real-time applications.

Addressing these limitations requires a multi-faceted approach. Future work should focus on:

- **Enhancing Commonsense and Affordance Reasoning:** A key area for improvement is the model's understanding of functional affordances (e.g., that a café provides Wi-Fi), a weakness particularly evident in smaller models. New training methodologies could focus on enriching models with this kind of real-world, commonsense knowledge.
- **Integrating Personalized Behavioral Priors:** While CitySeeker focuses on universal common sense, real-world navigation is often personalized. A promising future direction is to incorporate behavioral priors derived from GPS traces or check-in data. Once a VLM demonstrates robust spatial common sense, these empirical signals can be used to rank valid candidates, moving from general capability to personalized assistance.
- **Improving Geometric and Orientational Understanding:** Future research should explicitly target the model's ability to understand orientation and perform "mental rotations", bridging the gap between abstract 2D map representations and the first-person visual world.
- **Developing More Efficient Architectures:** Innovations in persistent memory architectures and more efficient inference frameworks are needed to tackle the challenges of long-horizon reasoning and real-time performance.

## I  LLM USAGE STATEMENT

In preparing this manuscript, LLM was used as a general-purpose assistive tool. Its application was confined to the final stages of writing, specifically for minor polishing and refinement of the English prose to improve clarity and readability. The LLM did not contribute to the research ideation, experimental methodology, data analysis, or the generation of results and conclusions in this work.

