# OpenReview forum: "CitySeeker: How Do VLMs Explore Embodied Urban Navigation with Implicit Human Needs?"
_ICLR.cc/2026/Conference — ICLR 2026 Poster_

### Official Review · Reviewer_4hXJ · 2025-10-20

**Soundness:** 3
**Presentation:** 4
**Contribution:** 2
**Rating:** 4
**Confidence:** 4

**Summary:**

This paper introduces *CitySeeker*, a novel benchmark for evaluating Vision-Language Models (VLMs) in embodied urban navigation driven by implicit human needs. Unlike prior VLN benchmarks that rely on explicit instructions, *CitySeeker* focuses on abstract, functional, and semantic goals (e.g., “I’m thirsty”), spanning 6,440 trajectories across 8 cities. The authors propose a framework for evaluating spatial reasoning and decision-making and identify key bottlenecks in current VLMs. They further introduce three human-inspired strategies—Backtracking, Spatial Cognition Enrichment, and Memory-Based Retrieval (BCR)—to enhance navigation performance.

**Strengths:**

- **Benchmark Contribution**: *CitySeeker* fills a critical gap in VLN research by targeting implicit human needs in dynamic, real-world urban environments. It is the first large-scale benchmark to do so across multiple cities and diverse task categories.
- **Realism and Diversity**:  The benchmark includes diverse urban layouts and visual characteristics, making it highly relevant for embodied AI applications.
-  Comprehensive Evaluation：
- **Comprehensive Evaluation**: The paper presents extensive empirical results across 27 VLMs, including proprietary and open-source models. The analysis spans task categories, cities, and trajectory patterns. The authors also  proposed BCR strategies are well-motivated and show measurable improvements in task completion and path efficiency.
- **Clarity and Presentation**:  The paper is generally well-written, with a clear logical flow and a well-motivated problem statement.

**Weaknesses:**

1.  **Problem Setup**:

    - The task formulation leans more toward object navigation than traditional VLN. In partially observable environments, defining the agent’s state solely as the current observation $o_t$ is insufficient.
    - A more appropriate formulation would involve belief state estimation using historical observations $\{o_0, o_1,...,o_t\}$, which is a regular practice in POMDP-based problem.

2. **Symbol and Notation Clarity**:

    - The definition of $v$ as a graph node and $v_t$ as the location is inconsistent.
    - Subscripts such as $o_i$ (different views at a single node) and $o_t$ (timesteps) are used ambiguously.
    - Lacks a clear explanation of how reasoning $\phi$ and action $a$ are computed -- what inputs are used ($O_t$ or $S_t$ or something else?).

4. **Evaluation Metrics Misalignment**:

    - Metrics like **nTCE**, which measure trajectory deviation from ground truth, are more suitable for instruction-following tasks (in previous VLN tasks, the agents are asked to follow the excact GT trajectory given a detailed user instruction).
    - In implicit-need-driven navigation, multiple valid trajectories may exist. Metrics like SPL (Success weighted by Path Length) or goal proximity might be more appropriate.


5. **Low Task Completion Rates**:

    -  Even the best-performing models and human baselines achieve low success rates under strict metrics (e.g., 5.7% TCE for humans), raising concerns about benchmark difficulty or metric suitability.

**Questions:**

**Questions**: See weakness.

---

> ### Author Response · Authors · 2025-11-20
> **Response to Reviewer 4hXJ (Part 1): Thanks for your detailed review and recognition.**
>
> We sincerely thank reviewer for the detailed and technical review. We are grateful that you recognized the strengths of our work, including the "critical gap" that CitySeeker fills, its "realism and diversity," "comprehensive evaluation," "clear logical flow" and "well-motivated problem statement."
>
> We address your concerns below, which primarily stem from a need to clarify our experimental design choices (establishing a memory-less baseline vs. exploring memory-based strategies) and the specific roles of our evaluation metrics. We hope these clarifications demonstrate that our methodology is robust and well-aligned with the problem.
>
> ### **1. Problem Setup (POMDP and Memory)**
>
> >**Concern:** *The agent state $s_t$ is defined "solely as the current observation," which is insufficient for a POMDP. A belief state based on historical observations should be used.*
>
> We thank the reviewer for this insightful comment. We fully agree that history is crucial for solving POMDPs.
>
> * **Rationale for Baseline Design:** Our baseline framework in Section 4.1 was intentionally designed to be memory-less for a specific scientific purpose: to isolate and evaluate the VLM's *intrinsic, zero-shot spatial reasoning ability* at each step, without the confounding variable of a complex memory module. As stated in Lines 244-245, we aim to establish a baseline of "what a VLM can do with pure visual reasoning" before introducing memory.
> * **Validation via BCR Strategies:** Your insight—that history is vital—is precisely the hypothesis we test and validate in Section 6 Exploratory Approaches. Our R-Series (Memory-Based Retrieval) strategies are explicitly designed to introduce history into the agent's state, directly addressing POMDP nature of task:
>     * **R1 (Topology-based Retrieval):** Retrieves historical node metadata and transition patterns based on graph connectivity.
>     * **R2 (Spatial-based Retrieval):** Retrieves context based on spatial proximity.
>     * **R3 (Historical Trajectory Lookup):** Directly implements a history-based agent by appending past trajectory data to the context, mimicking a belief state.
> **Our experiments confirm your point.** As shown in Table 3, introducing memory strategies significantly boosts performance. Specifically, for **Qwen2.5-VL-32B**, the **R3 strategy alone increases TCP from 19.9% to 25.4%**, and **R1 boosts it to 26.9%**. This confirms that while the memory-less baseline sets the floor, history-based reasoning is indeed essential for solving the POMDP.
>
> ### **2. Symbol and Notation Clarity**
>
> >**Concern:** *Ambiguity in symbols ($v$ vs. $v_t$, $n$ vs. $t$) and unclear inputs for $\Phi_t$ and $a_t$.*
>
> We appreciate you identifying this imprecision. We will revise Section 4.1 to be mathematically rigorous:
>
> * **Clarification of (a):** $v \in V$ refers to any node in the global graph set, while $v_t$ is the specific node occupied by the agent at timestep $t$. We use $t$ strictly for timesteps and $n$ for the number of discrete perspective views available at a node (e.g., **a panorama sliced into $n$ views**).
> * **Clarification of (b):** We will explicitly define the inputs for the reasoning ($\Phi_t$) and action ($a_t$) functions. At timestep $t$, the VLM-based policy $\pi$ takes the **instruction $W$** and the **current observation set $O_t = \{o_1, o_2, ..., o_n\}$** as inputs.
>     * Formally: $(a_t, \Phi_t) = \pi(W, O_t)$.
>     * Here, $\Phi_t$ is the internal "Thought" process (Line 241), and $a_t$ is the final "Action" decision (selecting a perspective index from $O_t$).
>
> ### **3. Evaluation Metrics alignment**
>
> >**Concern:** *Metrics like nDTW measure trajectory deviation (suitable for instruction following). Implicit navigation should use metrics like "goal proximity" or SPL (Success weighted by Path Length).*
>
> We are grateful for the constructive suggestion. We agree that evaluating implicit-need navigation requires a nuanced distinction between strict path following and efficient goal finding. Your feedback drove us to expand our evaluation.
>
> **3.1 Systematic and Flexible Evaluation Framework for Goal Proximity**
> We designed our metric system to be adaptable to the open-ended nature of the task, capturing different dimensions of success beyond rigid path adherence:
>
> * TCP (Task Completion - Proximity): This directly addresses your suggestion for "goal proximity." It considers a task successful if the agent stops within 50m of the target, acknowledging that large urban landmarks (e.g., malls) are visible from multiple nearby nodes. It is a realistic, flexible measure of success.
> * TCC (Task Completion - Category): This accounts for the semantic flexibility of implicit needs. If a user is "thirsty," stopping at any valid provider (e.g., a Bubble Tea shop) is a success, even if it wasn't the specific Ground Truth Convenience Store.
> * SPD (Shortest-Path Distance): We report SPD to quantify the straight-line distance to the goal, providing a pure proximity baseline.

---

> ### Author Response · Authors · 2025-11-20
> **Response to Reviewer 4hXJ (Part 2): Thanks for your detailed review and recognition.**
>
> ### **3. Evaluation Metrics alignment**
>
> **3.2 Integrating SPL following your suggestion**
>
> Driven by your feedback, we expanded evaluation to calculate SPL.
>
> **Table: SPL Leaderboard** (will be included in final manuscript)
>
> | Model | SPL (%) |
> | :--- | :--- |
> | **GPT-4o** | **13.27** |
> | GPT-4o-mini | 7.52 |
> | o4-mini | 11.61 |
> | Gemini-2.5-pro | 12.07 |
> | InternVL2.5-8B | 5.55 |
> | InternVL2.5-26B | 7.25 |
> | InternVL2.5-38B | 11.21 |
> | InternVL3-8B | 6.43 |
> | InternVL3-14B | 8.51 |
> | InternVL3-38B | 10.59 |
> | Qwen2-VL-7B | 5.24 |
> | Qwen2-VL-72B | 9.04 |
> | Qwen2.5-VL-7B | 4.57 |
> | Qwen2.5-VL-32B | 12.69 |
> | Qwen2.5-VL-72B | 9.12 |
> | Llama3.2-90B | 7.31 |
> | Llama-4-Scout-17B | 6.71 |
> | Llama-4-Maverick-17B | 3.96 |
> | MiniMax-01 | 8.82 |
> | MiniCPM-V-2.6 | 3.95 |
> | MiniCPM-o-2.6 | 5.56 |
> | Phi-4-Multimodal | 7.05 |
> | Llava-Llama3-8B | 5.06 |
> | Llava-Qwen2-7B | 6.21 |
> | **Human** | **21.23** |
> | Random Choice | 3.79 |
> | Forward Direction | 1.83 |
>
> * The results confirm that **Human (21.23%)** achieve the highest efficiency, validating that the task is solvable but requires high-level spatial reasoning to minimize detours. **GPT-4o (13.27%)** and the top open-source model **Qwen2.5-VL-32B (12.69%)** lead the pack, demonstrating the most effective path planning capabilities.
> * The metric effectively penalizes inefficient exploration and reveals hidden failures. While **InternVL2.5-38B** balances success with direct paths (11.21%), smaller models struggle. For instance, despite a decent TCP (15.8%), **Qwen2.5-VL-7B** drops sharply to an SPL of 4.57%, revealing that even when it succeeds, it suffers from severe Oscillatory Detours.
>
> ### **4. Low Task Completion Rates**
>
> >Concern: *Low success rates (e.g., 5.7% TCE for human) raise concerns about benchmark difficulty or metric suitability.*
>
> We appreciate your point regarding task difficulty. As noted in recent spatial intelligence research [1] (Yang et al., 2025), complex zero-shot navigation in open-world environments inherently challenges both biological and artificial agents. This complexity highlights the necessity of our systematic and multi-granular evaluation framework.
>
> * **The TCE vs. TCP Distinction:** As defined in Sec 4.2, TCE (Task Completion - Exact) requires a "strict, single-node endpoint match," which means stopping on exactly one pre-defined coordinate in a large city. This is difficult, especially since a target is often visible from multiple viewpoints. We included it for technical completeness.
> * **More Realistic Success Metric:** The more realistic metric for task success, as used in our main analysis (Sec 5.1), is TCP (Task Completion - Proximity). TCP threshold is a deliberate design choice. Our environment nodes are sampled every ~20 meters, meaning a single target is often clearly visible from 2-3 consecutive nodes (a 40-60m range). The 50m TCP radius correctly accounts for this spatial ambiguity and reflects a realistic visual range for identifying urban targets, whereas TCE metric is overly punitive. It simultaneously highlights the necessity of our comprehensive and flexible metrics and reveals important insights into human performance.
>
> **Human Superiority:** Under the reasonable metric, humans (**30.1% TCP**) significantly outperform the top-performing VLM, Qwen2.5-VL-32B (21.1% TCP). This confirms human superiority and credibility of our baseline. Furthermore, human advantage is robust. This lead is more pronounced against smaller models (e.g., more than **3x** that of Llava-Llama3-8B at 10.4% TCP). Humans also exhibit a clear advantage across diverse tasks, e.g., in Transit Hub navigation (**34.9% TCP** vs. the best VLM's 19.6%), common-sense inference in Abstract Demand tasks (**31.5%** vs. 24.1%), and goal-finding in Basic POI (**31.8%** vs. 22.2%).
>
> **Experimental Rigor:** The experimental design required strict consistency between human and model settings. Human participants operated under the same stringent constraints as models: a possible unfamiliar city environment, and notably, often facing different linguistic cues (e.g., an Asian participant navigating London, relying on visual recognition of non-native signs). Navigating an unknown city with 35-step limit and no map, relying purely on visual cues, is genuinely difficult. This setup ensures consistency with the model and rigorously tests the fundamental spatial cognition capability derived from general life experience.
>
> In summary, the human baseline is credible, performs as expected, and correctly highlights the challenge of the task. We will clarify this distinction between TCE and TCP in the main paper to avoid future confusion.
>
> We hope the clarification, along with **addition of SPL metric as per your valuable suggestion**, demonstrate the robustness of our setup and address your concerns.
>
> [1] Yang, J., Yang, S., Gupta, A. W., Han, R., Fei-Fei, L., & Xie, S. (2025). Thinking in space: How multimodal large language models see, remember, and recall spaces. In CVPR.

---

> ### Author Response · Authors · 2025-11-26
> **Response to Reviewer 4hXJ: We hope our rebuttal address your concerns and look forward to further discussion**
>
> We thanks again for your valuable time and the constructive feedback that significantly strengthen our paper.
>
> During rebuttal, we have focused on addressing your key concerns regarding problem setup, notation clarity, and evaluation metrics. Specifically, we have:
>
> 1.  **Clarified the POMDP formulation:** We explained that our **Exploratory Strategies (Section 6)**—specifically the Memory-Based Retrieval (R-Series)—explicitly incorporate historical context to address the partial observability inherent in the task.
> 2.  **Refined the formal definitions** of agent states and inputs in Section 4.1 to ensure mathematical rigor and remove ambiguity.
> 3.  **Integrated the SPL metric as per your constructive suggestion** across models and the human baseline. This new data (Human SPL: **21.23%** vs. SOTA VLM: **~13%**) provides a crucial, holistic view of navigation efficiency and further validates the benchmark's difficulty.
>
> **We deeply value the opportunity to engage in this discussion**. We would be grateful to know if our responses and the new experimental results have satisfactorily addressed your concerns.
>
> We hope these efforts demonstrate the robustness of our work and might warrant an updating of your evaluation. If there are any remaining questions or if further clarification is needed, we are eager to address them during the final days of the discussion phase.
>
> Best regards,
>
> The Authors

---

> > ### Comment · Reviewer_4hXJ · 2025-11-28
> >
> > I thank the authors for their comprehensive and thoughtful response. The rebuttal is exemplary and has directly addressed my initial technical concerns regarding the POMDP formulation, notation clarity, and the initial choice of metrics. The inclusion of the SPL analysis, following my suggestion, is a particularly valuable addition that significantly strengthens the evaluation section and provides deeper insights into the models' navigation efficiency.
> >
> > Based on the clarifications and the new results, I have updated my assessment. The rigour demonstrated in the rebuttal alleviates my initial technical concerns about the paper's execution.
> >
> > My final recommendation is a weak accept.

---

> ### Author Response · Authors · 2025-11-28
> **Thank you for your prompt response and recognition**
>
> We sincerely thank you for your prompt response and for upgrading your recommendation to a **6 Weak Accept**.
>
> We are glad to hear that our response, particularly the inclusion of the SPL metric and the POMDP clarification, has satisfactorily addressed your concerns. Your comments throughout the review process have been instrumental in refining our work.
>
> Best regards,
>
> The Authors

---

### Official Review · Reviewer_1jEc · 2025-10-31

**Soundness:** 4
**Presentation:** 4
**Contribution:** 3
**Rating:** 2
**Confidence:** 4

**Summary:**

As the ability of vision-language models (VLMs) to interpret *implicit human needs* in dynamic urban environments remains underexplored, this paper proposes CitySeeker, a novel benchmark designed to evaluate VLMs’ spatial reasoning and decision-making capabilities in *embodied urban navigation* tasks that involve implicit objectives. CitySeeker includes 6,440 trajectories across 8 cities, capturing diverse visual characteristics and implicit needs within 7 goal-driven scenarios. Extensive experiments demonstrate that even top-performing models (e.g., Qwen2.5-VL-32B-Instruct) achieve only 21.1% task completion, revealing fundamental weaknesses in long-horizon reasoning, spatial cognition, and experiential recall. To further diagnose these issues, the authors explore a set of human-inspired strategies—Backtracking Mechanisms, Enriched Spatial Cognition, and Memory-Based Retrieval (BCR)—reflecting iterative observation-reasoning cycles and adaptive path optimization in human navigation. The analysis provides valuable insights into developing VLMs with more robust *spatial intelligence* for tackling “last-mile” navigation challenges.

**Strengths:**

1. The motivation of this paper is well-grounded. Identifying that VLMs’ ability to interpret implicit human needs in dynamic urban environments remains underexplored is both timely and significant. It provides a new angle for examining VLMs’ world knowledge and decision-making capabilities.

2. The paper is clearly written and visually appealing. The figures effectively illustrate the framework and experiments, helping readers understand the design and reasoning process.

3. The authors conduct extensive experiments involving 27 different VLMs, and their findings are deep and thought-provoking, revealing critical gaps in current models’ embodied spatial reasoning.

**Weaknesses:**

1. Overall, this is a clear accept-level paper in terms of novelty, clarity, and experimental depth. However, there is a **serious ethics concern**. The paper states:
   *“CitySeeker dataset was sourced from publicly available APIs (Google Maps and Baidu Maps) and is used in accordance with their terms of service for non-commercial research purposes only.”*
   After reviewing Google Maps’ Terms of Service[1], it explicitly states:
   **“Downloading Street View images to use separately from Google services (such as an offline copy) is prohibited. These restrictions apply to all academic, nonprofit, and commercial projects.”**
   This implies that **the dataset collection may violate Google’s ToS**, introducing a significant **ethical and legal issue**. Consequently, the dataset **cannot be publicly released**, which severely limits the reproducibility and extensibility of this research. This is a **veto-level weakness** for a top-tier venue.

[1] [Brand Resource Center | Products and Services - Geo Guidelines](https://about.google/brand-resource-center/products-and-services/geo-guidelines/)

**Questions:**

1. Please clarify the ethical compliance issue.

2. The paper claims that these navigation tasks are highly challenging even for humans (e.g., “humans achieve only 5.7% accuracy”). However, tasks such as finding a restaurant or locating a place with Wi-Fi do not seem inherently difficult for human participants. The paper does not adequately explain why humans perform so poorly, or how the tasks are defined, constrained, or evaluated. This undermines the interpretability and credibility of the benchmark’s human baseline.

**Details Of Ethics Concerns:**

The paper states:
   *“CitySeeker dataset was sourced from publicly available APIs (Google Maps and Baidu Maps) and is used in accordance with their terms of service for non-commercial research purposes only.”*
   After reviewing Google Maps’ Terms of Service[1], it explicitly states:
   **“Downloading Street View images to use separately from Google services (such as an offline copy) is prohibited. These restrictions apply to all academic, nonprofit, and commercial projects.”**
   This implies that **the dataset collection may violate Google’s ToS**, introducing a significant **ethical and legal issue**. Consequently, the dataset **cannot be publicly released**, which severely limits the reproducibility and extensibility of this research. This is a **veto-level weakness** for a top-tier venue.

[1] [Brand Resource Center | Products and Services - Geo Guidelines](https://about.google/brand-resource-center/products-and-services/geo-guidelines/)

---

> ### Author Response · Authors · 2025-11-20
> **Response to Reviewer 1jEc (Part 1): Thanks for your recognition of our work’s novelty and suggestion.**
>
> We are deeply grateful for your recognition of our work’s novelty and experimental depth, and for identifying it as a "clear accept-level paper" based on its scientific merit.
>
> Regarding the concern on data release and ethical compliance, we view this as a vital opportunity to ensure the rigorousness of our work. We fully align with your assessment that distributing offline copies of Google Street View images is strictly prohibited. We hope to resolve this concern through a constructive dialogue, demonstrating that our actual release strategy allows for full reproducibility without distributing any copyrighted data. Below, we clarify our adherence to the standard, ToS-compliant API-and-Scripts methodology established by prior work, hoping this positive communication can clear the path for the paper's acceptance.
>
> ### **1. Response to the Google Data Ethics Concern (W1, Q1, Ethics Flag)**
>
> We want to thank the reviewer for their diligence in checking the Google ToS. We take this concern with the utmost seriousness. We wish to clarify that our data release strategy is ToS-compliant, adhering to the widely accepted and established methodologies currently used by the Vision-Language Navigation community, as well as broader urban research communities. To be clear:
>
> * **Compliant Access:** All street-view data used during our experiments was accessed by calling the official, paid APIs from Google and Baidu, in accordance with their terms (Line 492).
> * **Compliant Release Strategy:** Our public release will not involve distributing any raw, copyrighted Google image files. Instead, following the established, ToS-compliant practice in the VLN community [1, 2, 4, 5], our public release will only contain:
>     * The 6,440 trajectory definitions (node IDs, image coordinates and timestamps, graph edges, and connectivity) and the matched implicit-need queries.
>     * Our complete code and the BCR framework, which includes scripts designed to interact with the Google and Baidu APIs.
> * **Reproducibility:** Reproducibility is ensured by providing scripts that allow researchers to use released trajectory node IDs or coordinates to re-fetch the panoramas directly from official Google Street View API [A] for this purpose. This method integrates seamlessly with our provided codebase.
>
> **Precedent in Recognized Research：** This method of accessing and interacting with street-view imagery follows the established, community-vetted standard for ToS-compliant research in top-tier venues [1-19].
>
> * **Touchdown (Chen et al., CVPR 2019) [1]:** The foundational paper for this area explicitly states: "We release the environment graph as panorama IDs and edges, scripts to download the RGB panoramas using the Google API... Retention of downloaded panoramas should follow Google’s policies." Crucially, this compliance method did not hinder Touchdown from becoming influential benchmark in outdoor navigation.
> * **V-IRL (Yang et al., ECCV 2024) [2]:** A very recent paper using Street View data states: "Moreover, V-IRL complies with the Google Maps Platform license, similarly to notable existing works that also leverage Google’s street views [1, 19]."
> * **Widespread Adoption:** This release-scripts method is widely used for virtually all major Street View benchmarks, including MIT Place Pulse [8-13], Talk2Nav [3], VELMA [4], and GenEx [5]. It is also the standard approach for numerous studies in urban computing, autonomous driving, and geospatial analysis that leverage street-view data [14-19].
>
> **CitySeeker is a Framework Beyond Raw Imagery:** Furthermore, we wish to emphasize that the core contribution of our work is not the raw imagery itself, which merely provides visual environment for interaction. The scientific value lies in the CitySeeker Framework for continuous interaction with real urban environments, the 6,440 trajectories with implicit-need queries, the BCR exploration strategies, and the in-depth experimental findings that reveal VLM weaknesses.
>
> We recognize that our original Ethics Statement were ambiguous and led to this critical misunderstanding. We will revise these sections to be explicit:
> > **Ethics & Data Availability Statement:** "The street-view imagery... was sourced from publicly available APIs (Google Maps and Baidu Maps). To ensure full reproducibility while strictly adhering to API Terms of Service, we will publicly release all trajectory data (node IDs, coordinates, and graph connectivity) and all natural language instructions. We will not release the raw panorama images themselves. Instead, we will provide scripts that allow researchers to re-fetch the image data directly from the public APIs from Google/Baidu. This release-Coordinates/IDs-and-scripts methodology ensures full reproducibility and follows the established, ToS-compliant standard of prior work (e.g., Touchdown [1])."
>
> This clarification resolves the concern and the ethics flag: our work is compliant with the ToS, and our benchmark is reproducible.

---

> > ### Comment · Reviewer_1jEc · 2025-11-24
> >
> > Thank you for your reply. Regarding W1, I believe that when the paper was initially submitted, it was clearly intended that the street-view dataset would be made publicly available, and the authors even uploaded a Baidu Disk link, which indeed contained street-view images. In order to address the concerns I raised, the authors later changed their claim to say the data were obtained via an API.
> >
> > However, if an API must be used, it would require every researcher to spend a significant amount of money to download the images. Moreover, street-view imagery changes over time, which means the POIs, graphs, and street-view images in the paper would no longer match, making the benchmark unusable.
> >
> > As for Touchdown[1], it was actually released by the Google team, so they can directly provide download links for the street-view images. VIRL[2], on the other hand, is fully integrated with the Google Maps API, allowing open-world free exploration — but at the cost of high API fees and reduced accessibility for this paper. Regardless of which of these two approaches is used, neither aligns with the authors’ current method. Therefore, I have decided to maintain my score.
> >
> > I do appreciate the authors' effort in making this benchmark. I still believe this is indeed a good paper, but from a benchmark perspective, the current design choice may not be ideal.
> >
> > [1] Chen, H., Suhr, A., Misra, D., Snavely, N., & Artzi, Y. (2019). Touchdown: Natural language navigation and spatial reasoning in visual street environments. In CVPR.
> >
> > [2] Yang, J., Ding, R., Brown, E., Qi, X., & Xie, S. (2024). V-irl: Grounding virtual intelligence in real life. In ECCV.

---

> > > ### Author Response · Authors · 2025-11-25
> > > **Response to Reviewer 1jEc (Part 1): For the streetview image Data Accessibility and Cost Analysis**
> > >
> > > We sincerely thank the reviewer for the continued engagement and for acknowledging the scientific value of our work as "**indeed a good paper**" with "novelty" and "experimental depth." We understand your concerns regarding the data availability. We respectfully submit that addressing valid concerns and rectifying oversights is the very purpose of the rebuttal process. Below, we address the specific concerns regarding cost and compliance with verifiable facts.
> > >
> > > ### **1. Accessibility and Cost Analysis**
> > >
> > > Most importantly, we wish to demonstrate that the API cost is **extremely low** and **financially accessible** for researchers. Based on the Google Maps Platform pricing (*Static Street View*: First 10,000 requests free per month, then $7 per 1,000), the cost is highly favorable for reproduction.
> > >
> > > * **Scenario A: Reproducing Test Results**
> > >
> > >   The test dataset used in our paper contains 1,257 trajectories, involving approximately 7,000 unique nodes. Since each node corresponds to a single Street View image and one API call retrieves one image, utilizing the entire test set necessitates ~7,000 API calls:
> > >
> > >     * **Cost:** **Free**.
> > >     * The required ~7,000 images fall entirely within the standard monthly **"Free Usage Cap"** (the first 10,000 requests per month are free of charge). Therefore, researchers can reproduce our main evaluation results without incurring API fees.
> > >
> > >         [https://developers.google.com/maps/billing-and-pricing/billing-overview](https://developers.google.com/maps/billing-and-pricing/billing-overview)
> > >
> > > * **Scenario B: Full Dataset** (Affordable One-Time Cost)
> > >
> > >     For researchers intending to conduct in-depth training or analysis using the complete dataset (all trajectories involve 41,128 images):
> > >     * **For Google Map Platform New Users:** **Free**.  (41,128 images will be fully covered by the standard $300 new user trial credit).
> > >     * **For Google Map Platform Existing Users:** ~$217.90.
> > >     * **Calculation details:** The first 10,000 are free. The remaining ~31,128 images are billed at $7 per 1,000 (31.128×7≈ $ 217.90).
> > >
> > >         [https://developers.google.com/maps/billing-and-pricing/pricing](https://developers.google.com/maps/billing-and-pricing/pricing)
> > >
> > > **Testing and reproduction are effectively free of charge.** Even for full-scale training, the cost (at most ~$217) is negligible compared to standard AI research expenses, such as LLM API calls or GPU rentals. It is also worth noting that Google’s developer-friendly incentive policies actively facilitate research accessibility, which aligns perfectly with your recommendation to strictly follow the ToS while maintaining community access. This represents the most viable and ethically sound solution for this type of research.
> > >
> > > ### **2. Clarification on Data Persistence and Temporal Stability**
> > >
> > > The concern that *"street-view imagery changes over time... making the benchmark unusable"* is based on a misunderstanding of the API's architecture.
> > >
> > > Google Street View uses unique panorama ID (not just coordinates). A panorama ID is a permanent identifier linked to a specific historical snapshot.
> > >
> > > **To ensure Reproducibility:** Even if the physical street changes in 2025, querying the specific panorama ID will retrieve the exact same historical image used in our experiments. This mechanism aligns directly with our revised Ethics & Data Availability Statement, where we explicitly commit to releasing "all trajectory data (node IDs...)". By releasing these specific node panorama IDs, we ensure that researchers retrieve the exact image version, immune to future street updates. This guarantees consistency and reproducibility over time.
> > >
> > > ### **3. Community Precedents and Fairness**
> > >
> > > Regarding ***Touchdown*** [1], we respectfully refer to the text of the original CVPR 2019 publication. The authors (from Cornell University and ASAPP Inc.) explicitly state their release methodology:
> > >
> > > > *"We release the environment graph as panorama IDs... scripts to download the RGB panoramas using the Google API... Retention of downloaded panoramas should follow Google’s policies."*
> > >
> > > Our API method adheres to these standard practices established by the research community. Similarly, the influential ***MIT Place Pulse*** benchmark follows this same API-based access model. Since these top-tier benchmarks have successfully served the community for years using this method, our approach is proven to be viable. Importantly, the API-based method provides a fair and legal access mechanism for researchers globally, ensuring equal opportunity for reproduction.

---

> > > > ### Author Response · Authors · 2025-11-25
> > > > **Response to Reviewer 1jEc (Part 2): For the streetview image Data Accessibility and Final Clarity**
> > > >
> > > > ### **4. The Claim on Data Access**
> > > >
> > > > You noted that we "changed our claim." We wish to clarify that this was an immediate corrective action triggered by your valid ethical concern, which we took very seriously.
> > > >
> > > > **To ensure ToS compliance:** We have revised the manuscript to strictly adhere to the Google Maps Platform Terms of Service, and ensured the anonymous repository now contains only graph and trajectory data. The final released repository will undergo a rigorous check to guarantee code usability and strict data compliance.
> > > >
> > > > This shift follows **your suggestion** and **Google's ToS**. We view this prompt correction not as a mere change in claim, but as a necessary step to align with your constructive guidance and ethical standards. The discussion is based on this revised, compliant data statement.
> > > >
> > > > ### **Final Clarity**
> > > > **We deeply respect your commitment to rigor.** We have done our utmost to resolve the ethical dilemma by seeking the optimal balance between ToS compliance and data accessibility, adopting the standard methodology widely accepted by the community.
> > > >
> > > > **We aligned our revisions with your recommendations to ensure strict ToS compliance while guaranteeing low-cost reproducibility.** We commit that the final version of the paper will align with this compliant strategy to ensure reproducibility. As demonstrated, Google’s incentive policies make this approach financially accessible, and this API-based methodology has been validated by numerous recognized works as both compliant and effective. Thus, we believe this should be regarded as a fair and standard practice.
> > > >
> > > > Given that you recognize this as a "**clear accept-level paper**" in terms of science, we earnestly hope that the **resolved logistical concerns**—now addressed with verifiable facts regarding **negligible costs** and **data persistence**—will remove your reservations. We would be deeply grateful if you could view our submission favorably based on its scientific merit and our compliance alignment.

---

> > > > > ### Comment · Reviewer_1jEc · 2025-11-25
> > > > >
> > > > > Thank you for your effort in changing the open source policy. I appreciate the effort you made. And I will raise the score to 4.

---

> ### Author Response · Authors · 2025-11-20
> **Response to Reviewer 1jEc (Part 2): Thanks for your recognition of our work’s novelty and suggestion.**
>
> ### **2. Clarification on Human Baseline Performance (Q2)**
>
> We thank the reviewer for this nuanced observation regarding task difficulty. As noted in recent spatial intelligence research [20] (Yang et al., 2025), complex zero-shot navigation in open-world environments inherently challenges both biological and artificial agents. This complexity highlights the necessity of our systematic and multi-granular evaluation framework.
>
> * **The TCE vs. TCP Distinction:** As defined in Sec 4.2, TCE (Task Completion - Exact) requires a "strict, single-node endpoint match," which means stopping on exactly one pre-defined coordinate in a large city. This is difficult, especially since a target is often visible from multiple viewpoints. We included it only for technical completeness.
>
> * **A Realistic Success Metric:** The more realistic metric for task success, as used in our main analysis (Sec 5.1), is TCP (Task Completion - Proximity). The TCP threshold is a deliberate design choice. Our environment nodes are sampled every ~20 meters, meaning a single target is often clearly visible from 2-3 consecutive nodes (a 40-60m range). The 50m TCP radius correctly accounts for this spatial ambiguity and reflects a realistic visual range for identifying urban targets, whereas the "strict, single-node" TCE metric is overly punitive. It simultaneously highlights the necessity of our comprehensive and flexible metrics and reveals important insights into human performance.
>
> **Human Superiority:** Under this reasonable metric, results show that humans (**30.1% TCP**) significantly outperform the top-performing VLM, Qwen2.5-VL-32B (**21.1% TCP**). This confirms human superiority and the credibility of our baseline. Furthermore, the human advantage is robust. This lead is more pronounced against smaller models (e.g., more than 3x that of Llava-Llama3-8B at 10.4% TCP). Humans also exhibit a clear advantage across diverse task dimensions, such as in "Transit Hub" navigation (34.9% TCP vs. the best VLM's 19.6%), common-sense inference in "Abstract Demand" tasks (31.5% vs. 24.1%), and fundamental goal-finding in "Basic POI" tasks (31.8% vs. 22.2%).
>
> **Experimental Rigor:** The experimental design required strict consistency between human and model settings. Human participants operated under the exact same stringent constraints as the models: a possible unfamiliar city environment, and notably, often facing different linguistic cues (e.g., an Asian participant navigating London, relying solely on visual recognition of non-native signs). Navigating an unknown city with a strict 35-step limit and no map, relying purely on visual cues, is genuinely difficult. This setup ensures consistency with the model evaluation and rigorously tests the fundamental spatial cognition capability derived from general life experience.
>
> In summary, the human baseline is credible, performs as expected, and correctly highlights the challenge of this task. We will clarify this distinction between TCE and TCP in the main paper to avoid future confusion.

---

> > ### Author Response · Authors · 2025-11-20
> > **Response to Reviewer 1jEc (Part 3): Summary and References.**
> >
> > **Summary**: We are deeply passionate about this new research direction, and we thank you again for your positive ratings on the paper's core scientific contribution. We hope our clarification demonstrates that the critical concern stemmed from an ambiguity in our text regarding the standard, ToS-compliant practice, which we will explicitly correct in the revision. We sincerely appreciate your rigorous review, which we view as a collaborative effort that has helped us ensure strict ethical compliance.
> >
> > Given your assessment that this is otherwise a "clear accept-level paper," we earnestly hope this constructive exchange has resolved your reservations. We would be deeply grateful if you could reconsider your evaluation, allowing this work to contribute to the community.
> >
> > **References**
> >
> > **API Links**
> > [A] Google Street View API:
> > [https://developers.google.com/maps/documentation/streetview/overview](https://developers.google.com/maps/documentation/streetview/overview)
> > [https://developers.google.com/maps/documentation/javascript/streetview](https://developers.google.com/maps/documentation/javascript/streetview)
> >
> > **Cited Works**
> >
> > [1] Chen, H., Suhr, A., Misra, D., Snavely, N., & Artzi, Y. (2019). Touchdown: Natural language navigation and spatial reasoning in visual street environments. In CVPR.
> >
> > [2] Yang, J., Ding, R., Brown, E., Qi, X., & Xie, S. (2024). V-irl: Grounding virtual intelligence in real life. In ECCV.
> >
> > [3] Vasudevan, A. B., Dai, D., & Van Gool, L. (2021). Talk2nav: Long-range vision-and-language navigation with dual attention and spatial memory. IJCV.
> >
> > [4] Schumann, R., Zhu, W., Feng, W., Fu, T. J., Riezler, S., & Wang, W. Y. (2024). Velma: Verbalization embodiment of llm agents for vision and language navigation in street view. In AAAI.
> >
> > [5] Lu, T., Shu, T., Xiao, J., Ye, L., Wang, J., Peng, C., ... & Chen, J. (2025). GenEx: Generating an Explorable World. In ICLR.
> >
> > [6] Schumann, R., & Riezler, S. (2021). Generating landmark navigation instructions from maps as a graph-to-text problem. In ACL.
> >
> > [7] Mirowski, P., et al. (2018). Learning to navigate in cities without a map. In NeurIPS.
> >
> > [8] Salesses, P., Schechtner, K., & Hidalgo, C. A. (2013). The collaborative image of the city: mapping the inequality of urban perception. PloS one.
> >
> > [9] Naik, N., Philipoom, J., Raskar, R., & Hidalgo, C. (2014). Streetscore--predicting the perceived safety of one million streetscapes. In CVPR.
> >
> > [10] Naik, N., Raskar, R., & Hidalgo, C. A. (2016). Cities are physical too: Using computer vision to measure the quality and impact of urban appearance. The American Economic Review.
> >
> > [11] Dubey, A., Naik, N., Parikh, D., Raskar, R., & Hidalgo, C. A. (2016). Deep learning the city: Quantifying perception at a global scale. In ECCV.
> >
> > [12] De Nadai, M., Vieriu, R. L., Zen, G., Dragicevic, S., Naik, N., Caraviello, M., ... & Lepri, B. (2016). Are safer looking neighborhoods more lively?. In ACM Multimedia.
> >
> > [13] Naik, N., Kominers, S., Raskar, R., Glaeser, E., & Hidalgo, C. A. (2017). Computer vision uncovers predictors of physical urban change. PNAS.
> >
> > [14] Anguelov, D., Dulong, C., Filip, D., Frueh, C., Lafon, S., Lyon, R., ... & Vincent, L. (2010). Google street view: Capturing the world at street level. Computer.
> >
> > [15] Ro, J., Kim, N., & Yoon, Y. (2025). How Well Do Vision-Language Models Understand Cities? A Comparative Study on Spatial Reasoning from Street-View Images. In ICCV.
> >
> > [16] Xue, X., Tian, Z., Yang, Y., Wang, J., & Cao, S. J. (2025). Sustaining the local color of a global city. Nature Cities.
> >
> > [17] Dahir, N., Sheng, H., Yao, K., Goel, S., & Hwang, J. (2025). Surveillance camera prevalence and racial diversity in ten US cities. Nature Cities.
> >
> > [18] Quintana, M., Gu, Y., Liang, X., Hou, Y., Ito, K., Zhu, Y., ... & Biljecki, F. (2025). Global urban visual perception varies across demographics and personalities. Nature Cities.
> >
> > [19] Ardeshir, S., & Zamir, A. R. (2014). Image geo-localization based on multiple nearest neighbor feature matching using generalized graphs. TPAMI.
> >
> > [20] Yang, J., Yang, S., Gupta, A. W., Han, R., Fei-Fei, L., & Xie, S. (2025). Thinking in space: How multimodal large language models see, remember, and recall spaces. In CVPR.

---

> ### Author Response · Authors · 2025-11-25
> **Response to Reviewer 1jEc: We appreciate for your guidance and re-evaluation**
>
> We thank the reviewer for the time and effort spent engaging in this discussion and for acknowledging our efforts to ensure ethical compliance.
>
> **We are glad that our revised open-source policy alleviated your concerns, and appreciate your decision to raise the score to 4**. We fully agree that this rigorous check was necessary, and we believe the paper is now much stronger and safer for the community thanks to your supervision.
>
> We commit to **strictly implementing the API and scripts release method** in the final version, **ensuring full compliance with Google's ToS while maintaining reproducibility for future researchers.**
>
> Thank you again for recognizing the scientific value of our work.

---

### Official Review · Reviewer_Xv86 · 2025-11-01

**Soundness:** 2
**Presentation:** 3
**Contribution:** 2
**Rating:** 4
**Confidence:** 4

**Summary:**

This paper introduces CitySeeker, a framework that leverages VLMs to understand and predict human urban mobility behaviors based on natural-language queries. The system maps user needs to POI categories using multimodal embeddings and evaluates performance on multiple cities. The goal is to test VLMs’ ability to align semantic and spatial reasoning in real-world city environments.

**Strengths:**

Addresses an interesting interdisciplinary question.

The dataset construction across multiple major cities, combining geospatial and textual information.

The evaluation is systematic, covering both semantic matching and spatial reasoning tasks.

The idea of connecting natural language intent to spatial decision-making is novel and potentially impactful.

**Weaknesses:**

The mapping from need to POI type assumes a fixed, deterministic relationship, which may not hold in practice, human intent is subjective and context-dependent.

The model implicitly assumes people choose the shortest path or most direct POI option, which is unrealistic, behavioral factors like preference, familiarity, and accessibility play major roles.

Cross-cultural generalization is a concern: the same “need” may imply different POIs across societies.

The paper lacks an analysis of cultural and linguistic bias, despite using mixed data from Beijing and New York.

The need-to-POI mapping seems to reflect designer bias rather than emergent patterns from real user behavior.

No mention of whether the system can adapt to multi-intent or ambiguous queries.

It’s unclear whether the evaluation reflects real mobility choices or just semantic alignment accuracy.

**Questions:**

How is user intent variability modeled, are there multiple valid POIs for the same need, or just one ground truth?

How does the model handle ambiguous or multi-intent queries?

Is the need-POI mapping empirically validated with real mobility data (e.g., GPS traces, check-ins)?

Could the framework integrate behavioral priors (e.g., time-of-day, personal preferences) to better capture real-world decision patterns?

---

> ### Author Response · Authors · 2025-11-20
> **Response to Reviewer Xv86 (Part 1):  Thanks for your thoughtful review.**
>
> We sincerely thank Reviewer for the detailed review and constructive feedback. We are greatly encouraged that you recognize our work's "interesting interdisciplinary question, the value of our dataset, our "systematic evaluation," and the "novel and potentially impactful" idea of connecting language intent to spatial decision-making.
>
> We understand your concerns, which may stem from potential ambiguity in our paper regarding our task definition, evaluation focus, and mapping methodology. We address each concern below. We hope these clarifications—plus a new human supplementary experiment conducted in response to this review—will demonstrate the soundness and contribution of our work.
>
> ### **1. Validating Need-to-POI Mapping: Addressing Bias and Empirical Grounding (W1, W5, Q1, Q3)**
>
> Your core concern is that our Need-to-POI mapping is "fixed" (W1), may reflect "designer bias" (W5), and is not "empirically validated" (Q3). This is a nuanced point, and we offer a multi-part clarification.
>
> **1.1 Clarification on our Task Goal**
>
> First, we wish to clarify our scientific goal. This benchmark is not designed to build a personalized mobility predictor. Instead, our goal is to establish the first benchmark for evaluating a VLM's foundational **spatial common sense**: can it understand an abstract human need (e.g., "I'm thirsty") and visually ground it to any plausible real-world solution?
>
> **1.2 Clarification on Mapping Methodology (W1, Q1, W5)**
>
> Our framework is predefined based on common sense on query-to-POI categories (e.g., restaurant) rather than specific POI instances (e.g., James's Restaurant), to model intent variability.
>
> * **For "Common Sense" Needs (e.g., "Abstract Demand"):** As shown in Appendix B.1 (Tables 4 & 5), we explicitly map "I'm thirsty" to a set of POI categories, including {e.g., Convenience store, Supermarket, Cafe, Bubble tea store, Vending machine}. This is an one-to-many relationship, not a 1:1 relation.
> * **For "Subjective" and "Specific" Needs:** These mappings are already data-driven. As detailed in Appendix B.1, categories like "romantic restaurant" or "wheelchair accessible" are mapped from user-generated POI metadata tags (e.g., Google's POI attributes), rather than originating from designer bias.
> * **Localization Strategy:** Crucially, our mapping accounts for regional service differences by verifying brand-specific affordances through social media data retrieval. While broad categories (e.g., "Telecommunications Provider") are universal, specific retail functions vary by city. For example, for the query "I need to recharge my phone":
>     * In **Hong Kong**, the mapping includes **7-Eleven** and **Watsons**, which locally sell SIM top-up cards; In **New York**, it extends to **Best Buy** and **Target**; In **London**, it includes **Tesco**.
>
> These region-specific inclusions are not arbitrary but are grounded in verified service capabilities derived from local social data, ensuring the ground truth reflects actual local urban life.
>
> **1.3 New Supplementary Experiment: Human Consensus Survey (N=120) (W3, W5, Q3)**
>
> To prove that our common sense mappings are not designer bias (W5) but reflect robust human common sense (Q3) with cross-cultural generalization (W3), we conducted a new supplementary survey.
>
> **Methodology:** We surveyed 120 participants across four regions (30 North America, 40 Asia, 30 Europe, 20 Other). We presented our 10 abstract queries (e.g., "I'm feeling thirsty..."). For each question, participants selected from ~50 POI categories. These options included our pre-defined mapping categories mixed into a large, systematic pool of ~40 other POI categories drawn from the entire Baidu/Google POI classification system.
>
> **Key Finding 1 (for W5: Designer Bias):** The results show a massive consensus cliff.
> * Our pre-defined mapping options received a **Global Average Consensus of 83.39%**.
> * The unselected "other" POI categories received a **Global Average Consensus of only 1.90%**.
> This **81% gap** proves that our ground truth mappings are not designer bias but are the clear, emergent human common sense that surfaces from the noise.
>
> **Key Finding 2 (for W3: Cultural Generalization):** The consensus was highly consistent on average.
> The **Average Cross-Cultural Standard Deviation (SD)** for our ground truth options was only **7.02%**. This low SD value quantitatively proves high overall consistency across cultural regions. As shown in **Table R1**, universal concepts like Supermarket and Fruit and vegetable store (for "fruit") or Pharmacies and Hospitals (for "not well") show extremely low SDs.

---

> ### Author Response · Authors · 2025-11-20
> **Response to Reviewer Xv86 (Part 2): Thanks for your thoughtful review.**
>
> **Table R1: Cross-Cultural Consistency Analysis**
> *POI categories with Global Consensus ≥ 85% are shown. Low SD (<10%) across regions confirms these are universal common sense choices rather than designer bias. All analysis results will be updated in the appendix.*
>
> | Question | POI Category | Mean Consensus | Cross-Cultural SD |
> | :--- | :--- | :--- | :--- |
> | Q10: I want to buy fruits... | Supermarket | 97.78% | 3.85% |
> | Q10: I want to buy fruits... | Fruit and vegetable store | 95.28% | 2.41% |
> | Q10: I want to buy fruits... | Greengrocer | 91.11% | 1.92% |
> | Q1: I'm feeling hungry... | Restaurant | 94.72% | 2.41% |
> | Q1: I'm feeling hungry... | Supermarket | 93.06% | 3.37% |
> | Q1: I'm feeling hungry... | Convenience store | 87.50% | 4.33% |
> | Q2: I'm feeling thirsty... | Convenience store | 98.89% | 1.92% |
> | Q2: I'm feeling thirsty... | Cafe | 97.22% | 2.55% |
> | Q2: I'm feeling thirsty... | Supermarket | 90.00% | 6.67% |
> | Q2: I'm feeling thirsty... | Vending machine | 85.83% | 2.20% |
> | Q3: I'm not feeling well... | Pharmacies | 97.22% | 2.55% |
> | Q3: I'm not feeling well... | Hospitals | 95.56% | 5.09% |
> | Q3: I'm not feeling well... | Clinics | 90.00% | 6.67% |
> | Q4: I want to rest and read... | Public library | 92.50% | 6.61% |
> | Q4: I want to rest and read... | Library | 91.11% | 1.92% |
> | Q4: I want to rest and read... | Coffee shop | 89.72% | 5.55% |
> | Q4: I want to rest and read... | Cafe | 88.61% | 4.74% |
> | Q5: I want to work with Wi-Fi... | Cafe | 100.00% | 0.00% |
> | Q5: I want to work with Wi-Fi... | Coffee shop | 98.89% | 1.92% |
> | Q5: I want to work with Wi-Fi... | Library | 98.33% | 2.89% |
> | Q5: I want to work with Wi-Fi... | Public library | 95.00% | 1.67% |
> | Q5: I want to work with Wi-Fi... | Book store | 85.56% | 3.85% |
> | Q6: I need to recharge phone... | Telecommunications service provider | 97.78% | 3.85% |
> | Q6: I need to recharge phone... | Cell phone store | 96.39% | 3.76% |
> | Q6: I need to recharge phone... | 7-ELEVEn | 92.22% | 8.39% |
> | Q7: I want to exercise... | Gym | 100.00% | 0.00% |
> | Q7: I want to exercise... | Park | 99.17% | 1.44% |
> | Q7: I want to exercise... | City park | 91.11% | 1.92% |
> | Q7: I want to exercise... | Physical fitness program | 89.17% | 1.44% |
> | Q8: I want to bring my child... | Playground | 96.94% | 0.48% |
> | Q8: I want to bring my child... | City park | 96.67% | 5.77% |
> | Q8: I want to bring my child... | Children’s amusement center | 93.89% | 0.96% |
> | Q8: I want to bring my child... | Park | 92.78% | 2.55% |
> | Q8: I want to bring my child... | Museum | 90.56% | 4.19% |
> | Q9: I need airport transport... | Subway station | 93.89% | 3.47% |
> | Q9: I need airport transport... | Taxi service | 89.72% | 2.93% |
> | Q9: I need airport transport... | Bus stop | 89.44% | 4.81% |
> | Q9: I need airport transport... | Bus station | 88.89% | 5.09% |
>
> **1.4 Clarification on Route Validation (Q3)**
>
> Finally, while we did not use GPS traces (as our goal is not prediction), we did perform rigorous route validation. As stated in Sec 3.1 and Appendix B.1, we strictly "manually verify each route" of the 6,440 trajectories to ensure no other valid POIs exist along the path (except at the endpoint). This ensures our ground truth routes are unambiguous and the task is well-defined.
>
> ### **2. Clarifying Evaluation: Success is Path-Agnostic (W2, W7)**
>
> There is a misunderstanding that we assume agents must follow the "shortest path" (W2). This actually highlights the systematic nature and flexibility of our evaluation design.
>
> * **Shortest Path is for Generation, not Evaluation:** As described in Sec 3.1, we use A\* only to generate a single, efficient reference trajectory. This is standard practice (e.g., in R2R) to create a ground truth for efficiency metrics. It is not a constraint on the agent's behavior.
> * **Our Success Metrics are Flexible (W7):** Our primary success metrics (Section 4.2) like TC-Proximity (TCP) and TC-Category (TCC) are path-agnostic.
>     * **TCC (Category):** Even if the agent does not stop at the specific ground truth POI (e.g., the Convenience Store mapped for "thirsty"), as long as it finds a POI of a valid category (e.g., a Cafe), it is considered a success.
>     * **TCP (Proximity):** If the agent stops within 50m of the target, it is 100% successful, regardless of how far or circuitous the path was.
>     * **Only nDTW** is used to measure path efficiency.

---

> ### Author Response · Authors · 2025-11-20
> **Response to Reviewer Xv86 (Part 3): Thanks for your thoughtful review.**
>
> ### **3. On Regional Bias and Ambiguity (W3, W4, W6, Q2)**
>
> **3.1 Clarification on Ambiguous/Multi-Intent Queries Are a Core Feature (W6, Q2)**
>
> In fact, Cityseeker is designed for such queries. The point raised by reviewer is exactly one of the core challenges of our work.
>
> Our Abstract Demand (Category 5) and Semantic Preference (Category 7) are inherently ambiguous or multi-intent.
> * E.g.: "find the nearest **romantic** restaurant" is highly subjective.
> * E.g.: "**I want to work with Wi-Fi**" is a multi-intent need (Work + Wi-Fi).
> * E.g.: "**Please find the nearest restaurant with an accessible entrance**" (from Category 6) is a multi-intent query (Restaurant + Accessibility) (Table 6).
>
> As shown in Table 2, VLM performance on Abstract Demand (24.1%) and Semantic Preference (26.0%) is significantly lower than Brand-Specific (30.4%). This proves the reviewer's point—handling these ambiguous queries is a key weakness of current VLMs, and CitySeeker is designed to expose precisely this.
>
> **3.2 Clarification on Cultural/Linguistic Bias (W3, W4)**
>
> The reviewer notes a "lack of analysis of cultural and linguistic bias" (W4). We are happy to clarify that this analysis already exists in our paper.
>
> * **Linguistic Bias (W4):** We ran tests Appendix C.4 (Cross-Lingual Experiment) in Beijing using both English and localized Chinese prompts. The results (Table 10, excerpt below) showed no consistent improvement, strongly supporting our conclusion:
>     > "This high degree of variability strongly suggests that the observed performance gaps between cities are not primarily driven by linguistic factors but are likely rooted in deeper visual and geographic biases within the models’ training data."
>
> * **Cultural Generalization (W3):** Our benchmark's construction across 8 global cities is the test for this. Furthermore, our new N=120 supplementary experiment (Sec 1.3 above) now provides quantitative proof (Average Cross-Cultural SD: 8.40%) of the high cultural consistency of our core "common sense" mappings.
>
> **Table 10 (Excerpt): Cross-Lingual Performance**
> *Localizing prompts to Chinese did not yield consistent improvements.*
>
> | Model | City | Language | TCP (%) | nDTW |
> | :--- | :--- | :--- | :--- | :--- |
> | Qwen2.5-VL-32B | Beijing | English | 25.95 | 122.4 |
> | | Beijing | Chinese | 23.91 | 120.36 |
> | GPT-4o-mini | Beijing | English | 7.32 | 345.36 |
> | | Beijing | Chinese | 7.32 | 217.31 |
>
> ### **4. On Behavioral Priors and GPS-Based Mapping (Q3, Q4)**
>
> We agree that empirical data (GPS/LBSN) captures realistic behavioral patterns. We seriously considered the mobility-mining method but deliberately chose the semantic-consensus approach to ensure benchmark fairness, as GPS/LBSN suffer from fundamental limitations for reasoning evaluation:
>
> 1.  Sparsity and false negatives: Traces reflect specific user habits (e.g., strictly visiting Starbucks). Using them as ground truth would erroneously classify valid alternatives (e.g., a vending machine) as failures. This would evaluate the vlm's ability to mimic user history rather than solve the semantic problem.
>
> 2.  Intent ambiguity:
>     * **GPS Ambiguity:** Traces record behavioral outcome (visiting a location) but often miss the semantic motivation. A single location like McDonald's satisfies multiple latent needs (hunger, thirst, restroom, Wi-Fi). Reverse-engineering a specific need (e.g., hunger) from a trace is inherently speculative. Using such data would force us to inject unverified assumptions into the ground truth, polluting the semantic evaluation with noise.
>     * **Social Media Noise:** Check-ins often contain semantic noise. For instance, a user posting about a "breakup" at a  restaurant does not map to "I am hungry." Direct mapping may create false semantic correlations.
>
> 3.  Incomplete category coverage: Users frequently generate data for socially significant locations (e.g., popular cafe, tourist spots) but rarely leave traces for mundane but essential functional facilities (e.g., restroom, ATM). Relying on traces would leave these categories unevaluated.
>
> To address your valid concern about empirical grounding while mitigating the inherent noise of raw traces, we conducted the human consensus survey (N=120) described above. This serves as a controlled empirical alternative, aiming to maximize semantic coverage and minimize label ambiguity.
>
> For future work: We agree that incorporating behavioral priors (e.g., time-of-day, personal preferences) is the logical next step for personalization. Once a VLM demonstrates the foundational spatial common sense tested by CitySeeker, trace-based priors should be used to rank these valid candidates. We will explicitly discuss this as a key future direction inspired by your feedback.
>
> **Summary**: We hope these clarifications, supported by new supplementary experiment with your constructive feedback, have resolved your concerns. We respectfully hope this additional validation provides a solid basis for updating assessment.

---

> ### Author Response · Authors · 2025-11-26
> **Response to Reviewer Xv86: We thanks again for your valuable time and look forward to further discussion**
>
> We thank you again for your valuable time and the insightful review.
>
> In our rebuttal, we have prioritized addressing your core concerns regarding designer bias and cross-cultural generalization in our Need-to-POI mappings. Specifically, we have:
>
> 1.  **Conducted a new, large-scale supplementary experiment** (N=120 participants across 4 regions). The results (global consensus: **83.39%**, distractor: **1.90%**) provide robust empirical evidence that our mappings reflect emergent human common sense.
> 2.  **Clarified our localization strategy,** detailing how we account for regional affordances (e.g., 7-Eleven vs. Best Buy for phone charging) to address cultural nuances.
> 3.  **Reiterated our cross-lingual analysis** (Appendix C.4), which empirically suggests that the performance bottleneck lies in spatial reasoning rather than linguistic bias.
> 4.  **Addressed the limitations of mobility-based data** (e.g., sparsity, ambiguity) to explain our rationale for choosing semantic consensus as the ground truth.
>
> **We deeply value the opportunity to engage in this and further discussion**. We would be grateful to know if these clarifications and the new empirical data have satisfactorily addressed your concerns.
>
> We earnestly hope our extensive new experiments and clarifications demonstrate the rigor of our work and provide a solid basis for you to kindly update your evaluation. If there are any remaining questions, we are eager to address them with utmost effort during these final days.
>
> Best regards,
>
> The Authors

---

> > ### Author Response · Authors · 2025-11-28
> > **Response to Reviewer Xv86**
> >
> > We are writing to express our appreciation for your time. We put significant effort into addressing your concerns, while we feel a slight regret that we could not discuss these results with you further, we remain hopeful that our response has clarified the validity of our approach.
> >
> > If you have any further insights or if there are aspects we haven't fully covered, we would still value your perspective as we continue to refine this research in the future.
> >
> > Best regards,
> >
> > The Authors

---

### Official Review · Reviewer_agkw · 2025-11-01

**Soundness:** 3
**Presentation:** 3
**Contribution:** 4
**Rating:** 6
**Confidence:** 5

**Summary:**

This paper introduces CitySeeker, a comprehensive benchmark designed to evaluate the spatial reasoning and decision-making capabilities of vision-language models (VLMs) in the context of embodied urban navigation for addressing implicit user needs. Extensive evaluations across a wide range of VLMs reveal key limitations in long-horizon reasoning, and the authors propose effective strategies to improve model performance.

**Strengths:**

- Clear and Well-Structured: The paper is well-organized, with thorough explanations of the data collection process, benchmark design, and task formulation.

- Novel and Interesting Setting: The paper proposed the task of embodied urban navigation guided by implicit human needs. This task is currently not widely explored and has significant potential for real-world deployment.

- Extensive Evaluations: A wide range of VLMs are evaluated on the curated benchmarks, accompanied by comprehensive analysis and discussion.

- Actionable Insights: The authors propose concrete strategies to enhance VLM performance, offering insights for real-world deployment scenarios.

**Weaknesses:**

I don't find significant weaknesses in this submission. However, I do have some concerns as follows. Therefore, I give a conservative score of borderline accept. I may consider increasing the rating if the authors adequately address these concerns.

- Open-Source Model Superiority: The paper observes that open-source VLMs (such as Qwen) occasionally outperform the proprietary VLMs. The submission would benefit from a deeper analysis of the underlying reasons behind this phenomenon.

- Presentation Issue: Some figures in the submission (e.g., Figures 5 and 6) require further refinement to improve readability and visual clarity.

- The manuscript would benefit from including illustrations of typical failure cases.

**Questions:**

See Weaknesses

---

> ### Author Response · Authors · 2025-11-20
> **Response to Reviewer agkw: We appreciate your valuable insight and recognition of our work.**
>
> We sincerely thank Reviewer for the positive feedback and constructive comments. We are encouraged by your recognition of our work's "novel and interesting setting" and "extensive evaluations." You raised valuable concerns, we addressed each point in depth and made substantial revisions below.
>
> ### **Concern 1: Open-Source Model Superiority**
>
> Thank you for the insightful question. Building on your feedback, we have conducted a deeper analysis and hypothesize that their superiority stems from specific advantages in training data, architecture, and training strategies that align well with the CitySeeker task.
>
> **A) Analysis of Qwen2.5-VL (e.g., 32B):**
> We attribute its success to a strong alignment between its training curriculum and our task demands:
>
> **Highly Relevant Pre-training Data:** Qwen's technical report reveals its pre-training data includes types crucial for navigation:
> * **Specialized Spatial & OCR Data:** It was explicitly trained on multi-language OCR and, critically, localization data. This trains the model to understand spatial relationships and street-level text (e.g., storefront signs), a core challenge in our benchmark.
> * **Sequential Visual Knowledge:** The training mix included video descriptions and video localization. This exposure to sequential visual information is highly analogous to the step-by-step nature of a navigation trajectory.
>
> **Efficient Visual Architecture for High-Resolution Image:** CitySeeker provides large, multi-view panoramas at each step. Qwen's ViT architecture (Interleaved Window Attention) efficiently processes high-resolution imagery, allowing it to capture both global context (for direction) and fine-grained local details (like a distant sign).
>
> **SFT for Spatial Grounding:** Qwen's SFT uses "format-aware packing," a technique that specifically trains the model to understand the spatial correspondence between a textual description and pixel locations (e.g., linking "the cafe on the right" to the image). This directly hones the visual-spatial-text alignment our benchmark demands.
>
> **B) Analysis of InternVL3 (e.g., 38B):**
> InternVL3 achieved similarly strong performance (19.3% TCP). We analyze that its unique strategies offer complementary advantages:
>
> **Native Multimodal Pre-Training:** Unlike the train LLM, then align paradigm, InternVL3 is trained on text and image-text data natively and concurrently. This deeper fusion may enhance its ability to perform the complex cross-modal reasoning required by "implicit needs" (e.g., interpreting "I'm thirsty").
>
> **Dynamic High-Resolution Strategy:** For high-res inputs, InternVL3 uses a tile + thumbnail approach. This allows it to analyze local patches of the panorama in detail while retaining a sense of the global scene.
>
> **Advanced Mixed Preference Optimization (MPO):** InternVL3 uses Mixed Preference Optimization (combining DPO and Quality Loss) after SFT. This sophisticated strategy likely makes it more robust at multi-step reasoning and complex instruction following, which might reduce reasoning errors (Type 3) and parsing errors (Type 6).
>
> We will incorporate this comparative analysis into Section 5.2 to provide a clear answer to why these open-source models excel.
>
> ### **Concern 2: Presentation Issue (Figures 5 & 6)**
>
> We fully agree that clear and readable figures are essential. We have revised both figures based on your feedback to ensure they are publication-ready:
>
> **Global Improvements:** We have significantly increased the font size of all labels, annotations, and icons in both figures. We also refined the legends to ensure they are legible in both print and digital versions.
>
> **For Figure 6 (Trajectory Maps):** We bolded all trajectory lines to make them stand out clearly against the map background. Most importantly, we added explicit labels to each example trajectory (e.g., #47, #63), identifying it as a "Success" or "Failure" and specifying its failure category (e.g., "Failure: Oscillatory Detour" or "Failure: Critical Decision Error"). This directly links the visual evidence in the figure to our error analysis in Appendix G.
>
> ### **Concern 3: Failure Case Illustrations**
>
> We agree that connecting our error analysis to visual evidence is critical. Our Appendix G provides a detailed categorization of failure modes (e.g., the 6 error types in Figure 18).  As per your suggestion, we will expand Appendix G in the revision to include more visual examples for each of the primary error types identified in Figure 18. These examples follow the detailed format of Figure 10, clearly showing what the VLM saw, what it thought (Rationale), and what it did wrong (Action) at the critical step, providing strong visual backing for our error analysis.
>
> We commit to incorporating all these visual enhancements and figure updates into the final version of the manuscript. We thank again for this valuable feedback and hope these revisions address your concerns.

---

> > ### Comment · Reviewer_agkw · 2025-11-25
> >
> > Thanks for the rebuttal. I have updated my rating to 8.

---

> ### Author Response · Authors · 2025-11-25
> **Response to Reviewer agkw: We appreciate your recognition of our work.**
>
> We are deeply grateful for your support and recognition for updating your rating to **8**. We truly appreciate your constructive feedback throughout this process.
>
> We remain fully committed to incorporating all the discussed revisions into the final camera-ready version to ensure a high-quality contribution to the community.
>
> Thank you again for your time and valuable endorsement.

---

> > ### Author Response · Authors · 2025-11-28
> > **Response to Reviewer agkw: heartfelt thanks for your inspiring support**
> >
> > We would like to express our deepest gratitude once again.
> >
> > Your high recognition is not only an approval of this specific work but also serves as a light that warms us. Thank you for being such an inspiring part of our research journey.
> >
> > Best regards,
> >
> > The Authors

---

### Author Response · Authors · 2025-11-30
**Rebuttal Summary for ACs and PCs**

Dear ACs and PCs,

**We deeply appreciate your dedication** in overseeing the review process, especially given the unforeseen challenges and heavy workload. We respectfully submit the summary to highlight the turnaround achieved during discussion phase. Amidst this, we have remained solely focused on the scientific discourse, striving to address every concern with **pure, unadulterated effort**.

### **Current Status Overview:**

Based on the **explicit recommendations provided by reviewers** during discussion, the **average score has risen to 5.5**, while one reviewer did not reply.

### **Average score:  5.5**
* 1. Reviewer agkw: **6 to 8 (Accept)**
* 2. Reviewer 4hXJ: **4  to 6 (Weak Accept)**
* 3. Reviewer 1jEc: **2  to 4 (Ethics Concern Resolved)**
* 4. Reviewer Xv86: **4 (Regrettably, we did not receive further engagement)**

### **1. Reviewer agkw: Score raised 6 $\rightarrow$ 8 (Accept)**
**Initial Assessment:** Highlighted the work's "Novel and Interesting Setting" "Extensive Evaluations" and "don't find significant weaknesses," while requesting a deeper analysis of open-source model superiority and visual refinements.

**Our Solution:** We provided a deep theoretical analysis of why open-source VLMs outperform proprietary ones and refine visualizations.

**Final Quote:** *"**I have updated my rating to 8.** "* The reviewer highly endorsed our analysis and the value of benchmark.

### **2. Reviewer 4hXJ: Score raised 4 $\rightarrow$ 6 (Weak Accept)**
**Initial Assessment:** Recognized that CitySeeker "fills a critical gap" with "Realism and Diversity," noted the "Comprehensive Evaluation" across 27 VLMs and "well-written, with a clear logical flow". Raised concerns regarding POMDP formulation and metric suitability.

**Our Solution:** We clarified the connection between our BCR strategies and POMDPs, and crucially, integrated the **Success weighted by Path Length (SPL)** metric to measure efficiency, as suggested.

**Final Quote:** ***"The rebuttal is exemplary... The inclusion of the SPL analysis... is a particularly valuable addition... The rigour demonstrated in the rebuttal alleviates my initial technical concerns.** "* The reviewer recognized SPL analysis and updated recommendation.

### **3. Reviewer 1jEc: Score raised 2 $\rightarrow$ 4 (Ethics Resolved)**
**Initial Assessment:** The reviewer assessed scientific substance as "**clear accept-level paper** in terms of novelty, clarity, and experimental depth."', identifying data policy issue as concern on the initial rating.

**Our Solution:** We **actively followed reviewer's suggestion** to resolve the ethics concern by committing **not to release offline street view images**. Instead, we updated our policy to the "API Scripts" method. We clarified that raw imagery is not our core contribution and that our **updated data acquisition method**—validated by **top-tier benchmarks/work like Touchdown (CVPR), MIT Place Pulse, and GenEx (ICLR)**—is strictly compliant with Google’s ToS and **incurs no financial cost** for standard reproduction.

**Final Quote:** ***"Thank you for your effort in changing the open source policy... I appreciate the effort you made... raise the score to 4.** "* Our response resolved reviewer's concerns regarding potential data ethics. We demonstrated the dataset's reproducibility by confirming the wide applicability of retrieving images via Panorama IDs using the API.

### **4. Reviewer Xv86: Score 4 (No Response)**
**Initial Assessment:** Acknowledged the work as "Novel and potentially impactful" and "systematic," questioned potential "designer bias" in mappings and lack of behavioral priors (e.g., GPS, checkins).

**Our Solution:** We addressed this by conducting a **new large-scale Human Consensus Survey** showing **83.39% global consensus** (vs. 1.90% noise), and clarifying that GPS/checkins confounds behavioral outcomes with ambiguous latent intent, introducing significant noise. Our approach filters this noise to isolate **foundational spatial common sense**: establishing whether a VLM can truly understand an abstract human need and visually ground it to a functional real-world entity.

**Status:** Although the reviewer has not responded, our new data provides a robust, objective defense against the subjectivity argument.

### **Summary**
We are deeply grateful for the constructive challenges to make our work more legally compliant, technically robust, and empirically grounded. This effort has crystallized into a **consensus** on work's scientific contribution, supported by **R-agkw's rating update to 8**, **R-4hXJ's confirmation** that "rigour... alleviates initial technical concerns," and **R-1jEc's recognition** of a "clear accept-level paper" with resolved ethics.

**We trust in the fair assessment of Area Chair and ICLR. We earnestly hope you consider this whole trajectory and the quality recognized by reviewers, as we remain passionate about advancing this research direction.**

Best regards,

The Authors

---

> ### Author Response · Authors · 2025-12-04
> **Rebuttal Summary for ACs and PCs (Supplementary)**
>
> Dear ACs and PCs,
>
> We sincerely thank you for your dedication in overseeing this review process. We also affirm our strict adherence to academic integrity and transparency throughout this rebuttal. We have taken this opportunity to significantly strengthen the manuscript through the execution of large-scale supplementary human experiments, rigorous metric integration, and strict policy adjustments. We hope these substantial improvements demonstrate the work's alignment with the scientific standards of the venue. The revised manuscript, incorporating all discussant feedback, has been uploaded.
>
> ### **1. Current Status Overview**
>
> Following the rebuttal, the average score has risen to **5.5**. We have secured a strong support with **Accept (8)** and **Confidence (5)** with **"don't find significant weaknesses"**  from Reviewer agkw and a **Weak Accept (6)** from Reviewer 4hXJ, both of whom explicitly endorsed the work's novelty and rigorous evaluation.
>
> ### **2. Response to Potential Residual Concerns**
>
> We wish to address two specific points to ensure a comprehensive assessment of the paper’s merit:
>
> **2.1 Regarding Data Compliance and Accessibility (Reviewer 1jEc)**:
>
> While Reviewer 1jEc raised their score to 4 after we resolved the ethics flag, we believe the evaluation should reflect the reviewer's initial assessment of the work as a **"clear accept-level paper in terms of novelty, clarity, and experimental depth."**
> * **Full Adherence to Community Standards:** Our revised data release strategy, i.e., providing scripts to access street view imagery via official APIs, is an **established community methodology** for complying with Terms of Service. **This approach is validated by numerous top-tier works, including GenEx (ICLR), Touchdown (CVPR), and MIT Place Pulse**.
> * **Accessibility and Reproducibility:** We have verified that this method is **financially accessible** (covered by standard free usage tiers) and ensures **long-term reproducibility** via unique Panorama IDs—a reliable mechanism **validated by extensive street-view literature to guarantee data consistency over time**.
> * **Core Contribution:** Crucially, we emphasize that the raw street-view imagery itself is *not* our primary contribution; rather, the scientific value lies in the **CitySeeker framework**, the reasoning benchmark, and the BCR strategies. With the compliance barrier now removed, we respectfully submit that the work is positioned to be evaluated on its scientific contributions, which the reviewer explicitly recognized as ''clear accept-level''.
>
> **2.2 Regarding Potential Bias and Task Definition (Reviewer Xv86)**:
>
> Regrettably, we did not receive further engagement from Reviewer Xv86. However, we have robustly addressed their concerns regarding potential designer bias and behavioral priors through new empirical evidence:
> * **Empirical Validation (new large-scale survey):** We conducted a large-scale human consensus survey which yielded an **83.39% global consensus** on our mappings (vs. 1.90% noise), objectively demonstrating that our ground truth reflects human common sense rather than subjective bias.
> * **Scientific Scope:** We clarified that while GPS or social media data are valuable, they inherently confound *behavioral outcomes* with *ambiguous latent intent*, introducing **significant noise**. Our work explicitly targets **foundational spatial common sense**—the intrinsic ability to visually ground abstract needs. **Personalization** using behavioral priors is a distinct downstream task that builds upon this foundational capability, that models must first grasp universal affordances before optimizing for individual preferences. We have thus integrated this as the logical next step in our future work.
>
> ### **Conclusion**
>
> We trust in the Area Chair's fair and comprehensive judgment. We are grateful for the opportunity to refine our work through this rigorous review process. We believe that the resolved technical and ethical alignment, combined with the consensus on the benchmark's utility, positions CitySeeker to make a meaningful contribution to the field. We would be honored to share these findings with the ICLR community.
>
> Best regards,
>
> The Authors

---

### Meta-Review · Area_Chair_a3bm · 2026-01-05

**Summary:**

This paper introduces a new benchmark evaluating the spatial reasoning and decision-making capability of VLMs in embodied urban navigation scenarios, with a focus on addressing implicit needs. The reviewers raised several concerns and questions, including data ethics concerns, potential human design bias, and regional bias, as well as questions such as why open models perform better than proprietary models, presentation clarity, etc. After the rebuttal, most concerns and questions were addressed and confirmed by the reviewers. The outstanding one is the ethical concern regarding data compliance. Although the authors provide an alternative solution of downloading images via an API to comply with legal policy, this is still not ideal, as this process may not be free of charge. Overall, however, reviewers believe this is an interesting work.

**Reviewer Concerns:**

Reviewer agkw does not have major concerns, with a few questions regarding why open-sourced models perform better than proprietary VLMs and presentation clarity. The rebuttal clarifies these questions, and the reviewer raised the score to 8.

Reviewer 4hXJ points out a number of presentation issues. The rebuttal provides sufficient details and seems satisfactory to the reviewer.

Reviewer 1jEc has a major concern about data compliance, since the street images are from Google/Baidu. In the rebuttal, the authors argue that the images could be downloaded via APIs. While this seems compliant with the policy, it is not free of charge and is still not ideal.

Reviewer Xv86 lists a number of concerns about potential biases toward the dataset designer, region, culture, and linguistics, etc. The authors provide a detailed response explaining that the current curated dataset actually satisfies the reviewer’s requests. Additionally, the authors conducted a human consensus survey to show that human choices are mostly aligned with the generated benchmark labels. I believe this response would alleviate some of the questions and concerns, if not all.

**Reviewer Scores:**

Three reviewers have decided to raise their scores. The only reviewer who did not respond may also raise the score to 6, since the detailed response, including the new human study, would partially, if not fully, address the bias concerns.

---

### Decision · Program_Chairs · 2026-01-26

Accept (Poster)